# Editable Concept Bottleneck Models

Lijie Hu [* 1 2]   Chenyang Ren [* 1 2 3]   Zhengyu Hu [* 4 1]   Hongbin Lin [4 1]
Cheng-Long Wang [1 2]   Zhen Tan [5]   Weimin Lyu [6]   Jingfeng Zhang [7 1 2]   Hui Xiong [4]   Di Wang [1 2]

## Abstract

Concept Bottleneck Models (CBMs) have garnered much attention for their ability to elucidate the prediction process through a human-understandable concept layer. However, most previous studies focused on cases where the data, including concepts, are clean. In many scenarios, we often need to remove/insert some training data or new concepts from trained CBMs for reasons such as privacy concerns, data mislabelling, spurious concepts, and concept annotation errors. Thus, deriving efficient editable CBMs without retraining from scratch remains a challenge, particularly in large-scale applications. To address these challenges, we propose Editable Concept Bottleneck Models (ECBMs). Specifically, ECBMs support three different levels of data removal: concept-label-level, concept-level, and data-level. ECBMs enjoy mathematically rigorous closed-form approximations derived from influence functions that obviate the need for retraining. Experimental results demonstrate the efficiency and adaptability of our ECBMs, affirming their practical value in CBMs. Code is available on https://github.com/kaustpradalab/ECBM

## 1. Introduction

Modern deep learning models, such as large language models (Zhao et al., 2023; Yang et al., 2024a;b; Xu et al., 2023; Yang et al., 2024c) and large multimodal (Yin et al., 2023; Ali et al., 2024; Cheng et al., 2024), often exhibit intricate

non-linear architectures, posing challenges for end-users seeking to comprehend and trust their decisions. This lack of interpretability presents a significant barrier to adoption, particularly in critical domains such as healthcare (Ahmad et al., 2018; Yu et al., 2018) and finance (Cao, 2022), where transparency is paramount. To address this demand, explainable artificial intelligence (XAI) models (Das & Rad, 2020; Hu et al., 2023b;a) have emerged, offering explanations for their behavior and insights into their internal mechanisms. Among these, Concept Bottleneck Models (CBMs) (Koh et al., 2020) have gained prominence for explaining the prediction process of end-to-end AI models. CBMs add a bottleneck layer for placing human-understandable concepts. In the prediction process, CBMs first predict the concept labels using the original input and then predict the final classification label using the predicted concept in the bottleneck layer, which provides a self-explained decision to users.

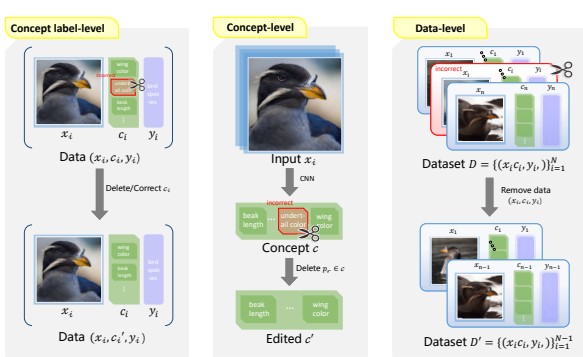

Figure 1: An illustration of Editable Concept Bottleneck Models with three settings.

Existing research on CBMs predominantly addresses two primary concerns: Firstly, CBMs heavily rely on laborious dataset annotation. Researchers have explored solutions to these challenges in unlabeled settings (Oikarinen et al., 2023; Yuksekgonul et al., 2023; Lai et al., 2023). Secondly, the performance of CBMs often lags behind that of original models lacking the concept bottleneck layer, attributed to incomplete information extraction from original data to bottleneck features. Researchers aim to bridge this utility gap (Sheth & Ebrahimi Kahou, 2023; Yuksekgonul et al., 2023; Espinosa Zarlenga et al., 2022). However, few of them considered the adaptivity or editability of CBMs, cru-

[*]Equal contribution [1]Provable Responsible AI and Data Analytics (PRADA) Lab [2]King Abdullah University of Science and Technology [3]Shanghai Jiao Tong University [4]Thrust of Artificial Intelligence, The Hong Kong University of Science and Technology (Guangzhou), China Department of Computer Science and Engineering, The Hong Kong University of Science and Technology, Hong Kong SAR, China [5]Arizona State University [6]Stony Brook University [7]The University of Auckland. Correspondence to: Di Wang <di.wang@kaust.edu.sa>.

*Proceedings of the 42nd International Conference on Machine Learning*, Vancouver, Canada. PMLR 267, 2025. Copyright 2025 by the author(s).

cial aspects encompassing annotation errors, data privacy considerations, or concept updates. Actually, these demands are increasingly pertinent in the era of large models. We delineate the editable setting into three key aspects (illustrated in Figure 1):

- *Concept-label-level:* In most scenarios, concept labels are annotated by humans or experts. Thus, it is unavoidable that there are some annotation errors, indicating that there is a need to correct some concept labels in a trained CBM.

- *Concept-level:* In CBMs, the concept set is pre-defined by LLMs or experts. However, in many cases, evolving situations demand concept updates, as evidenced by discoveries such as chronic obstructive pulmonary disease as a risk factor for lung cancer, and doctors have the requirements to add related concepts. For another example, recent research found a new factor, obesity (Sattar et al., 2020) are risky for severe COVID-19 and factors (e.g., older age, male gender, Asian race) are risk associated with COVID-19 infection (Rozenfeld et al., 2020). On the other hand, one may also want to remove some spurious or unrelated concepts for the task. This demand is even more urgent in some rapidly evolving domains like the pandemic.

- *Data-level:* Data issues can arise in CBMs when training data is erroneous or poisoned. For example, if a doctor identifies a case as erroneous or poisoned, this data sample becomes unsuitable for training. Therefore, it is essential to have the capability to completely delete such data from the learned models. We need such an editable model that can interact effectively with doctors.

The most direct way to address the above three problems is retraining from scratch on the data after correction. However, retraining models in such cases prove prohibitively expensive, especially in large models, which is resource-intensive and time-consuming. Therefore, developing an efficient method to approximate prediction changes becomes paramount. Providing users with an adaptive and editable CBM is both crucial and urgent.

We propose Editable Concept Bottleneck Models (ECBMs) to tackle these challenges. Specifically, compared to retraining, ECBMs provide a mathematically rigorous closed-form approximation for the above three settings to address editability within CBMs efficiently. Leveraging the influence function (Cook, 2000; Cook & Weisberg, 1980), we quantify the impact of individual data points, individual concept labels, and the concept for all data on model parameters. Despite the growing attention and utility of influence functions in machine learning (Koh & Liang, 2017), their application

in CBMs remains largely unexplored due to their composite structure, i.e., the intermediate representation layer.

To the best of our knowledge, we are the first to work to fill this gap by demonstrating the effectiveness of influence functions in elucidating the behavior of CBMs, especially in identifying mislabeled data and discerning the data influence. Comprehensive experiments on benchmark datasets show that our ECBMs are efficient and effective. Our contributions are summarized as follows.

- We delineate three different settings that need various levels of data or concept removal in CBMs: concept-label-level, concept-level, and data-level. To the best of our knowledge, our research marks the first exploration of data removal issues within CBMs.

- To make CBMs able to remove data or concept influence without retraining, we propose the Editable Concept Bottleneck Models (ECBMs). Our approach in ECBMs offers a mathematically rigorous closed-form approximation. Furthermore, to improve computational efficiency, we present streamlined versions integrating Eigenvalue-corrected Kronecker-Factored Approximate Curvature (EK-FAC).

- To showcase the effectiveness and efficiency of our ECBMs, we conduct comprehensive experiments across various benchmark datasets to demonstrate our superior performance.

## 2. Related Work

**Concept Bottleneck Models.** CBM (Koh et al., 2020) stands out as an innovative deep-learning approach for image classification and visual reasoning. It introduces a concept bottleneck layer into deep neural networks, enhancing model generalization and interpretability by learning specific concepts. However, CBM faces two primary challenges: its performance often lags behind that of original models lacking the concept bottleneck layer, attributed to incomplete information extraction from the original data to bottleneck features. Additionally, CBM relies on laborious dataset annotation. Researchers have explored solutions to these challenges. Chauhan et al. (2023) extend CBM into interactive prediction settings, introducing an interaction policy to determine which concepts to label, thereby improving final predictions. Oikarinen et al. (2023) address CBM limitations and propose a novel framework called Label-free CBM. This innovative approach enables the transformation of any neural network into an interpretable CBM without requiring labeled concept data, all while maintaining high accuracy. Post-hoc Concept Bottleneck models (Yuksekgonul et al., 2023) can be applied to various neural networks without compromising model performance, preserving interpretability advantages. CBMs work on the image field

also includes the works of Havasi et al. (2022),Kim et al. (2023),Keser et al. (2023),Sawada & Nakamura (2022) and Sheth & Kahou (2023). Despite many works on CBMs, we are the first to investigate the interactive influence between concepts through influence functions. Our research endeavors to bridge this gap by utilizing influence functions in CBMs, thereby deciphering the interaction of concept models and providing an adaptive solution to concept editing. For more related work, please refer to Appendix I.

## 3. Preliminaries

**Concept Bottleneck Models.** In this paper, we consider the original CBM, and we adopt the notations used by Koh et al. (2020). We consider a classification task with a concept set denoted as $\{p_1, \cdots, p_k\}$ with each $p_i$ being a concept given by experts or LLMs, and a training dataset represented as $\mathcal{D} = \{z_i\}_{i=1}^n$, where $z_i = (x_i, y_i, c_i)$. Here, for $i \in [n]$, $x_i \in \mathbb{R}^{d_i}$ represents the input feature vector, $y_i \in \mathbb{R}^{d_o}$ denotes the label (with $d_o$ corresponding to the number of classes) and $c_i = (c_i^1, \cdots, c_i^k) \in \mathbb{R}^k$ represents the concept vector. In this context, $c_i^j$ represents the label of the concept $p_j$ of the $i$-th data. In CBMs, our goal is to learn two representations: one called concept predictor that transforms the input space into the concept space, denoted as $g : \mathbb{R}_i^d \to \mathbb{R}^k$, and the other called label predictor which maps the concept space to the prediction space, denoted as $f : \mathbb{R}^k \to \mathbb{R}^{d_o}$. Usually, here the map $f$ is linear. For each training sample $z_i = (x_i, y_i, c_i)$, we consider two empirical loss functions: concept predictor $\hat{g}$ and label predictor $\hat{f}$:

$$\hat{g} = \arg\min_g \sum_{i=1}^n \sum_{j=1}^k g^j(x_i)^\top \log(c_i^j), \qquad (1)$$

where $g^j(*)$ is the predicted $j$-th concept. For brevity, write the loss function as $L_C(g(x_i), c_i) = \sum_{j=1}^k L_C^j(g(x_i), c_i)$ for data $(x_i, c_i)$. Once we obtain the concept predictor $\hat{g}$, the label predictor is defined as:

$$\hat{f} = \arg\min_f \sum_{i=1}^n L_Y\big(f(\hat{g}(x_i)), y_i\big), \qquad (2)$$

where $L_Y$ represents the cross-entropy loss, similar to (1). CBMs enforce dual precision in predicting interpretable concept vectors $\hat{c} = \hat{g}(x)$ (matching concept $c$) and final outputs $\hat{y} = \hat{f}(\hat{c})$ (matching label $y$), ensuring transparent reasoning through explicit concept mediation. Furthermore, in this paper, we focus primarily on the scenarios in which the label predictor $f$ is a linear transformation, motivated by their interpretability advantages in tracing concept-to-label relationships. For details on the symbols used, please refer to the notation table in Appendix 2.

**Influence Function.** The influence function measures the dependence of an estimator on the value of individ-

ual point in the sample. Consider a neural network $\hat{\theta} = \arg\min_\theta \sum_{i=1}^n \ell(z_i; \theta)$ with loss function $\ell$ and dataset $D = \{z_i\}_{i=1}^n$. If we remove $z_m$ from the training dataset, the parameters become $\hat{\theta}_{-z_m} = \arg\min_\theta \sum_{i \neq m} \ell(z_i; \theta)$. The influence function provides an efficient model approximation by defining a series of $\epsilon$-parameterized models as $\hat{\theta}_{\epsilon, -z_m} = \operatorname{argmin} \sum_{i=1}^n \ell(z_i; \theta) + \epsilon\ell(z_m; \theta)$. By performing a first-order Taylor expansion on the gradient of the objective function corresponding to the $\arg\min$ process, the influence function is defined as:

$$\mathcal{I}_{\hat{\theta}}(z_m) \triangleq \left. \frac{\mathrm{d}\hat{\theta}_{\epsilon, -z_m}}{\mathrm{d}\epsilon} \right|_{\epsilon=0} = -H_{\hat{\theta}}^{-1} \cdot \nabla_\theta \ell(z_m; \hat{\theta}),$$

where $H_{\hat{\theta}}^{-1} = \nabla_\theta^2 \sum_{i=1}^n \ell(z_i; \hat{\theta})$ is the Hessian matrix. When the loss function $\ell$ is twice-differentiable and strongly convex in $\theta$, the Hessian $H_{\hat{\theta}}$ is positive definite and thus the influence function is well-defined. For non-convex loss functions, Bartlett (1953) proposed replacing the Hessian $H_{\hat{\theta}}$ with $\hat{H} = G_{\hat{\theta}} + \delta I$, where $G_{\hat{\theta}}$ is the Fisher information matrix defined as $\sum_{i=1}^n \nabla_\theta \ell(z_i; \hat{\theta})^\top \nabla_\theta \ell(z_i; \theta)$, and $\delta$ is the damping term used to ensure the positive definiteness of $\hat{H}$. We can employ the Eigenvalue-corrected Kronecker-Factored Approximate Curvature (EK-FAC) method to further accelerate the computation. See Appendix C for additional details.

## 4. Editable Concept Bottleneck Models

In this section, we introduce our EBCMs for the three settings mentioned in the introduction, leveraging the influence function. Specifically, at the concept-label level, we calculate the influence of a set of data samples' individual concept labels; at the concept level, we calculate the influence of multiple concepts; and at the data level, we calculate the influence of multiple samples.

### 4.1. Concept Label-level Editable CBM

In many cases, certain data samples contain erroneous annotations for specific concepts, yet their other information remains valuable. This is particularly relevant in domains such as medical imaging, where acquiring data is often costly and time-consuming. In such scenarios, it is common to correct the erroneous concept annotations rather than removing the entire data from the dataset. Estimating the retrained model parameter is crucial in this context. We refer to this scenario as the concept label-level editable CBM.

Mathematically, we have a set of erroneous data $D_e$ and its associated index set $S_e \subseteq [n] \times [k]$ such that for each $(w, r) \in S_e$, $(x_w, y_w, c_w) \in D_e$ with $c_w^r$ is mislabeled and $\tilde{c}_w^r$ is corrected concept label. Our goal is to estimate the retrained CBM. The retrained concept predictor and label

predictor are represented as follows:

$$\hat{g}_e = \arg\min_{g} \sum_{(i,j)\notin S_e} L_C^j\left(g(x_i), c_i\right)$$
$$+ \sum_{(i,j)\in S_e} L_C^j\left(g(x_i), \tilde{c}_i\right), \tag{3}$$

$$\hat{f}_e = \arg\min_{f} \sum_{i=1}^{n} L_Y\left(f\left(\hat{g}_e\left(x_i\right)\right), y_i\right). \tag{4}$$

For simple neural networks, we can use the influence function approach directly to estimate the retrained model. However, for CBM architecture, if we intervene with the true concepts, the concept predictor $\hat{g}$ fluctuates to $\hat{g}_e$ accordingly. Observe that the input data of the label predictor comes from the output of the concept predictor, which is also subject to change. Therefore, we need to adopt a two-stage editing approach. Here we consider the influence function for (3) and (4) separately. We first edit the concept predictor from $\hat{g}$ to $\bar{g}_e$, and then edit from $\hat{f}$ to $\bar{f}_e$ based on our approximated concept predictor. To begin, we provide the following definitions:

**Definition 4.1.** Define the gradient of the $j$-th concept predictor and the label predictor for the $i$-th data point $x_i$ as:

$$G_C^j(x_i, c_i; g) \triangleq \nabla_g L_C^j\left(g(x_i), c_i\right),$$
$$G_Y(x_i; g, f) \triangleq \nabla_f L_Y\left(f(g(x_i)), y_i\right).$$

**Theorem 4.2.** *The retrained concept predictor $\hat{g}_e$ defined by (3) can be approximated by $\bar{g}_e$, defined by:*

$$\hat{g} - H_{\hat{g}}^{-1} \cdot \sum_{(w,r)\in S_e} \left(G_C^r(x_w, \tilde{c}_w; \hat{g}) - G_C^r(x_w, c_w; \hat{g})\right),$$

*where $H_{\hat{g}} = \nabla_{\hat{g}} \sum_{i,j} G_C^j(x_i, c_i; \hat{g})$ is the Hessian matrix of the loss function with respect to $\hat{g}$.*

**Theorem 4.3.** *The retrained label predictor $\hat{f}_e$ defined by (4) can be approximated by $\bar{f}_e$, defined by:*

$$\hat{f} + H_{\hat{f}}^{-1} \cdot \sum_{i=1}^{n} \left(G_Y(x_i; \hat{g}, \hat{f}) - G_Y(x_i; \bar{g}_e, \hat{f})\right),$$

*where $H_{\hat{f}} = \nabla_{\hat{f}} \sum_{i=1}^{n} G_Y(x_i; \hat{g}, \hat{f})$ is the Hessian matrix, and $\bar{g}_e$ is given in Theorem 4.2.*

**Difference from Test-Time Intervention.** The ability to intervene in CBMs allows human users to interact with the model during the prediction process. For example, a medical expert can directly replace an erroneously predicted concept value $\hat{c}$ and observe its impact on the final prediction $\hat{y}$. However, the underlying flaws in the concept predictor remain unaddressed, meaning similar errors may persist when applied to new test data. In contrast, under the editable

CBM framework, not only can test-time interventions be performed, but the concept predictor of the CBM can also be further refined based on test data that repeatedly produces errors. Our ECBM method incorporates the corrected test data into the training dataset without requiring full retraining. This approach extends the rectification process from the data level to the model level.

## 4.2. Concept-level Editable CBM

In this case, a set of concepts is removed due to incorrect attribution or spurious concepts, termed concept-level edit. [1]Specifically, for the concept set, denote the erroneous concept index set as $M \subset [k]$, we aim to delete these concept labels in all training samples. We aim to investigate the impact of updating the concept set within the training data on the model's predictions. It is notable that compared to the above concept label case, the dimension of output (input) of the retrained concept predictor (label predictor) will change. If we delete $t$ concepts from the dataset, then $g$ becomes $g' : \mathbb{R}^{d_i} \to \mathbb{R}^{k-t}$ and $f$ becomes $f' : \mathbb{R}^{k-t} \to \mathbb{R}^{d_o}$. More specifically, if we retrain the CBM with the revised dataset, the corresponding concept predictor becomes:

$$\hat{g}_{-p_M} = \arg\min_{g'} \sum_{j\notin M} \sum_{i=1}^{n} L_C^j(g'(x_i), c_i). \tag{5}$$

The variation of the parameters in dimension renders the application of influence function-based editing challenging for the concept predictor. This is because the influence function implements the editorial predictor by approximate parameter change from the original base after $\epsilon$-weighting the corresponding loss for a given sample, and thus, it is unable to deal with changes in parameter dimensions.

To overcome the challenge, our strategy is to develop some transformations that need to be performed on $\hat{g}_{-p_M}$ to align its dimension with $\hat{g}$ so that we can apply the influence function to edit the CBM. We achieve this by mapping $\hat{g}_{-p_M}$ to $\hat{g}_{-p_M}^* \triangleq \mathrm{P}(\hat{g}_{-p_M})$, which has the same amount of parameters as $\hat{g}$ and has the same predicted concepts $\hat{g}_{-p_M}^*(j)$ as $\hat{g}_{-p_M}(j)$ for all $j \in [d_i] - M$. We achieve this effect by inserting a zero row vector into the $r$-th row of the matrix in the final layer of $\hat{g}_{-p_M}$ for $r \in M$. Thus, we can see that the mapping $P$ is one-to-one. Moreover, assume the parameter space of $\hat{g}$ is $T$ and that of $\hat{g}_{-p_M}^*$, $T_0$ is the subset of $T$. Noting that $\hat{g}_{-p_M}^*$ is the optimal model of the following objective function:

$$\hat{g}_{-p_M}^* = \arg\min_{g'\in T_0} \sum_{j\notin M} \sum_{i=1}^{n} L_C^j(g'(x_i), c_i), \tag{6}$$

i.e., it is the optimal model of the concept predictor loss on the remaining concepts under the constraint $T_0$. Now we

---

[1]For convenience, in this paper, we only consider concept removal; our method can directly extend to concept insertion.

can apply the influence function to edit $\hat{g}$ to approximate $\hat{g}^*_{-p_M}$ with the restriction on the value of 0 for rows indexed by $M$ with the last layer of the neural network, denoted as $\bar{g}^*_{-p_M}$. After that, we remove from $\bar{g}^*_{-p_M}$ the parameters initially inserted to fill in the dimensional difference, which always equals 0 because of the restriction we applied in the editing stage, thus approximating the true edited concept predictor $\hat{g}_{-p_M}$. We now detail the editing process from $\hat{g}$ to $\hat{g}^*_{-p_M}$ using the following theorem.

**Theorem 4.4.** *For the retrained concept predictor $\hat{g}_{-p_M}$ defined in (5), we map it to $\hat{g}^*_{-p_M}$ as (6). And we can edit the initial $\hat{g}$ to $\hat{g}^*_{-p_M}$, defined as:*

$$\bar{g}^*_{-p_M} \triangleq \hat{g} - H_{\hat{g}}^{-1} \cdot \sum_{j \notin M} \sum_{i=1}^{n} G_C^j(x_i, c_i; \hat{g}),$$

*where $H_{\hat{g}} = \nabla_g \sum_{j \notin M} \sum_{i=1}^{n} G_C^j(x_i, c_i; \hat{g})$. Then, by removing all zero rows inserted during the mapping phase, we can naturally approximate $\hat{g}_{-p_M} \approx \mathrm{P}^{-1}(\hat{g}^*_{-p_M})$.*

For the second stage of training, assume we aim to remove concept $p_r$ for $r \in M$ and the new optimal model is $\hat{f}_{-p_M}$. We will encounter the same difficulty as in the first stage, i.e., the number of parameters of the label predictor will change. To address the issue, our key observation is that in the existing literature on CBMs, we always use linear transformation for the label predictor, meaning that the dimensions of the input with values of 0 will have no contribution to the final prediction. To leverage this property, we fill the missing values in the input of the updated predictor with 0, that is, replacing $\hat{g}_{-p_M}$ with $\hat{g}^*_{-p_M}$ and consider $\hat{f}_{p_M=0}$ defined by

$$\hat{f}_{p_M=0} = \arg\min_f \sum_{i=1}^{n} L_Y\left(f\left(\hat{g}^*_{-p_M}(x_i)\right), y_i\right). \quad (7)$$

In total, we have the following lemma:

**Lemma 4.5.** *In the CBM, if the label predictor utilizes linear transformations of the form $\hat{f} \cdot c$ with input $c$, then, for each $r \in M$, we remove the $r$-th concept from $c$ and denote the new input as $c'$; set the $r$-th concept to 0 and denote the new input as $c^0$. Then we have $\hat{f}_{-p_M} \cdot c' = \hat{f}_{p_M=0} \cdot c^0$ for any input $c$.*

Lemma 4.5 demonstrates that the retrained $\hat{f}_{-p_M}$ and $\hat{f}_{p_M=0}$, when given inputs $\hat{g}_{-p_M}(x)$ and $\hat{g}^*_{-p_M}(x)$ respectively, yield identical outputs. Consequently, we can utilize $\hat{f}_{p_M=0}$ as the editing target in place of $\hat{f}_{-p_M}$.

**Theorem 4.6.** *For the revised retrained label predictor $\hat{f}_{p_M=0}$ defined by (7), we can edit the initial label predictor $\hat{f}$ to $\bar{f}_{p_M=0}$ by the following equation as a substitute for $\hat{f}_{p_M=0}$:*

$$\hat{f}_{p_M=0} \approx \bar{f}_{p_M=0} \triangleq \hat{f} - H_{\hat{f}}^{-1} \cdot \sum_{l=1}^{n} G_Y(x_l; \bar{g}^*_{-p_M}, \hat{f}),$$

*where $H_{\hat{f}} = \nabla_{\hat{f}} \sum_{i=1}^{n} G_Y(x_l; \bar{g}^*_{-p_M}, \hat{f})$ is the Hessian matrix. Deleting the $r$-th dimension of $\bar{f}_{p_M=0}$ for $r \in M$, then we can map it to $\bar{f}_{-p_M}$, which is the approximation of the final edited label predictor $\hat{f}_{-p_M}$ under concept level.*

## 4.3. Data-level Editable CBM

In this scenario, we are more concerned about fully removing the influence of data samples on CBMs due to different reasons, such as the training data involving poisoned or erroneous issues. Specifically, we have a set of samples to be removed $\{(x_i, y_i, c_i)\}_{i \in G}$ with $G \subset [n]$. Then, we define the retrained concept predictor as

$$\hat{g}_{-z_G} = \arg\min_g \sum_{j=1}^{k} \sum_{i \in [n]-G} L_C^j(g(x_i), c_i), \quad (8)$$

which can be evaluated by the following theorem:

**Theorem 4.7.** *For dataset $\mathcal{D} = \{(x_i, y_i, c_i)\}_{i=1}^{n}$, given a set of data $z_r = (x_r, y_r, c_r)$, $r \in G$ to be removed. Suppose the updated concept predictor $\hat{g}_{-z_G}$ is defined by (8), then we have the following approximation for $\hat{g}_{-z_G}$*

$$\hat{g}_{-z_G} \approx \bar{g}_{-z_G} \triangleq \hat{g} + H_{\hat{g}}^{-1} \cdot \sum_{r \in G} \sum_{j=1}^{M} G_C^j(x_r, c_r; \hat{g}), \quad (9)$$

*where $H_{\hat{g}} = \nabla_g \sum_{i,j} G_C^j(x_i, c_i; \hat{g})$ is the Hessian matrix of the loss function with respect to $\hat{g}$.*

Based on $\hat{g}_{-z_G}$, the label predictor becomes $\hat{f}_{-z_G}$ which is defined by

$$\hat{f}_{-z_G} = \arg\min_f \sum_{i \in [n]-G} L_Y\left(f(\hat{g}_{-z_G}(x_i), y_i)\right). \quad (10)$$

Compared with the original loss before unlearning in (2), we can observe two changes in (10). First, we remove $|G|$ data points in the loss function $L_Y$. Secondly, the input for the loss is also changed from $\hat{g}(x_i)$ to $\hat{g}_{-z_G}$. Therefore, it is difficult to estimate directly with an influence function. Here we introduce an intermediate label predictor as

$$\tilde{f}_{-z_G} = \arg\min \sum_{i \in [n]-G} L_Y(f(\hat{g}(x_i), y_i), \quad (11)$$

and split the estimate of $\hat{f}_{-z_G} - \hat{f}$ into $\hat{f}_{-z_G} - \tilde{f}_{-z_G}$ and $\tilde{f}_{-z_G} - \hat{f}$.

**Theorem 4.8.** *For dataset $\mathcal{D} = \{(x_i, y_i, c_i)\}_{i=1}^{n}$, given a set of data $z_r = (x_r, y_r, c_r)$, $r \in G$ to be removed. The intermediate label predictor $\tilde{f}_{-z_G}$ is defined in (11). Then we have*

$$\tilde{f}_{-z_G} - \hat{f} \approx H_{\hat{f}}^{-1} \sum_{i \in [n]-G} G_Y(x_i; \hat{g}, \hat{f}) \triangleq A_G.$$

We denote the edited version of $\tilde{f}_{-z_G}$ as $\bar{f}^*_{-z_G} \triangleq \hat{f} + A_G$. Define $B_G$ as

$$-H^{-1}_{\bar{f}^*_{-z_G}} \sum_{i \in [n]-G} G_Y(x_i; \bar{g}_{-z_G}, \bar{f}^*_{-z_G}) - G_Y(x_i; \hat{g}, \bar{f}^*_{-z_G}),$$

where $H_{\bar{f}^*_{-z_G}} = \nabla_{\bar{f}} \sum_{i \in [n]-G} G_Y(x_i; \hat{g}, \bar{f}^*_{-z_G})$ is the Hessian matrix concerning $\bar{f}^*_{-z_G}$. Then $\hat{f}_{-z_G}$ can be estimated by $\tilde{f}_{-z_G} + B_G$. Combining the above two-stage approximation, then, the final edited label predictor $\bar{f}_{-z_G}$ can be obtained by

$$\bar{f}_{-z_G} = \bar{f}^*_{-z_G} + B_G = \hat{f} + A_G + B_G. \quad (12)$$

**Acceleration via EK-FAC.** As mentioned in Section 3, the loss function in CBMs is non-convex, meaning the Hessian matrices in all our theorems may not be well-defined. To address this, we adopt the EK-FAC approach, where the Hessian is approximated as $\hat{H}_\theta = G_\theta + \delta I$. Here, $G_\theta$ represents the Fisher information matrix of the model $\theta$, and $\delta$ is a small damping term introduced to ensure positive definiteness. For details on applying EK-FAC to CBMs, see Appendix C.1. Additionally, refer to Algorithms 6-8 in the Appendix for the EK-FAC-based algorithms corresponding to our three levels, with their original (Hessian-based) versions provided in Algorithms 1-3, respectively.

**Theoretical Bounds.** We provide error bounds for the concept predictor between retraining and ECBM across all three levels; see Appendix D.2, E.2 and F.2 for details. We show that under certain scenarios, the approximation error becomes tolerable theoretically when leveraging some damping term $\delta$ regularized in the Hessian matrix.

## 5. Experiments

In this section, we demonstrate our main experimental results on utility evaluation, edition efficiency, and interpretability evaluation. Details and additional results are in Appendix H due to space limit.

### 5.1. Experimental Settings

**Dataset.** We utilize three datasets: *X-ray Grading (OAI)* (Nevitt et al., 2006), *Bird Identification (CUB)* (Wah et al., 2011), and the *Large-scale CelebFaces Attributes Dataset (CelebA)* (Liu et al., 2015). OAI is a multi-center observational study of knee osteoarthritis, comprising 36,369 data points. Specifically, we configure n=10 concepts that characterize crucial osteoarthritis indicators such as joint space narrowing, osteophytes, and calcification. Bird identification (CUB)[2] consists of 11,788 data points, which belong to 200 classes and include 112 binary attributes to

describe detailed visual features of birds. CelebA comprises 202,599 celebrity images, each annotated with 40 binary attributes that detail facial features, such as hair color, eyeglasses, and smiling. As the dataset lacks predefined classification tasks, following (Espinosa Zarlenga et al., 2022), we designate 8 attributes as labels and the remaining 32 attributes as concepts. For all the above datasets, we follow the same network architecture and settings outlined in (Koh et al., 2020).

**Ground Truth and Baselines.** We use retrain as the ground truth method. *Retrain*: We retrain the CBM from scratch by removing the samples, concept labels, or concepts from the training set. We employ two baseline methods: CBM-IF, and ECBM. *CBM-IF*: This method is a direct implementation of our previous theorems of model updates in the three settings. See Algorithms 1-3 in Appendix for details. *ECBM*: As we discussed above, all of our model updates can be further accelerated via EK-FAC, ECBM corresponds to the EK-FAC accelerated version of Algorithms 1-3 (refer to Algorithms 6-8 in Appendix).

**Evaluation Metric.** We utilize two primary evaluation metrics to assess our models: the F1 score and runtime (RT). *F1 score* measures the model performance by balancing precision and recall. *Runtime*, measured in minutes, evaluates the total running time of each method to update the model.

**Implementation Details.** Our experiments utilized an Intel Xeon CPU and an RTX 3090 GPU. For utility evaluation, at the concept level, one concept was randomly removed for the OAI dataset and repeated while ten concepts were randomly removed for the CUB dataset, with five different seeds. At the data level, 3% of the data points were randomly deleted and repeated 10 times with different seeds. At the concept-label level, we randomly selected 3% of the data points and modified one concept of each data randomly, repeating this 10 times for consistency across iterations.

### 5.2. Evaluation of Utility and Editing Efficiency

Our experimental results, as illustrated in Table 1, demonstrate the effectiveness of ECBMs compared to traditional retraining and CBM-IF, particularly emphasizing computational efficiency without compromising accuracy. Specifically, ECBMs achieved F1 scores close to those of retraining (0.8808 vs. 0.8825) while significantly reducing the runtime from 297.77 minutes to 2.36 minutes. This pattern is consistent in the CUB dataset, where the runtime was decreased from 85.56 minutes for retraining to 0.65 minutes for ECBMs, with a negligible difference in the F1 score (0.7971 to 0.7963). These results highlight the potential of ECBMs to provide substantial time savings—approximately 22-30% of the computational time required for retraining—while maintaining comparable accuracy. Compared to CBM-IF, ECBM also showed a slight

---

[2]The original dataset is processed. Detailed explanation can be found in H.

Table 1: Performance comparison of different methods on the three datasets.

| Edit Level | Method | OAI | | CUB | | CelebA | |
|---|---|---|---|---|---|---|---|
| | | F1 score | RT (minute) | F1 score | RT (minute) | F1 score | RT (minute) |
| Concept Label | Retrain | 0.8825±0.0054 | 297.77±35.01 | 0.7971±0.0066 | 85.56±4.22 | 0.3827±0.0272 | 304.71±35.15 |
| | CBM-IF(Ours) | 0.8639±0.0033 | 4.63±0.89 | 0.7699±0.0035 | 1.33±0.09 | 0.3561±0.0134 | 5.54±0.82 |
| | ECBM(Ours) | **0.8808±0.0039** | **2.36±0.54** | **0.7963±0.0050** | **0.65±0.08** | **0.3845±0.0327** | **2.49±0.53** |
| Concept | Retrain | 0.8448±0.0191 | 258.84±42.48 | 0.7811±0.0047 | 87.21±7.62 | 0.3776±0.0350 | 355.85±25.39 |
| | CBM-IF(Ours) | 0.8214±0.0071 | 4.94±0.91 | 0.7579±0.0065 | 1.45±0.08 | 0.3609±0.0202 | 5.51±0.89 |
| | ECBM(Ours) | **0.8403±0.0090** | **2.36±0.60** | **0.7787±0.0058** | **0.59±0.17** | **0.3761±0.0280** | **2.48±0.52** |
| Data | Retrain | 0.8811±0.0065 | 319.37±31.08 | 0.7838±0.0051 | 86.20±7.74 | 0.3797±0.0375 | 325.62±50.59 |
| | CBM-IF(Ours) | 0.8472±0.0046 | 5.07±0.75 | 0.7623±0.0031 | 1.46±0.08 | 0.3536±0.0166 | 5.97±0.75 |
| | ECBM(Ours) | **0.8797±0.0038** | **2.50±0.49** | **0.7827±0.0088** | **0.65±0.19** | **0.3748±0.0347** | **2.49±0.61** |

reduction in runtime and a significant improvement in F1 score. The former verifies the effective acceleration of our algorithm by EK-FAC. This efficiency is particularly crucial in scenarios where frequent updates to model annotations are needed, confirming the utility of ECBMs in dynamic environments where running time and accuracy are critical.

We can also see that the original version of ECBM, i.e., CBM-IF, also has a lower runtime than retraining but a lower F1 score than ECBM. Such results may be due to different reasons. For example, our original theorems depend on the inverse of the Hessian matrices, which may not be well-defined for non-convex loss. Moreover, these Hessian matrices may be ill-conditioned or singular, which makes calculating their inverse imprecise and unstable.

**Editing Multiple Samples.** To comprehensively evaluate the editing capabilities of ECBM in various scenarios, we conducted experiments on the performance with multiple samples that need to be removed. Specifically, for the concept label/data levels, we consider the different ratios of samples (1-10%) for edit, while for the concept level, we consider removing different numbers of concepts $\in \{2, 4, 6, \cdots, 20\}$. We compared the performance of retraining, CBM-IF, and ECBM methods. As shown in Figure 2, except for certain cases at the concept level, the F1 score of the ECBM method is generally around 0.0025 lower than that of the retrain method, which is significantly better than the corresponding results of the CBM-IF method. Recalling Table 1, the speed of ECBM is more than three times faster than that of retraining. Consequently, ECBM is an editing method that achieves a trade-off between speed and effectiveness.

### 5.3. Results on Interpretability

**ECBM can measure concepts importance.** The original motivation of the influence function is to calculate the importance score of each sample. Here, we will show that the influence function for the concept level in Theorem 4.4 can be used to calculate the importance of each concept in CBMs, which provides an explainable tool for CBMs. In detail, we conduct our experiments on the CUB dataset. We first select 1-10 most influential and 1-10 least influential concepts by our influence function. Then, we will remove these concepts and update the model via retraining or our ECBM and analyze the change (F1 Score Difference) w.r.t. the original CBM before removal.

The results in Figure 3(a) demonstrate that when we remove the 1-10 most influential concepts identified by the ECBM method, the F1 score decreases by more than 0.025 compared to the CBM before removal. In contrast, Figure 3(b) shows that the change in the F1 score remains consistently below 0.005 when removing the least influential concepts. These findings strongly indicate that the influence function in ECBM can successfully determine the importance of concepts. Furthermore, we observe that the gap between the F1 score of retraining and ECBM is consistently smaller than 0.005, and even smaller in the case of least important concepts. This further suggests that when ECBM edits various concepts, its performance is very close to the ground truth.

**ECBMs can erase data influence.** For the data level, ECBMs aim to facilitate an efficient removal of samples. We perform membership inference attacks (MIAs) to provide direct evidence that ECBMs can indeed erase data influence. MIA is a privacy attack that aims to infer whether a specific data sample was part of the training dataset used to train a model. The attacker exploits the model's behavior, such as overconfidence or overfitting, to distinguish between *training (member)* and *non-training (non-member)* data points. In MIAs, the attacker typically queries the model with a data sample and observes its prediction confidence or loss values, which tend to be higher for members of the training set than non-members (Shokri et al., 2017).

To quantify the success of these edits, we calculate the RMIA (Removed Membership Inference Attack) score for each category. The RMIA score is defined as the model's confidence in classifying whether a given sample belongs to the training set. Lower RMIA values indicate that the sample behaves more like a test set (non-member) sample (Zarifzadeh et al., 2024). This metric is especially crucial for edited samples, as a successful ECBM should make the removed members behave similarly to non-members,

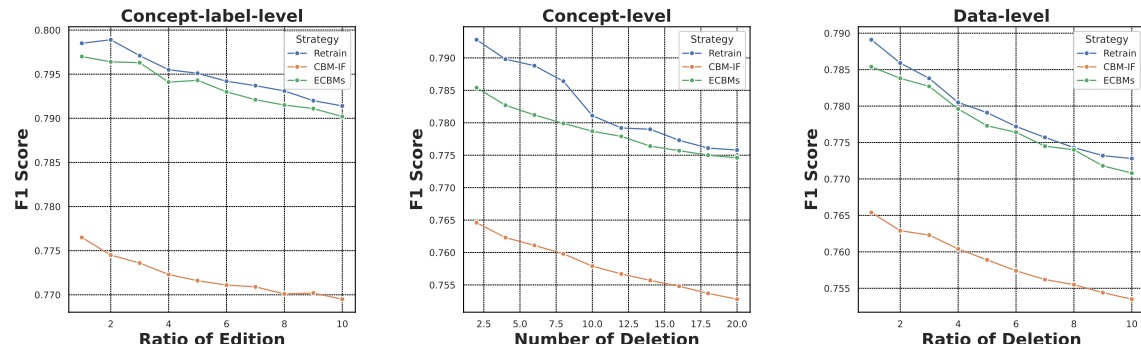

Figure 2: Impact of edition ratio on three settings on CUB dataset.

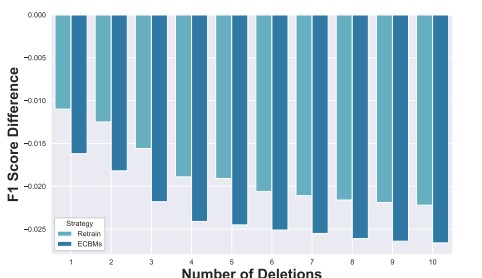

(a) Results on the 1-10 most influential concepts

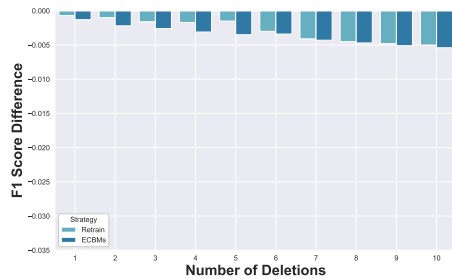

(b) Results on the 1-10 least influential concepts

Figure 3: F1 score difference after removing most and least influential concepts given by ECBM.

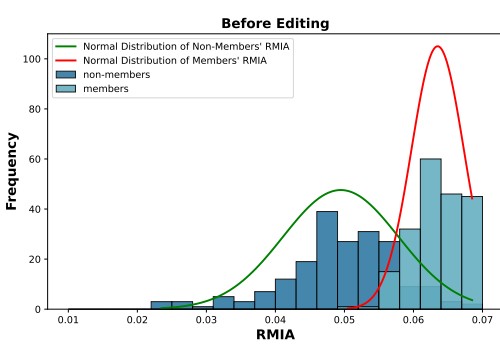

(a) RMIA Score Before Editing

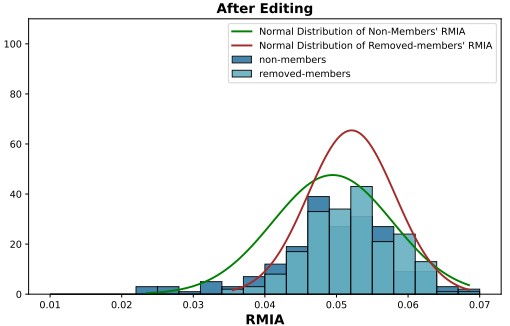

(b) RMIA Score After Editing

Figure 4: RMIA scores of data before and after removal.

reducing their membership vulnerability. See Appendix H for its definition.

We conducted experiments by randomly selecting 200 samples from the training set (members) and 200 samples from the test set (non-members) of the CUB dataset. We calculated the RMIA scores for these samples and plotted their frequency distributions, as shown in Figure 4(a). The mean RMIA score for non-members was 0.049465, while members had a mean score of 0.063505. Subsequently, we applied ECBMs to remove the 200 training samples from the model, updated the model parameters, and then recalculated the RMIA scores. After editing, the mean RMIA score for the removed members decreased to 0.052105, significantly

closer to the non-members' mean score. This shift in RMIA values demonstrates the effectiveness of ECBMs in editing the model, as the removed members now exhibit behavior closer to that of non-members. The post-editing RMIA score distributions are shown in Figure 4(b). These results provide evidence of the effectiveness of ECBMs in editing the model's knowledge about specific samples.

## 6. Conclusion

In this paper, we propose Editable Concept Bottleneck Models (ECBMs). ECBMs can address issues of removing/inserting some training data or new concepts from

trained CBMs for different reasons, such as privacy concerns, data mislabelling, spurious concepts, and concept annotation errors retraining from scratch. Furthermore, to improve computational efficiency, we present streamlined versions integrating EK-FAC. Experimental results show our ECBMs are efficient and effective.

## Impact Statement

This research propose editable concept bottleneck models. While we acknowledge the importance of evaluating societal impacts, our analysis suggests that there are no immediate ethical risks requiring specific mitigation measures beyond standard practices in machine learning.

## Acknowledgements

Lijie Hu, Chenyang Ren, Cheng-Long Wang, and Di Wang are supported in part by the funding BAS/1/1689-01-01, URF/1/4663-01-01, REI/1/5232-01-01, REI/1/5332-01-01, and URF/1/5508-01-01 from KAUST, and funding from KAUST - Center of Excellence for Generative AI, under award number 5940. Zhengyu Hu, Hongbin Lin, and Hui Xiong are supported by the National Key R&D Program of China (Grant No. 2023YFF0725001), the National Natural Science Foundation of China (Grant No. 92370204), the Guangdong Basic and Applied Basic Research Foundation (Grant No. 2023B1515120057), the Guangzhou-HKUST(GZ) Joint Funding Program (Grant No. 2023A03J0008), and the Education Bureau of Guangzhou Municipality.

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

# A. Notation Table

Table 2: Notation Table.

| Symbol | Description |
| --- | --- |
| $c = \{p_1, \ldots, p_k\}$ | Set of concepts provided by experts or LLMs. |
| $\mathcal{D} = \{z_i\}_{i=1}^n$ | Training dataset, where $z_i = (x_i, y_i, c_i)$. |
| $x_i \in \mathbb{R}^{d_i}$ | Feature vector for the $i$-th sample. |
| $y_i \in \mathbb{R}^{d_o}$ | Label for the $i$-th sample, with $d_z$ being the number of classes. |
| $c_i = (c_i^1, \ldots, c_i^k) \in \mathbb{R}^k$ | Concept vector for the $i$-th sample. |
| $\tilde{c}_w^r$ | Corrected concept label for the $w$-th sample and $r$-th concept. |
| $c_i^j$ | Weight of the concept $p_j$ in the concept vector $c_i$. |
| $g : \mathbb{R}_i^d \to \mathbb{R}^k$ | Concept predictor mapping input space to concept space. |
| $f : \mathbb{R}^k \to \mathbb{R}^{d_o}$ | Label predictor mapping concept space to prediction space. |
| $L_C^j(g(x), c)$ | Loss function for the $j$-th concept predictor. |
| $L_Y(f(\hat{g}(x)), y)$ | Loss function from concept space to output space. |
| $G_C^j(x_i, c_i; g)$ | Gradient of the loss $L_C^j(g(x_i), c_i)$. |
| $G_Y(x_i, c_i; g)$ | Gradient of the loss $L_Y(f(g(x_i)), c_i)$. |
| $H_{\hat{\theta}}$ | Hessian matrix of the loss function with respect to $\hat{\theta}$. |
| $G_{\hat{\theta}}$ | Fisher information matrix of model $\hat{\theta}$. |
| $\delta$ | Damping term for ensuring positive definiteness of the Hessian. |
| $\hat{g}$ | Estimated concept predictor. |
| $\hat{f}$ | Estimated label predictor. |
| $\hat{g}_e$ | Retrained concept predictor after correcting erroneous data. |
| $\hat{f}_e$ | Retrained label predictor after correcting erroneous data. |
| $\hat{g}_{-p_M}$ | Retrained concept predictor after removing concepts indexed by $M$. |
| $\hat{g}_{-p_M}^*$ | Mapped concept predictor with the same dimensionality as $\hat{g}$. |
| $\bar{g}_{-p_M}$ | Approximation of the retrained concept predictor $\hat{g}_{-p_M}$. |
| $\hat{f}_{p_M=0}$ | Label predictor after setting the $r$-th concept to zero for $r \in M$. |
| $\bar{f}_{p_M=0}$ | Approximation of the label predictor $\hat{f}_{p_M=0}$. |
| $H_{\hat{g}}$ | Hessian matrix of the loss function with respect to $\hat{g}$. |
| $H_{\hat{f}}$ | Hessian matrix of the loss function with respect to $\hat{f}$. |
| $M \subset [k]$ | Set of erroneous concept indices to be removed. |
| $G \subset [n]$ | Set of indices of samples to be removed from the dataset. |
| $z_r = (x_r, y_r, c_r)$ | Data sample to be removed, where $r \in G$. |
| $\hat{g}_{-z_G}$ | Retrained concept predictor after removing samples indexed by $G$. |
| $\bar{g}_{-z_G}$ | Approximation of the retrained concept predictor $\hat{g}_{-z_G}$. |
| $\hat{f}_{-z_G}$ | Intermediate label predictor. |
| $\bar{f}_{-z_G}$ | Final edited label predictor after removing samples indexed by $G$. |

# B. Influence Function

Consider a neural network $\hat{\theta} = \arg\min_\theta \sum_{i=1}^n \ell(z_i, \theta)$ with loss function $L$ and dataset $D = \{z_i\}_{i=1}^n$. That is $\hat{\theta}$ minimize the empirical risk

$$R(\theta) = \sum_{i=1}^n L(z_i, \theta)$$

Assume $R$ is strongly convex in $\theta$. Then $\theta$ is uniquely defined. If we remove a point $z_m$ from the training dataset, the parameters become $\hat{\theta}_{-z_m} = \arg\min_\theta \sum_{i \neq m} L(z_i, \theta)$. Up-weighting $z_m$ by $\epsilon$ small enough, then the revised risk $R(\theta)' = \frac{1}{n} \sum_{i=1}^n L(z_i; \theta) + \epsilon L(z_m; \theta)$ is still strongly convex. Then the response function $\hat{\theta}_{\epsilon, -z_m} = R(\theta)'$ is also uniquely defined. The parameter change is denoted as $\Delta_\epsilon = \hat{\theta}_{\epsilon, -z_m} - \hat{\theta}$. Since $\hat{\theta}_{\epsilon, -z_m}$ is the minimizer of $R(\theta)'$, we have the

first-order optimization condition as

$$\nabla_{\hat{\theta}_{\epsilon,-z_m}} R(\theta) + \epsilon \cdot \nabla_{\hat{\theta}_{\epsilon,-z_m}} L(z_m, \hat{\theta}_{\epsilon,-z_m}) = 0$$

Since $\hat{\theta}_{\epsilon,-z_m} \to \hat{\theta}$ as $\epsilon \to 0$, we perform a Taylor expansion of the right-hand side:

$$\left[ \nabla R(\hat{\theta}) + \epsilon \nabla L(z_m, \hat{\theta}) \right] + \left[ \nabla^2 R(\hat{\theta}) + \epsilon \nabla^2 L(z_m, \hat{\theta}) \right] \Delta_\epsilon \approx 0$$

Noting $\epsilon \nabla^2 L(z_m, \hat{\theta}) \Delta_\epsilon$ is $o(\|\Delta_\epsilon\|)$ term, which is smaller than other parts, we drop it in the following analysis. Then the Taylor expansion equation becomes

$$\left[ \nabla R(\hat{\theta}) + \epsilon \nabla L(z_m, \hat{\theta}) \right] + \nabla^2 R(\hat{\theta}) \cdot \Delta_\epsilon \approx 0$$

Solving for $\Delta_\epsilon$, we obtain:

$$\Delta_\epsilon = - \left[ \nabla^2 R(\hat{\theta}) + \epsilon \nabla^2 L(z, \hat{\theta}) \right]^{-1} \left[ \nabla R(\hat{\theta}) + \epsilon \nabla L(z, \hat{\theta}) \right].$$

Remember $\theta$ minimizes $R$, then $\nabla R(\hat{\theta}) = 0$. Dropping $o(\epsilon)$ term, we have

$$\Delta_\epsilon = -\epsilon \nabla^2 R(\hat{\theta})^{-1} \nabla L(z, \hat{\theta}).$$

$$\left. \frac{d\hat{\theta}_{\epsilon,-z_m}}{d\epsilon} \right|_{\epsilon=0} = \left. \frac{d\Delta_\epsilon}{d\epsilon} \right|_{\epsilon=0} = -H_{\hat{\theta}}^{-1} \nabla L(z, \hat{\theta}) \equiv \mathcal{I}_{up,params}(z).$$

Besides, we can obtain the approximation of $\hat{\theta}_{-z_m}$ directly by $\hat{\theta}_{-z_m} \approx \hat{\theta} + \mathcal{I}_{up,params}(z)$.

## C. Acceleration for Influence Function

**EK-FAC.** EK-FAC method relies on two approximations to the Fisher information matrix, equivalent to $G_{\hat{\theta}}$ in our setting, which makes it feasible to compute the inverse of the matrix.

Firstly, assume that the derivatives of the weights in different layers are uncorrelated, which implies that $G_{\hat{\theta}}$ has a block-diagonal structure. Suppose $\hat{g}_\theta$ can be denoted by $\hat{g}_\theta(x) = g_{\theta_L} \circ \cdots \circ g_{\theta_l} \circ \cdots \circ g_{\theta_1}(x)$ where $l \in [L]$. We fold the bias into the weights and vectorize the parameters in the $l$-th layer into a vector $\theta_l \in \mathbb{R}^{d_l}$, $d_l \in \mathbb{N}$ is the number of $l$-th layer parameters. Then $G_{\hat{\theta}}$ can be reaplcaed by $\left( G_1(\hat{\theta}), \cdots, G_L(\hat{\theta}) \right)$, where $G_l(\hat{\theta}) \triangleq n^{-1} \sum_{i=1}^n \nabla_{\hat{\theta}_l} \ell_i \nabla_{\theta_l} \ell_i^{\mathrm{T}}$. Denote $h_l$, $o_l$ as the output and pre-activated output of $l$-th layer. Then $G_l(\theta)$ can be approximated by

$$G_l(\theta) \approx \hat{G}_l(\theta) \triangleq \frac{1}{n} \sum_{i=1}^n h_{l-1}(x_i) h_{l-1}(x_i)^T \otimes \frac{1}{n} \sum_{i=1}^n \nabla_{o_l} \ell_i \nabla_{o_l} \ell_i^T \triangleq \Omega_{l-1} \otimes \Gamma_l.$$

Furthermore, in order to accelerate transpose operation and introduce the damping term, perform eigenvalue decomposition of matrix $\Omega_{l-1}$ and $\Gamma_l$ and obtain the corresponding decomposition results as $Q_\Omega \Lambda_\Omega Q_\Omega^\top$ and $Q_\Gamma \Lambda_\Gamma Q_\Gamma^\top$. Then the inverse of $\hat{H}_l(\theta)$ can be obtained by

$$\hat{H}_l(\theta)^{-1} \approx \left( \hat{G}_l(\hat{g}) + \lambda_l I_{d_l} \right)^{-1} = \left( Q_{\Omega_{l-1}} \otimes Q_{\Gamma_l} \right) \left( \Lambda_{\Omega_{l-1}} \otimes \Lambda_{\Gamma_l} + \lambda_l I_{d_l} \right)^{-1} \left( Q_{\Omega_{l-1}} \otimes Q_{\Gamma_l} \right)^{\mathrm{T}}.$$

Besides, George et al. (2018) proposed a new method that corrects the error in equation 13 which sets the $i$-th diagonal element of $\Lambda_{\Omega_{l-1}} \otimes \Lambda_{\Gamma_l}$ as $\Lambda_{ii}^* = n^{-1} \sum_{j=1}^n \left( \left( Q_{\Omega_{l-1}} \otimes Q_{\Gamma_l} \right) \nabla_{\theta_l} \ell_j \right)_i^2$.

### C.1. EK-FAC for CBMs

In our CBM model, the label predictor is a single linear layer, and Hessian computing costs are affordable. However, the concept predictor is based on Resnet-18, which has many parameters. Therefore, we perform EK-FAC for $\hat{g}$.

$$\hat{g} = \arg\min_g \sum_{j=1}^k L_{C_j} = \arg\min_g \sum_{j=1}^k \sum_{i=1}^n L_C(g^j(x_i), c_i^j),$$

we define $H_{\hat{g}} = \nabla_{\hat{g}}^2 \sum_{i,j} L_{C_j}(g(x_i), c_i)$ as the Hessian matrix of the loss function with respect to the parameters.

To this end, consider the $l$-th layer of $\hat{g}$ which takes as input a layer of activations $\{a_{j,t}\}$ where $j \in \{1, 2, \ldots, J\}$ indexes the input map and $t \in \mathcal{T}$ indexes the spatial location which is typically a 2-D grid. This layer is parameterized by a set of weights $W = (w_{i,j,\delta})$ and biases $b = (b_i)$, where $i \in \{1, \ldots, I\}$ indexes the output map, and $\delta \in \Delta$ indexes the spatial offset (from the center of the filter).

The convolution layer computes a set of pre-activations as

$$[S_l]_{i,t} = s_{i,t} = \sum_{\delta \in \Delta} w_{i,j,\delta} a_{j,t+\delta} + b_i.$$

Denote the loss derivative with respect to $s_{i,t}$ as

$$\mathcal{D}s_{i,t} = \frac{\partial \sum L_{C_j}}{\partial s_{i,t}},$$

which can be computed during backpropagation.

The activations are actually stored as $A_{l-1}$ of dimension $|\mathcal{T}| \times J$. Similarly, the weights are stored as an $I \times |\Delta|J$ array $W_l$. The straightforward implementation of convolution, though highly parallel in theory, suffers from poor memory access patterns. Instead, efficient implementations typically leverage what is known as the expansion operator $[\![ \cdot ]\!]$. For instance, $[\![ A_{l-1} ]\!]$ is a $|\mathcal{T}| \times J|\Delta|$ matrix, defined as

$$[\![ A_{l-1} ]\!]_{t,j|\Delta|+\delta} = [A_{l-1}]_{(t+\delta),j} = a_{j,t+\delta},$$

In order to fold the bias into the weights, we need to add a homogeneous coordinate (i.e. a column of all 1's) to the expanded activations $[\![ A_{l-1} ]\!]$ and denote this as $[\![ A_{l-1} ]\!]_{\mathrm{H}}$. Concatenating the bias vector to the weights matrix, then we have $\theta_l = (b_l, W_l)$.

Then, the approximation for $H_{\hat{g}}$ is given as:

$$G^{(l)}(\hat{g}) = \mathbb{E}\left[ \mathcal{D}w_{i,j,\delta} \mathcal{D}w_{i',j',\delta'} \right] = \mathbb{E}\left[ \left( \sum_{t \in \mathcal{T}} a_{j,t+\delta} \mathcal{D}s_{i,t} \right) \left( \sum_{t' \in \mathcal{T}} a_{j',t'+\delta'} \mathcal{D}s_{i',t'} \right) \right]$$

$$\approx \mathbb{E}\left[ [\![ A_{l-1} ]\!]_{\mathrm{H}}^{\top} [\![ A_{l-1} ]\!]_{\mathrm{H}} \right] \otimes \frac{1}{|\mathcal{T}|} \mathbb{E}\left[ \mathcal{D}S_l^{\top} \mathcal{D}S_l \right] \triangleq \Omega_{l-1} \otimes \Gamma_l.$$

Estimate the expectation using the mean of the training set,

$$G^{(l)}(\hat{g}) \approx \frac{1}{n} \sum_{i=1}^{n} \left( [\![ A_{l-1}^i ]\!]_{\mathrm{H}}^{\top} [\![ A_{l-1}^i ]\!]_{\mathrm{H}} \right) \otimes \frac{1}{n} \sum_{i=1}^{n} \left( \frac{1}{|\mathcal{T}|} \mathcal{D}S_l^{i\top} \mathcal{D}S_l^i \right) \triangleq \hat{\Omega}_{l-1} \otimes \hat{\Gamma}_l.$$

Furthermore, if the factors $\hat{\Omega}_{l-1}$ and $\hat{\Gamma}_l$ have eigen decomposition $Q_{\Omega} \Lambda_{\Omega} Q_{\Omega}^{\top}$ and $Q_{\Gamma} \Lambda_{\Gamma} Q_{\Gamma}^{\top}$, respectively, then the eigen decomposition of $\hat{\Omega}_{l-1} \otimes \hat{\Gamma}_l$ can be written as:

$$\hat{\Omega}_{l-1} \otimes \hat{\Gamma}_l = Q_{\Omega} \Lambda_{\Omega} Q_{\Omega}^{\top} \otimes Q_{\Gamma} \Lambda_{\Gamma} Q_{\Gamma}^{\top}$$

$$= (Q_{\Omega} \otimes Q_{\Gamma}) (\Lambda_{\Omega} \otimes \Lambda_{\Gamma}) (Q_{\Omega} \otimes Q_{\Gamma})^{\top}.$$

Since subsequent inverse operations are required and the current approximation for $G^{(l)}(\hat{g})$ is PSD, we actually use a damped version as

$$\hat{G}^l(\hat{g})^{-1} = (G_l(\hat{g}) + \lambda_l I_{d_l})^{-1} = (Q_{\Omega_{l-1}} \otimes Q_{\Gamma_l}) (\Lambda_{\Omega_{l-1}} \otimes \Lambda_{\Gamma_l} + \lambda_l I_{d_l})^{-1} (Q_{\Omega_{l-1}} \otimes Q_{\Gamma_l})^{\mathrm{T}}. \tag{13}$$

Besides, (George et al., 2018) proposed a new method that corrects the error in equation 13 which sets the $i$-th diagonal element of $\Lambda_{\Omega_{l-1}} \otimes \Lambda_{\Gamma_l}$ as

$$\Lambda_{ii}^* = n^{-1} \sum_{j=1}^{n} \left( (Q_{\Omega_{l-1}} \otimes Q_{\Gamma_l}) \nabla_{\theta_l} \ell_j \right)_i^2.$$

## D. Concept-label-level Influence

### D.1. Proof of Concept-label-level Influence Function

We have a set of erroneous data $D_e$ and its associated index set $S_e \subseteq [n] \times [k]$ such that for each $(w, r) \in S_e$, we have $(x_w, y_w, c_w) \in D_e$ with $c_w^r$ is mislabeled and $\tilde{c}_w^r$ is its corrected concept label. Thus, our goal is to approximate the new CBM without retraining.

**Proof Sketch.** Our goal is to edit $\hat{g}$ and $\hat{f}$ to $\hat{g}_e$ and $\hat{f}_e$. (i) First, we introduce new parameters $\hat{g}_{\epsilon,e}$ that minimize a modified loss function with a small perturbation $\epsilon$. (ii) Then, we perform a Newton step around $\hat{g}$ and obtain an estimate for $\hat{g}_e$. (iii) Then, we consider changing the concept predictor at one data point $(x_{i_c}, y_{i_c}, c_{i_c})$ and retraining the model to obtain a new label predictor $\hat{f}_{i_c}$, obtain an approximation for $\hat{f}_{i_c}$. (iv) Next, we iterate $i_c$ over $1, 2, \cdots, n$, sum all the equations together, and perform a Newton step around $\hat{f}$ to obtain an approximation for $\hat{f}_e$. (v) Finally, we bring the estimate of $\hat{g}$ into the equation for $\hat{f}_e$ to obtain the final approximation.

**Theorem D.1.** *Define the gradient of the $j$-th concept predictor of the $i$-th data $x_i$ as*

$$G_C^j(x_i, \boldsymbol{c}_i; g) \triangleq \nabla_g L_C^j(g(x_i), \boldsymbol{c}_i), \tag{14}$$

*then the retrained concept predictor $\hat{g}_e$ defined by*

$$\hat{g}_e = \arg\min \left[ \sum_{(i,j) \notin S_e} L_C^j(g(x_i), c_i) + \sum_{(i,j) \in S_e} L_C^j(g(x_i), \tilde{c}_i) \right], \tag{15}$$

*can be approximated by $\bar{g}_e$, defined by:*

$$\hat{g} - H_{\hat{g}}^{-1} \cdot \sum_{(w,r) \in S_e} \left( G_C^r(x_w, \tilde{c}_w; \hat{g}) - G_C^r(x_w, \boldsymbol{c}_w; \hat{g}) \right) \tag{16}$$

*where $H_{\hat{g}} = \nabla_{\hat{g}}^2 \sum_{i,j} L_C^j(\hat{g}(x_i), c_i)$ is the Hessian matrix of the loss function with respect to $\hat{g}$.*

*Proof.* For the index $(w, r) \in S_e$, indicating the $r$-th concept of the $w$-th data is wrong, we correct this concept $c_w^r$ to $\tilde{c}_w^r$. Rewrite $\hat{g}_e$ as

$$\hat{g}_e = \arg\min \left[ \sum_{i,j} L_C^j(g(x_i), c_i) + \sum_{(w,r) \in S_e} L_C^r(g(x_w), \tilde{c}_w) - \sum_{(w,r) \in S_e} L_C^r(g(x_w), c_w) \right]. \tag{17}$$

To approximate this effect, define new parameters $\hat{g}_{\epsilon,e}$ as

$$\hat{g}_{\epsilon,e} \triangleq \arg\min \left[ \sum_{i,j} L_C^j(g(x_i), c_i) + \sum_{(w,r) \in S_e} \epsilon \cdot L_C^r(g(x_w), \tilde{c}_w) - \sum_{(w,r) \in S_e} \epsilon \cdot L_C^r(g(x_w), c_w) \right]. \tag{18}$$

Then, because $\hat{g}_{\epsilon,e}$ minimizes equation 18, we have

$$\nabla_{\hat{g}} \sum_{i,j} L_C^j(\hat{g}_{\epsilon,e}(x_i), c_i) + \sum_{(w,r) \in S_e} \epsilon \cdot \nabla_{\hat{g}} L_C^r(\hat{g}_{\epsilon,e}(x_w), \tilde{c}_w) - \sum_{(w,r) \in S_e} \epsilon \cdot \nabla_{\hat{g}} L_C^r(\hat{g}_{\epsilon,e}(x_w), c_w) = 0.$$

Perform a Taylor expansion of the above equation at $\hat{g}$,

$$\nabla_{\hat{g}} \sum_{i,j} L_C^j(\hat{g}(x_i), c_i) + \sum_{(w,r) \in S_e} \epsilon \cdot \nabla_{\hat{g}} L_C^r(\hat{g}(x_w), \tilde{c}_w) - \sum_{(w,r) \in S_e} \epsilon \cdot \nabla_{\hat{g}} L_C^r(\hat{g}(x_w), c_w)$$

$$+ \nabla_{\hat{g}}^2 \sum_{i,j} L_C^j(\hat{g}(x_i), c_i) \cdot (\hat{g}_{\epsilon,e} - \hat{g}) \approx 0. \tag{19}$$

Because of equation 15, the first term of equation 19 equals $0$. Then we have

$$\hat{g}_{\epsilon,e} - \hat{g} = - \sum_{(w,r)\in S_e} \epsilon \cdot H_{\hat{g}}^{-1} \cdot \left( \nabla_{\hat{g}} L_C^r \left( \hat{g}(x_w), \tilde{c}_w \right) - \nabla_{\hat{g}} L_C^r \left( \hat{g}(x_w), c_w \right) \right),$$

where

$$H_{\hat{g}} = \nabla_{\hat{g}}^2 \sum_{i,j} L_C^j \left( \hat{g}(x_i), c_i \right).$$

Then, we do a Newton step around $\hat{g}$ and obtain

$$\hat{g}_e \approx \bar{g}_e \triangleq \hat{g} - H_{\hat{g}}^{-1} \cdot \sum_{(w,r)\in S_e} \left( \nabla_{\hat{g}} L_C^r \left( \hat{g}(x_w), \tilde{c}_w \right) - \nabla_{\hat{g}} L_C^r \left( \hat{g}(x_w), c_w \right) \right). \tag{20}$$

Define the gradient of the $j$-th concept predictor of the $i$-th data $x_i$ as

$$G_C^j(x_i, \boldsymbol{c}_i; g) \triangleq \nabla_g L_C^j \left( g(x_i), \boldsymbol{c}_i \right), \tag{21}$$

then the retrained concept predictor $\hat{g}_e$ defined by

$$\hat{g}_e = \arg\min \left[ \sum_{(i,j)\notin S_e} L_C^j \left( g(x_i), c_i \right) + \sum_{(i,j)\in S_e} L_C^j \left( g(x_i), \tilde{c}_i \right) \right], \tag{22}$$

can be approximated by $\bar{g}_e$, defined by:

$$\hat{g} - H_{\hat{g}}^{-1} \cdot \sum_{(w,r)\in S_e} \left( G_C^r(x_w, \tilde{c}_w; \hat{g}) - G_C^r(x_w, \boldsymbol{c}_w; \hat{g}) \right) \tag{23}$$

$\square$

**Theorem D.2.** *Define the gradient of the label predictor of the $i$-th data $x_i$ as*

$$G_Y(x_i; g, f) \triangleq \nabla_f L_Y \left( f(g(x_i)), y_i \right), \tag{24}$$

*then the retrained label predictor $\hat{f}_e$ defined by*

$$\hat{f}_e = \arg\min \left[ \sum_{i=1}^n L_Y \left( f \left( \hat{g}_e(x_i) \right), y_i \right) \right]$$

*can be approximated by:*

$$\hat{f} + H_{\hat{f}}^{-1} \cdot \sum_{i=1}^n \left( G_Y(x_i; \hat{g}, \hat{f}) - G_Y(x_i; \bar{g}_e, \hat{f}) \right),$$

*where $H_{\hat{f}} = \nabla_{\hat{f}} \sum_{i=1}^n G_Y(x_i; \hat{g}, \hat{f})$ is the Hessian matrix, and $\bar{g}_e$ is given in Theorem D.1.*

*Proof.* Now we come to deduce the edited label predictor towards $\hat{f}_e$.

First, we consider only changing the concept predictor at one data point $(x_{i_c}, y_{i_c}, c_{i_c})$ and retrain the model to obtain a new label predictor $\hat{f}_{i_c}$.

$$\hat{f}_{i_c} = \arg\min \left[ \sum_{i=1, i\neq i_c}^n L_Y \left( f \left( \hat{g}(x_i) \right), y_i \right) + L_Y \left( f \left( \hat{g}_e(x_{i_c}) \right), y_{i_c} \right) \right].$$

We rewrite the above equation as follows:

$$\hat{f}_{i_c} = \arg\min \left[ \sum_{i=1}^n L_Y \left( f \left( \hat{g}(x_i) \right), y_i \right) + L_Y \left( f \left( \hat{g}_e(x_{i_c}) \right), y_{i_c} \right) - L_Y \left( f \left( \hat{g}(x_{i_c}) \right), y_{i_c} \right) \right].$$

We define $\hat{f}_{\epsilon,i_c}$ as:

$$\hat{f}_{\epsilon,i_c} = \arg\min \left[ \sum_{i=1}^{n} L_Y \left( f \left( \hat{g} \left( x_i \right) \right), y_i \right) + \epsilon \cdot L_Y \left( f \left( \hat{g}_e \left( x_{i_c} \right) \right), y_{i_c} \right) - \epsilon \cdot L_Y \left( f \left( \hat{g} \left( x_{i_c} \right) \right), y_{i_c} \right) \right].$$

Derive with respect to $f$ at both sides of the above equation. we have

$$\nabla_{\hat{f}} \sum_{i=1}^{n} L_Y \left( \hat{f}_{\epsilon,i_c} \left( \hat{g} \left( x_i \right) \right), y_i \right) + \epsilon \cdot \nabla_{\hat{f}} L_Y \left( \hat{f}_{\epsilon,i_c} \left( \hat{g}_e \left( x_{i_c} \right) \right), y_{i_c} \right) - \epsilon \cdot \nabla_{\hat{f}} L_Y \left( \hat{f}_{\epsilon,i_c} \left( \hat{g} \left( x_{i_c} \right) \right), y_{i_c} \right) = 0$$

Perform a Taylor expansion of the above equation at $\hat{f}$,

$$\nabla_{\hat{f}} \sum_{i=1}^{n} L_Y \left( \hat{f} \left( \hat{g} \left( x_i \right) \right), y_i \right) + \epsilon \cdot \nabla_{\hat{f}} L_Y \left( \hat{f} \left( \hat{g}_e \left( x_{i_c} \right) \right), y_{i_c} \right)$$

$$- \epsilon \cdot \nabla_{\hat{f}} L_Y \left( \hat{f} \left( \hat{g} \left( x_{i_c} \right) \right), y_{i_c} \right) + \nabla_{\hat{f}}^2 \sum_{i=1}^{n} L_Y \left( \hat{f} \left( \hat{g} \left( x_i \right) \right), y_i \right) \cdot \left( \hat{f}_{\epsilon,i_c} - \hat{f} \right) = 0$$

Then we have

$$\hat{f}_{\epsilon,i_c} - \hat{f} \approx -\epsilon \cdot H_{\hat{f}}^{-1} \cdot \nabla_f \left( L_Y \left( \hat{f} \left( \hat{g}_e \left( x_{i_c} \right) \right), y_{i_c} \right) - L_Y \left( \hat{f} \left( \hat{g} \left( x_{i_c} \right) \right), y_{i_c} \right) \right),$$

where $H_{\hat{f}}^{-1} = \nabla_{\hat{f}}^2 \sum_{i=1}^{n} L_Y \left( \hat{f} \left( \hat{g} \left( x_i \right) \right), y_i \right)$.

Iterate $i_c$ over $1, 2, \cdots, n$, and sum all the equations together, we can obtain:

$$\hat{f}_{\epsilon,e} - \hat{f} \approx -\epsilon \cdot H_{\hat{f}}^{-1} \cdot \sum_{i=1}^{n} \nabla_f \left( L_Y \left( \hat{f} \left( \hat{g}_e \left( x_i \right) \right), y_i \right) - L_Y \left( \hat{f} \left( \hat{g} \left( x_i \right) \right), y_i \right) \right).$$

Perform a Newton step around $\hat{f}$ and we have

$$\hat{f}_e \approx \hat{f} - H_{\hat{f}}^{-1} \cdot \sum_{i=1}^{n} \nabla_f \left( L_Y \left( \hat{f} \left( \hat{g}_e \left( x_i \right) \right), y_i \right) - L_Y \left( \hat{f} \left( \hat{g} \left( x_i \right) \right), y_i \right) \right). \tag{25}$$

Bringing the edited (20) of $g$ into equation (25), we have

$$\hat{f}_e \approx \hat{f} - H_{\hat{f}}^{-1} \cdot \sum_{i=1}^{n} \nabla_f \left( L_Y \left( \hat{f} \left( \bar{g}_e \left( x_i \right) \right), y_i \right) - L_Y \left( \hat{f} \left( \hat{g} \left( x_i \right) \right), y_i \right) \right) \triangleq \bar{f}_e.$$

Define the gradient of the label predictor of the $i$-th data $x_i$ as

$$G_Y(x_i; g, f) \triangleq \nabla_f L_Y \left( f(g(x_i)), y_i \right), \tag{26}$$

then the retrained label predictor $\hat{f}_e$ defined by (4) can be approximated by $\bar{f}_e$, defined by:

$$\hat{f} + H_{\hat{f}}^{-1} \cdot \sum_{i=1}^{n} \left( G_Y(x_i; \hat{g}, \hat{f}) - G_Y(x_i; \bar{g}_e, \hat{f}) \right),$$

where $H_{\hat{f}} = \nabla_{\hat{f}} \sum_{i=1}^{n} G_Y(x_i; \hat{g}, \hat{f})$ is the Hessian matrix, and $\bar{g}_e$ is given in Theorem D.1. $\qquad \square$

## D.2. Theoretical Bound for the Influence Function

Consider the dataset $\mathcal{D} = \{(x_i, c_i, y_i) i = 1^n$, the loss function of the concept predictor $g$ is defined as:

$$L_{\text{Total}}(\mathcal{D}; g) = \sum_{i=1}^{n} L_C(g(x_i), c_i) + \frac{\delta}{2} \cdot \|g\|^2 = \sum_{i=1}^{n} \sum_{j=1}^{k} L_C^j(g(x_i), c_i) + \frac{\delta}{2} \cdot \|g\|^2 = \sum_{i=1}^{n} \sum_{j=1}^{k} g^j(x_i)^\top \log(c_i{}^j) + \frac{\delta}{2} \cdot \|g\|^2.$$

Mathematically, we have a set of erroneous data $D_e$ and its associated index set $S_e \subseteq [n] \times [k]$ such that for each $(w, r) \in S_e$, we have $(x_w, y_w, c_w) \in D_e$ with $c_w^r$ is mislabeled and $\tilde{c}_w^r$ is corrected concept label. Thus, our goal is to estimate the retrained CBM. The retrained concept predictor and label predictor will be represented in the following manner.

$$\hat{g}_e = \arg\min \left[ \sum_{(i,j)\notin S_e} L_C^j(g(x_i), c_i) + \sum_{(i,j)\in S_e} L_C^j(g(x_i), \tilde{c}_i) + \frac{\delta}{2} \cdot \|g\|^2 \right], \tag{27}$$

Define the corrected dataset as $\mathcal{D}^*$. Then the loss function with the influence of erroneous data $D_e$ removed becomes

$$L^-(\mathcal{D}^*; g) = \sum_{(i,j)\notin S_e} L_C^j(g(x_i), c_i) + \sum_{(i,j)\in S_e} L_C^j(g(x_i), \tilde{c}_i) + \frac{\delta}{2} \cdot \|g\|^2. \tag{28}$$

Assume $\hat{g} = \arg\min L_{\text{Total}}(\mathcal{D}; g)$ is the original model parameter, and $\hat{g}_e(\mathcal{D}^*)$ is the minimizer of $L^-(\mathcal{D}^*; g)$, which is obtained from retraining. Denote $\bar{g}_e(\mathcal{D}^*)$ as the updated model with the influence of erroneous data $D_e$ removed and is obtained by the influence function method in theorem D.1, which is an estimation for $\hat{g}_e(\mathcal{D}^*)$.

To simplify the problem, we concentrate on the removal of erroneous data $D_e$ and neglect the process of adding the corrected data back. Once we obtain the bound for $\hat{g}_e(\mathcal{D}^*) - \bar{g}_e(\mathcal{D}^*)$ under this circumstance, the bound for the case where the corrected data is added back can naturally be derived using a similar approach. For brevity, we use the same notations.

Then, the loss function $L^-(\mathcal{D}^*; g)$ becomes

$$L^-(\mathcal{D}^*; g) = \sum_{(i,j)\notin S_e} L_C^j(g(x_i), c_i) + \frac{\delta}{2} \cdot \|g\|^2 = L_{\text{Total}}(\mathcal{D}; g) - \sum_{(i,j)\in S_e} L_C^j(g(x_i), c_i) \tag{29}$$

And the definition of $\bar{g}_e(\mathcal{D}^*)$ becomes

$$\hat{g} + H_{\hat{g}}^{-1} \cdot \sum_{(w,r)\in S_e} G_C^r(x_w, c_w; \hat{g}) \tag{30}$$

where $H_{\hat{g}} = \nabla_{\hat{g}}^2 \sum_{i,j} L_C^j(\hat{g}(x_i), c_i) + \delta \cdot I$ is the Hessian matrix of the loss function with respect to $\hat{g}$. Here $\delta \cdot I$ is a small damping term for ensuring positive definiteness of the Hessian. Introducing the damping term into the Hessian is essentially equivalent to adding a regularization term to the initial loss function. Consequently, $\delta$ can also be interpreted as the regularization strength.

In this part, we will study the error between the estimated influence given by the theorem D.1 method and retraining. We use the parameter changes as the evaluation metric:

$$|(\bar{g}_e - \hat{g}) - (\hat{g}_e - \hat{g})| = |\bar{g}_e - \hat{g}_e| \tag{31}$$

**Assumption D.3.** The loss $L_C(x, c; g)$

$$L_C(x, c; g; j) = L_C^j(g(x), c)$$

is convex and twice-differentiable in $g$, with positive regularization $\delta > 0$. There exists $C_H \in \mathbb{R}$ such that

$$\|\nabla_g^2 L_C(x, c; g_1) - \nabla_g^2 L_C(x, c; g_2)\|_2 \leq C_H \|g_1 - g_2\|_2$$

for all $(x, c) \in \mathcal{D} = \{(x_i, c_i)\}_{i=1}^{n}$, $j \in [k]$ and $g_1, g_2 \in \Gamma$.

Then the function $L'(\mathcal{D}, S_e; g)$:

$$L'(\mathcal{D}, S_e; g) = \sum_{(i,j) \in S_e} L_C^j(g(x_i), c_i) = \sum_{(i,j) \in S_e} L_C(x_i, c_i; g; j)$$

is convex and twice-differentiable in $g$, with some positive regularization. Then we have

$$\|\nabla_g^2 L'(\mathcal{D}, S_e; g_1) - \nabla_g^2 L'(\mathcal{D}, S_e; g_2)\|_2 \leq |S_e| \cdot C_H \|g_1 - g_2\|_2$$

for $g_1, g_2 \in \Gamma$.

**Corollary D.4.**
$$\|\nabla_g^2 L^-(\mathcal{D}^*; g_1) - \nabla_g^2 L^-(\mathcal{D}^*; g_2)\|_2 \leq ((nk + |S_e|) \cdot C_H) \|g_1 - g_2\|$$

*Define* $C_H^- \triangleq (nk + |S_e|) \cdot C_H$

**Definition D.5.** Define $|\mathcal{D}|$ as the number of pairs

$$C_L' = \|\nabla_g L'(\mathcal{D}, S_e; \hat{g})\|_2,$$

$$\sigma_{\min}' = \text{smallest singular value of } \nabla_g^2 L^-(\mathcal{D}^*; \hat{g}),$$

$$\sigma_{\min} = \text{smallest singular value of } \nabla_g^2 L_{\text{Total}}(\mathcal{D}; \hat{g}),$$

Based on above corollaries and assumptions, we derive the following theorem.

**Theorem D.6.** *We obtain the error between the actual influence and our predicted influence as follows:*

$$\|\hat{g}_e(\mathcal{D}^*) - \bar{g}_e(\mathcal{D}^*)\|$$
$$\leq \frac{C_H^- C_L'^2}{2(\sigma_{min}' + \delta)^3} + \left| \frac{2\delta + \sigma_{min} + \sigma_{min}'}{(\delta + \sigma_{min}') \cdot (\delta + \sigma_{min})} \right| \cdot C_L'.$$

*Proof.* We will use the one-step Newton approximation as an intermediate step. Define $\Delta g_{Nt}(\mathcal{D}^*)$ as

$$\Delta g_{Nt}(\mathcal{D}^*) \triangleq H_\delta^{-1} \cdot \nabla_g L'(\mathcal{D}, S_e; \hat{g}),$$

where $H_\delta = \delta \cdot I + \nabla_g^2 L^-(\mathcal{D}^*; \hat{g})$ is the regularized empirical Hessian at $\hat{g}$ but reweighed after removing the influence of wrong data. Then the one-step Newton approximation for $\hat{g}(\mathcal{D}^*)$ is defined as $g_{Nt}(\mathcal{D}^*) \triangleq \Delta g_{Nt}(\mathcal{D}^*) + \hat{g}$.

In the following, we will separate the error between $\bar{g}_e(\mathcal{D}^*)$ and $\hat{g}_e(\mathcal{D}^*)$ into the following two parts:

$$\hat{g}_e(\mathcal{D}^*) - \bar{g}_e(\mathcal{D}^*) = \underbrace{\hat{g}_e(\mathcal{D}^*) - g_{Nt}(\mathcal{D}^*)}_{\text{Err}_{\text{Nt, act}}(\mathcal{D}^*)} + \underbrace{(g_{Nt}(\mathcal{D}^*) - \hat{g}) - (\bar{g}_e(\mathcal{D}^*) - \hat{g})}_{\text{Err}_{\text{Nt, if}}(\mathcal{D}^*)}$$

Firstly, in **Step** 1, we will derive the bound for Newton-actual error $\text{Err}_{\text{Nt, act}}(\mathcal{D}^*)$. Since $L^-(g)$ is strongly convex with parameter $\sigma_{\min}' + \delta$ and minimized by $\hat{g}_e(\mathcal{D}^*)$, we can bound the distance $\|\hat{g}_e(\mathcal{D}^*) - g_{Nt}(\mathcal{D}^*)\|_2$ in terms of the norm of the gradient at $g_{Nt}$:

$$\|\hat{g}_e(\mathcal{D}^*) - g_{Nt}(\mathcal{D}^*)\|_2 \leq \frac{2}{\sigma_{\min}' + \delta} \|\nabla_g L^-(g_{Nt}(\mathcal{D}^*))\|_2 \tag{32}$$

Therefore, the problem reduces to bounding $\|\nabla_g L^-(g_{Nt}(\mathcal{D}^*))\|_2$. Noting that $\nabla_g L'(\hat{g}) = -\nabla_g L^-$. This is because $\hat{g}$ minimizes $L^- + L'$, that is,

$$\nabla_g L^-(\hat{g}) + \nabla_g L'(\hat{g}) = 0.$$

Recall that $\Delta g_{Nt} = H_\delta^{-1} \cdot \nabla_g L'(\mathcal{D}, S_e; \hat{g}) = -H_\delta^{-1} \cdot \nabla_g L^-(\mathcal{D}^*; \hat{g})$. Given the above conditions, we can have this bound for $\text{Err}_{\text{Nt, act}}(-\mathcal{D}^*)$.

$$
\begin{aligned}
&\left\| \nabla_g L^- (g_{Nt}(\mathcal{D}^*)) \right\|_2 \\
=& \left\| \nabla_g L^- (\hat{g} + \Delta g_{Nt}(\mathcal{D}^*)) \right\|_2 \\
=& \left\| \nabla_g L^- (\hat{g} + \Delta g_{N_t}(\mathcal{D}^*)) - \nabla_g L^- (\hat{g}) - \nabla_g^2 L^- (\hat{g}) \cdot \Delta g_{N_t}(\mathcal{D}^*) \right\|_2 \\
=& \left\| \int_0^1 \left( \nabla_g^2 L^- (\hat{g} + t \cdot \Delta g_{Nt}(\mathcal{D}^*)) - \nabla_g^2 L^- (\hat{g}) \right) \Delta g_{Nt}(\mathcal{D}^*) \, dt \right\|_2 \\
\leq& \frac{C_H^-}{2} \left\| \Delta g_{Nt}(\mathcal{D}^*) \right\|_2^2 = \frac{C_H^-}{2} \left\| \left[ \nabla_g^2 L^- (\hat{g}) \right]^{-1} \nabla_g L^- (\hat{g}) \right\|_2^2 \\
\leq& \frac{C_H^-}{2(\sigma'_{\min} + \delta)^2} \left\| \nabla_g L^- (\hat{g}) \right\|_2^2 = \frac{C_H^-}{2(\sigma'_{\min} + \delta)^2} \left\| \nabla_g L'(\hat{g}) \right\|_2^2 \\
\leq& \frac{C_H^- {C_L'}^2}{2(\sigma'_{\min} + \delta)^2}.
\end{aligned}
\tag{33}
$$

Now we come to **Step** 2 to bound $\text{Err}_{\text{Nt, if}}(-\mathcal{D}^*)$, and we will bound the difference in parameter change between Newton and our ECBM method.

$$
\begin{aligned}
&\left\| (g_{Nt}(\mathcal{D}^*) - \hat{g}) - (\bar{g}_e(\mathcal{D}^*) - \hat{g}) \right\| \\
=& \left\| \left[ \left( \delta \cdot I + \nabla_g^2 L^- (\hat{g}) \right)^{-1} + \left( \delta \cdot I + \nabla_g^2 L_{\text{Total}} (\hat{g}) \right)^{-1} \right] \cdot \nabla_g L'(\mathcal{D}, S_e; \hat{g}) \right\|
\end{aligned}
$$

For simplification, we use matrix $A$, $B$ for the following substitutions:

$$
A = \delta \cdot I + \nabla_g^2 L^- (\hat{g})
$$
$$
B = \delta \cdot I + \nabla_g^2 L_{\text{Total}} (\hat{g})
$$

And $A$ and $B$ are positive definite matrices with the following properties

$$
\delta + \sigma'_{\min} \prec A \prec \delta + \sigma'_{\max}
$$
$$
\delta + \sigma_{\min} \prec B \prec \delta + \sigma_{\max}
$$

Therefore, we have

$$
\begin{aligned}
&\left\| (g_{Nt}(\mathcal{D}^*) - \hat{g}) - (\bar{g}_e(\mathcal{D}^*) - \hat{g}) \right\| \\
=& \left\| \left( A^{-1} + B^{-1} \right) \cdot \nabla_g L^- (\mathcal{D}^*; \hat{g}) \right\| \\
\leq& \left\| A^{-1} + B^{-1} \right\| \cdot \left\| \nabla_g L^- (\mathcal{D}^*; \hat{g}) \right\| \\
\leq& \left| \frac{2\delta + \sigma_{\min} + \sigma'_{\min}}{(\delta + \sigma'_{\min}) \cdot (\delta + \sigma_{\min})} \right| \cdot \left\| \nabla_g L^- (\mathcal{D}^*; \hat{g}) \right\| \\
\leq& \left| \frac{2\delta + \sigma_{\min} + \sigma'_{\min}}{(\delta + \sigma'_{\min}) \cdot (\delta + \sigma_{\min})} \right| \cdot C_L'
\end{aligned}
\tag{34}
$$

By combining the conclusions from Step I and Step II in Equations 32, 33 and 34, we obtain the error between the actual influence and our predicted influence as follows:

$$
\begin{aligned}
&\left\| \hat{g}_e(\mathcal{D}^*) - \bar{g}_e(\mathcal{D}^*) \right\| \\
\leq& \frac{C_H^- {C_L'}^2}{2(\sigma'_{\min} + \delta)^3} + \left| \frac{2\delta + \sigma_{\min} + \sigma'_{\min}}{(\delta + \sigma'_{\min}) \cdot (\delta + \sigma_{\min})} \right| \cdot C_L'.
\end{aligned}
$$

$\square$

*Remark* D.7. Theorem D.6 reveals one key finding about influence function estimation: The estimation error scales inversely with the regularization parameter $\delta$ ($\mathcal{O}(1/\delta)$), indicating that increased regularization improves approximation accuracy.

*Remark* D.8. In CBM, retraining is the most accurate way to handle the removal of a training data point. For the concept predictor, we derive a theoretical error bound for an influence function-based approximation. However, the label predictor differs. As a single-layer linear model, the label predictor is computationally inexpensive to retrain. However, its input depends on the concept predictor, making theoretical analysis challenging due to: (1) Input dependency: Changes in the concept predictor affect the label predictor's input, coupling their updates. (2) Error propagation: Errors from the concept predictor propagate to the label predictor, introducing complex interactions. Given the label predictor's low retraining cost, direct retraining is more practical and accurate. Thus, we focus our theoretical analysis on the concept predictor.

# E. Concept-level Influence

### E.1. Proof of Concept-level Influence Function

We address situations that delete $p_r$ for $r \in M$ concept removed dataset. Our goal is to estimate $\hat{g}_{-p_M}, \hat{f}_{-p_M}$, which is the concept and label predictor trained on the $p_r$ for $r \in M$ concept removed dataset.

**Proof Sketch.** The main ideas are as follows: (i) First, we define a new predictor $\hat{g}^*_{p_M}$, which has the same dimension as $\hat{g}$ and the same output as $\hat{g}_{-p_M}$. Then deduce an approximation for $\hat{g}^*_{p_M}$. (ii) Then, we consider setting $p_r = 0$ instead of removing it, we get $\hat{f}_{p_M=0}$, which is equivalent to $\hat{f}_{-p_M}$ according to lemma E.1. We estimate this new predictor as a substitute. (iii) Next, we assume we only use the updated concept predictor $\hat{g}^*_{p_M}$ for one data $(x_{i_r}, y_{i_r}, c_{i_r})$ and obtain a new label predictor $\hat{f}_{ir}$, and obtain a one-step Newtonian iterative approximation of $\hat{f}_{ir}$ with respect to $\hat{f}$. (iv) Finally, we repeat the above process for all data points and combine the estimate of $\hat{g}$ in Theorem E.2, we obtain a closed-form solution of the influence function for $\hat{f}$.

First, we introduce our following lemma:

**Lemma E.1.** *For the concept bottleneck model, if the label predictor utilizes linear transformations of the form $\hat{f} \cdot c$ with input $c$, then, for each $r \in M$, we remove the $r$-th concept from $c$ and denote the new input as $c'$. Set the $r$-th concept to 0 and denote the new input as $c^0$. Then we have $\hat{f}_{-p_M} \cdot c' = \hat{f}_{p_M=0} \cdot c^0$ for any $c$.*

*Proof.* Assume the parameter space of $\hat{f}_{-p_M}$ and $\hat{f}_{p_M=0}$ are $\Gamma$ and $\Gamma_0$, respectively, then there exists a surjection $P : \Gamma \to \Gamma_0$. For any $\theta \in \Gamma$, $P(\theta)$ is the operation that removes the $r$-th row of $\theta$ for $r \in M$. Then we have:

$$P(\theta) \cdot c' = \sum_{t \notin M} \theta[j] \cdot c'[j] = \sum_t \theta[t] \mathbb{I}\{t \notin M\} c[t] = \theta \cdot c^0.$$

Thus, the loss function $L_Y(\theta, c^0) = L_Y(P(\theta), c')$ of both models is the same for every sample in the second stage. Besides, by formula derivation, we have, for $\theta' \in \Gamma_0$, for any $\theta$ in $P^{-1}(\theta')$,

$$\frac{\partial L_Y(\theta, c^0)}{\partial \theta} = \frac{\partial L_Y(P(\theta), c')}{\partial \theta'}$$

Thus, if the same initialization is performed, $\hat{f}_{-p_M} \cdot c' = \hat{f}_{p_M=0} \cdot c^0$ for any $c$ in the dataset. $\qquad\square$

**Theorem E.2.** *For the retrained concept predictor $\hat{g}_{-p_M}$ defined by*

$$\hat{g}_{-p_M} = \arg\min_{g'} \sum_{j \notin M} \sum_{i=1}^n L_C^j(g'(x_i), c_i),$$

*we map it to $\hat{g}^*_{-p_M}$ as*

$$\hat{g}^*_{-p_M} = \arg\min_{g' \in T_0} \sum_{j \notin M} \sum_{i=1}^n L_C^j(g'(x_i), c_i).$$

*And we can edit the initial $\hat{g}$ to $\hat{g}^*_{-p_M}$, defined as:*

$$\bar{g}_{-p_M} \triangleq \hat{g} - H_{\hat{g}}^{-1} \cdot \sum_{j \notin M} \sum_{i=1}^n D_C^j(x_i, c_i; \hat{g}), \tag{35}$$

*where $H_{\hat{g}} = \nabla_g \sum_{j \notin M} \sum_{i=1}^{n} D_C^j(x_i, c_i; \hat{g})$. Then, by removing all zero rows inserted during the mapping phase, we can naturally approximate $\hat{g}_{-p_M} \approx \mathrm{P}^{-1}(\hat{g}_{-p_M}^*)$.*

*Proof.* At this level, we consider the scenario that removes a set of mislabeled concepts or introduces new ones. Because after removing concepts from all the data, the new concept predictor has a different dimension from the original. We denote $g^j(x_i)$ as the $j$-th concept predictor with $x_i$, and $c_i^j$ as the $j$-th concept in data $z_i$. For simplicity, we treat $g$ as a collection of $k$ concept predictors and separate different columns as a vector $g^j(x_i)$. Actually, the neural network gets $g$ as a whole.

For the comparative purpose, we introduce a new notation $\hat{g}_{-p_M}^*$. Specifically, we define weights of $\hat{g}$ and $\hat{g}_{-p_M}^*$ for the last layer of the neural network as follows.

$$\hat{g}_{-p_M}(x) = \underbrace{\begin{pmatrix} w_{11} & w_{12} & \cdots & w_{1d_i} \\ w_{21} & w_{22} & \cdots & w_{2d_i} \\ \vdots & \vdots & & \vdots \\ w_{(k-1)1} & w_{(k-1)2} & \cdots & w_{(k-1)d_i} \end{pmatrix}}_{(k-1)\times d_i} \cdot \underbrace{\begin{pmatrix} x^1 \\ x^2 \\ \vdots \\ x^{d_i} \end{pmatrix}}_{d_i \times 1} = \underbrace{\begin{pmatrix} c_1 \\ \vdots \\ c_{r-1} \\ c_{r+1} \\ \vdots \\ c_k \end{pmatrix}}_{(k-1)\times 1}$$

$$\hat{g}_{-p_M}^*(x) = \underbrace{\begin{pmatrix} w_{11} & w_{12} & \cdots & w_{1d_i} \\ \vdots & \vdots & & \vdots \\ w_{(r-1)1} & w_{(r-1)2} & \cdots & w_{(r-1)d_i} \\ 0 & 0 & \cdots & 0 \\ w_{(r+1)1} & w_{(r+1)2} & \cdots & w_{(r+1)d_i} \\ \vdots & \vdots & & \vdots \\ w_{k1} & w_{k2} & \cdots & w_{kd_i} \end{pmatrix}}_{k \times d_i} \cdot \underbrace{\begin{pmatrix} x^1 \\ \vdots \\ x^{r-1} \\ x^r \\ x^{r+1} \\ \vdots \\ x^{d_i} \end{pmatrix}}_{d_i \times 1} = \underbrace{\begin{pmatrix} c_1 \\ \vdots \\ c_{r-1} \\ 0 \\ c_{r+1} \\ \vdots \\ c_k \end{pmatrix}}_{k \times 1},$$

where $r$ is an index from the index set $M$.

Firstly, we want to edit to $\hat{g}_{-p_M}^* \in T_0 = \{w_{\text{final}} = 0\} \subseteq T$ based on $\hat{g}$, where $w_{\text{final}}$ is the parameter of the final layer of neural network. Let us take a look at the definition of $\hat{g}_{-p_M}^*$:

$$\hat{g}_{-p_M}^* = \arg\min_{g' \in T_0} \sum_{j \notin M} \sum_{i=1}^{n} L_C^j(g'(x_i), c_i).$$

Then, we separate the $r$-th concept-related item from the rest and rewrite $\hat{g}$ as the following form:

$$\hat{g} = \arg\min_{g \in T} \left[ \sum_{j \notin M} \sum_{i=1}^{n} L_C^j(g(x_i), c_i) + \sum_{r \in M} \sum_{i=1}^{n} L_C^r(g(x_i), c_i) \right].$$

Then, if the $r$-th concept part is up-weighted by some small $\epsilon$, this gives us the new parameters $\hat{g}_{\epsilon,p_M}$, which we will abbreviate as $\hat{g}_\epsilon$ below.

$$\hat{g}_{\epsilon,p_M} \triangleq \arg\min_{g \in T} \left[ \sum_{j \notin M} \sum_{i=1}^{n} L_C^j(g(x_i), c_i) + \epsilon \cdot \sum_{r \in M} \sum_{i=1}^{n} L_C^r(g(x_i), c_i) \right].$$

Obviously, when $\epsilon \to 0$, $\hat{g}_\epsilon \to \hat{g}_{-p_M}^*$. We can obtain the minimization conditions from the definitions above.

$$\nabla_{\hat{g}_{-p_M}^*} \sum_{j \notin M} \sum_{i=1}^{n} L_C^j(\hat{g}_{-p_M}^*(x_i), c_i) = 0. \tag{36}$$

$$\nabla_{\hat{g}_\epsilon} \sum_{j\notin M} \sum_{i=1}^{n} L_C^j(\hat{g}_\epsilon(x_i), c_i) + \epsilon \cdot \nabla_{\hat{g}_\epsilon} \sum_{r\in M} \sum_{i=1}^{n} L_C^r(\hat{g}_\epsilon(x_i), c_i) = 0.$$

Perform a first-order Taylor expansion of equation 36 with respect to $\hat{g}_\epsilon$, then we get

$$\nabla_g \sum_{j\notin M} \sum_{i=1}^{n} L_C^j(\hat{g}_\epsilon(x_i), c_i) + \nabla_g^2 \sum_{j\notin M} \sum_{i=1}^{n} L_C^j(\hat{g}_\epsilon(x_i), c_i) \cdot (\hat{g}^*_{-p_M} - \hat{g}_\epsilon) \approx 0.$$

Then we have

$$\hat{g}^*_{-p_M} - \hat{g}_\epsilon = -H_{\hat{g}_\epsilon}^{-1} \cdot \nabla_g \sum_{j\notin M} \sum_{i=1}^{n} L_C^j(\hat{g}_\epsilon(x_i), c_i).$$

Where $H_{\hat{g}_\epsilon} = \nabla_g^2 \sum_{j\notin M} \sum_{i=1}^{n} L_C^j(\hat{g}_\epsilon(x_i), c_i)$.

We can see that:

When $\epsilon = 0$,

$$\hat{g}_\epsilon = \hat{g}^*_{-p_M},$$

When $\epsilon = 1$, $\hat{g}_\epsilon = \hat{g}$,

$$\hat{g}^*_{-p_M} - \hat{g} \approx -H_{\hat{g}}^{-1} \cdot \nabla_g \sum_{j\notin M} \sum_{i=1}^{n} L_C^j(\hat{g}(x_i), c_i),$$

where $H_{\hat{g}} = \nabla_g^2 \sum_{j\notin M} \sum_{i=1}^{n} L_C^j(\hat{g}(x_i), c_i)$.

Then, an approximation of $\hat{g}^*_{-p_M}$ is obtained.

$$\hat{g}^*_{-p_M} \approx \hat{g} - H_{\hat{g}}^{-1} \cdot \nabla_g \sum_{j\notin M} \sum_{i=1}^{n} L_C^j(\hat{g}(x_i), c_i). \tag{37}$$

Recalling the definition of the gradient:

$$G_C^j(x_i, c_i; \hat{g}) = L_C^j(\hat{g}(x_i), c_i)) = \hat{g}^j(x_i)^\top \cdot \log(c_i^j).$$

Then the approximation of $\hat{g}^*_{-p_M}$ becomes

$$\bar{g}_{-p_M} \triangleq \hat{g} - H_{\hat{g}}^{-1} \cdot \sum_{j\notin M} \sum_{i=1}^{n} G_C^j(x_i, c_i; \hat{g}),$$

$\square$

**Theorem E.3.** *For the retrained label predictor $\hat{f}_{-p_M}$ defined as*

$$\hat{f}_{-p_M} = \arg\min_{f'} \sum_{i=1}^{n} L_Y = \arg\min_{f'} \sum_{i=1}^{n} L_Y(f'(\hat{g}_{-p_M}(x_i)), y_i),$$

*We can consider its equivalent version $\hat{f}_{p_M=0}$ as:*

$$\hat{f}_{p_M=0} = \arg\min_{f} \sum_{i=1}^{n} L_Y\left(f\left(\hat{g}^*_{-p_M}(x_i)\right), y_i\right),$$

*which can be edited by*

$$\hat{f}_{p_M=0} \approx \bar{f}_{p_M=0} \triangleq \hat{f} - H_{\hat{f}}^{-1} \cdot \sum_{l=1}^{n} G_Y(x_l; \bar{g}^*_{-p_M}, \hat{f}),$$

*where $H_{\hat{f}} = \nabla_{\hat{f}} \sum_{i=1}^{n} G_Y(x_l; \bar{g}^*_{-p_M}, \hat{f})$ is the Hessian matrix. Deleting the $r$-th dimension of $\bar{f}_{p_M=0}$ for $r \in M$, then we can map it to $\bar{f}_{-p_M}$, which is the approximation of the final edited label predictor $\hat{f}_{-p_M}$ under concept level.*

*Proof.* Now, we come to the approximation of $\hat{f}_{-p_M}$. Noticing that the input dimension of $f$ decreases to $k - |M|$. We consider setting $p_r = 0$ for all data points in the training phase of the label predictor and get another optimal model $\hat{f}_{p_M=0}$. From lemma E.1, we know that for the same input $x$, $\hat{f}_{p_M=0}(x) = \hat{f}_{-p_M}$. And the values of the corresponding parameters in $\hat{f}_{p_M=0}$ and $\hat{f}_{-p_M}$ are equal.

Now, let us consider how to edit the initial $\hat{f}$ to $\hat{f}_{p_M=0}$. Firstly, assume we only use the updated concept predictor $\hat{g}^*_{-p_M}$ for one data $(x_{i_r}, y_{i_r}, c_{i_r})$ and obtain the following $\hat{f}_{ir}$, which is denoted as

$$\hat{f}_{ir} = \arg\min_f \left[ \sum_{i=1}^n L_Y(f(\hat{g}(x_i)), y_i) + L_Y(f(\hat{g}^*_{-p_M}(x_{ir})), y_{ir}) - L_Y(f(\hat{g}(x_{ir})), y_{ir}) \right].$$

Then up-weight the $i_r$-th data by some small $\epsilon$ and have the following new parameters:

$$\hat{f}_{\epsilon,ir} = \arg\min_f \left[ \sum_{i=1}^n L_Y(f(\hat{g}(x_i)), y_i) + \epsilon \cdot L_Y(f(\hat{g}^*_{-p_M}(x_{ir})), y_{ir}) - \epsilon \cdot L_Y(f(\hat{g}(x_{ir})), y_{ir}) \right].$$

Deduce the minimized condition subsequently,

$$\nabla_f \sum_{i=1}^n L_Y(\hat{f}_{ir}(\hat{g}(x_i)), y_i) + \epsilon \cdot \nabla_f L_Y(\hat{f}_{ir}(\hat{g}^*_{-p_M}(x_{ir})), y_{ir}) - \epsilon \cdot \nabla_f L_Y(\hat{f}_{ir}(\hat{g}(x_{ir})), y_{ir}) = 0.$$

If we expand first term of $\hat{f}$, which $\hat{f}_{ir,\epsilon} \to \hat{f}(\epsilon \to 0)$, then

$$\nabla_f \sum_{i=1}^n L_Y \left( \hat{f}(\hat{g}(x_i)), y_i \right) + \epsilon \cdot \nabla_f L_Y(\hat{f}(\hat{g}^*_{-p_M}(x_{ir})), y_{ir}) - \epsilon \cdot \nabla_f L_Y(\hat{f}(\hat{g}(x_{ir})), y_{ir})$$

$$+ \left( \nabla_f^2 \sum_{i=1}^n L_Y \left( \hat{f}(\hat{g}(x_i)), y_i \right) \right) \cdot (\hat{f}_{ir,\epsilon} - \hat{f}) = 0.$$

Note that $\nabla_f \sum_{i=1}^n L_Y(\hat{f}(\hat{g}(x_i)), y_i) = 0$. Thus we have

$$\hat{f}_{ir,\epsilon} - \hat{f} = H_{\hat{f}}^{-1} \cdot \epsilon \left( \nabla_f L_Y(\hat{f}(\hat{g}^*_{-p_M}(x_{ir})), y_{ir}) - \nabla_f L_Y(\hat{f}(\hat{g}(x_{ir})), y_{ir}) \right).$$

We conclude that

$$\left. \frac{\mathrm{d}\hat{f}_{\epsilon,ir}}{\mathrm{d}\epsilon} \right|_{\epsilon=0} = H_{\hat{f}}^{-1} \cdot \left( \nabla_{\hat{f}} L_Y(\hat{f}(\hat{g}^*_{-p_M}(x_{ir})), y_{ir}) - \nabla_{\hat{f}} L_Y(\hat{f}(\hat{g}(x_{ir})), y_{ir}) \right).$$

Perform a one-step Newtonian iteration at $\hat{f}$ and we get the approximation of $\hat{f}_{i_r}$.

$$\hat{f}_{ir} \approx \hat{f} + H_{\hat{f}}^{-1} \cdot \left( \nabla_{\hat{f}} L_Y(\hat{f}(\hat{g}(x_{ir})), y_{ir}) - \nabla_{\hat{f}} L_Y(\hat{f}(\hat{g}^*_{-p_M}(x_{ir})), y_{ir}) \right).$$

Reconsider the definition of $\hat{f}_{i_r}$, we use the updated concept predictor $\hat{g}^*_{-p_M}$ for one data $(x_{i_r}, y_{i_r}, c_{i_r})$. Now we carry out this operation for all the other data and estimate $\hat{f}_{p_M=0}$. Combining the minimization condition from the definition of $\hat{f}$, we have

$$\hat{f}_{p_M=0} \approx \hat{f} + H_{\hat{f}}^{-1} \cdot \left( \nabla_{\hat{f}} \sum_{i=1}^n L_Y(\hat{f}(\hat{g}(x_i)), y_i) - \nabla_{\hat{f}} \sum_{i=1}^n L_Y(\hat{f}(\hat{g}^*_{-p_M}(x_i)), y_i) \right)$$

$$= \hat{f} + H_{\hat{f}}^{-1} \cdot \left( -\nabla_{\hat{f}} \sum_{i=1}^n L_Y(\hat{f}(\hat{g}^*_{-p_M}(x_i)), y_i) \right)$$

$$= \hat{f} - H_{\hat{f}}^{-1} \sum_{l=1}^n \nabla_{\hat{f}} L_Y(\hat{f}(\hat{g}^*_{-p_M}(x_l)), y_l). \tag{38}$$

Theorem E.2 gives us the edited version of $\hat{g}^*_{-p_M}$. Substitute it into equation 38, and we get the final closed-form edited label predictor under concept level:

$$\hat{f}_{p_M=0} \approx \bar{f}_{p_M=0} \triangleq \hat{f} - H_{\hat{f}}^{-1} \cdot \nabla_{\hat{f}} \sum_{l=1}^{n} L_{Y_l}\left(\hat{f}, \bar{g}^*_{-p_M}\right),$$

where $H_{\hat{f}} = \nabla_{\hat{f}}^2 \sum_{i=1}^{n} L_{Y_i}(\hat{f}, \hat{g})$ is the Hessian matrix of the loss function respect to is the Hessian matrix of the loss function respect to $\hat{f}$. Recalling the definition of the gradient:

$$G_Y(x_l; \bar{g}^*_{-p_M}, \hat{f}) = \nabla_{\hat{f}} L_Y\left(\hat{f}\left(\bar{g}^*_{-p_M}(x_l)\right), y_l\right),$$

then the approximation becomes

$$\hat{f}_{p_M=0} \approx \bar{f}_{p_M=0} \triangleq \hat{f} - H_{\hat{f}}^{-1} \cdot \sum_{l=1}^{n} G_Y(x_l; \bar{g}^*_{-p_M}, \hat{f}).$$

$\square$

## E.2. Theoretical Bound for the Influence Function

Consider the dataset $\mathcal{D} = \{(x_i, c_i, y_i\}_{i=1}^{n}$, the loss function of the concept predictor $g$ is defined as:

$$L_{\text{Total}}(\mathcal{D}; g) = \sum_{i=1}^{n} L_C(g(x_i), c_i) + \frac{\delta}{2} \cdot \|g\|^2 = \sum_{i=1}^{n} \sum_{j=1}^{k} L_C^j(g(x_i), c_i) + \frac{\delta}{2} \cdot \|g\|^2 = \sum_{i=1}^{n} \sum_{j=1}^{k} g^j(x_i)^\top \log(c_i^j) + \frac{\delta}{2} \cdot \|g\|^2.$$

Mathematically, we have a set of erroneous concepts need to be removed, which are denoted as $p_r$ for $r \in M$. Then the retrained concept predictor becomes

$$\hat{g}_{-p_M} = \arg\min_{g'} \sum_{j \notin M} \sum_{i=1}^{n} L_C^j(g'(x_i), c_i) + \frac{\delta}{2} \cdot \|g\|^2.$$

We map it to $\hat{g}^*_{-p_M}$ as $\hat{g}_{-p_M}$ to $\hat{g}^*_{-p_M} \triangleq \mathrm{P}(\hat{g}_{-p_M})$, which has the same amount of parameters as $\hat{g}$ and has the same predicted concepts $\hat{g}^*_{-p_M}(j)$ as $\hat{g}_{-p_M}(j)$ for all $j \in [d_i] - M$. We achieve this effect by inserting a zero row vector into the $r$-th row of the matrix in the final layer of $\hat{g}_{-p_M}$ for $r \in M$. Thus, we can see that the mapping $P$ is one-to-one. Moreover, assume the parameter space of $\hat{g}$ is $T$ and that of $\hat{g}^*_{-p_M}$, $T_0$ is the subset of $T$. Noting that $\hat{g}^*_{-p_M}$ is the optimal model of the following objective function:

$$\hat{g}^*_{-p_M} = \arg\min_{g' \in T_0} \sum_{j \notin M} \sum_{i=1}^{n} L_C^j(g'(x_i), c_i) + \frac{\delta}{2} \cdot \|g\|^2.$$

Then the loss function with the influence of erroneous concepts removed becomes

$$L^-(\mathcal{D}; g) = \sum_{j \notin M} \sum_{i=1}^{n} L_C^j(g'(x_i), c_i) + \frac{\delta}{2} \cdot \|g\|^2 = L_{\text{Total}}(\mathcal{D}; g) - \sum_{j \in M} \sum_{i=1}^{n} L_C^j(g(x_i), c_i). \tag{39}$$

Assume $\hat{g} = \arg\min L_{\text{Total}}(\mathcal{D}; g)$ is the original model parameter. $\hat{g}_{-p_M}(\mathcal{D})$ and $\hat{g}^*_{-p_M}(\mathcal{D})$ is the minimizer of $L^-(\mathcal{D}; g)$, which is obtained from retraining in different parameter space. $\hat{g}^*_{-p_M}(\mathcal{D})$ shares the same dimensionality as the original model. Because $\hat{g}_{-p_M}(\mathcal{D})$ and $\hat{g}^*_{-p_M}(\mathcal{D})$ produces identical outputs given identical inputs, to simplify the proof, we use $\hat{g}^*_{-p_M}(\mathcal{D})$ as the retrained model.

Denote $\bar{g}_{-p_M}$ as the updated model with the influence of erroneous concepts removed and is obtained by the influence function method in theorem E.2, which is an estimation for $\hat{g}^*_{-p_M}(\mathcal{D})$.

$$\bar{g}_{-p_M}(\mathcal{D}) \triangleq \hat{g} - H_{\hat{g}}^{-1} \cdot \sum_{j \notin M} \sum_{i=1}^{n} G_C^j(x_i, c_i; \hat{g}),$$

In this part, we will study the error between the estimated influence given by the theorem E.2 method and $\hat{g}^*_{-p_M}(\mathcal{D})$. We use the parameter changes as the evaluation metric:

$$\left|(\bar{g}_{-p_M} - \hat{g}) - (\hat{g}^*_{-p_M} - \hat{g})\right| = \left|\bar{g}_{-p_M} - \hat{g}^*_{-p_M}\right| \tag{40}$$

**Assumption E.4.** The loss $L_C(x, c; g; j)$

$$L_C(\mathcal{D}; g; j) = \sum_{i=1}^{n} L_C^j(g(x_i), c_i).$$

is convex and twice-differentiable in $g$, with positive regularization $\delta > 0$. There exists $C_H \in \mathbb{R}$ such that

$$\|\nabla_g^2 L_C(\mathcal{D}; g_1; j) - \nabla_g^2 L_C(\mathcal{D}; g_2; j)\|_2 \leq C_H \|g_1 - g_2\|_2$$

for all $j \in [k]$ and $g_1, g_2 \in \Gamma$.

**Definition E.5.**

$$C'_L = \max_j \|\nabla_g L_C(\mathcal{D}; \hat{g}; j)\|_2,$$

$$\sigma'_{min} = \text{smallest singular value of } \nabla_g^2 L^-(\mathcal{D}; \hat{g}),$$

$$\sigma_{min} = \text{smallest singular value of } \nabla_g^2 L_{\text{Total}}(\mathcal{D}; \hat{g}),$$

$$L'(\mathcal{D}, M; g) = \sum_{j \in M} L_C(\mathcal{D}; g; j) \tag{41}$$

**Corollary E.6.**

$$L^-(\mathcal{D}; g) = L_{Total}(\mathcal{D}; g) - L'(\mathcal{D}, M; g) \tag{42}$$

$$\|\nabla_g^2 L^-(\mathcal{D}; g_1) - \nabla_g^2 L^-(\mathcal{D}; g_2)\|_2 \leq ((k + |M|) \cdot C_H) \|g_1 - g_2\|$$

*Define $C_H^- \triangleq (k + |M|) \cdot C_H$*

Based on above corollaries and assumptions, we derive the following theorem.

**Theorem E.7.** *We obtain the error between the actual influence and our predicted influence as follows:*

$$\left\|\hat{g}^*_{-p_M}(\mathcal{D}) - \bar{g}_{-p_M}(\mathcal{D})\right\|$$

$$\leq \frac{C_H^- C'_L |M|^2}{2(\sigma'_{min} + \delta)^3} + \left|\frac{2\delta + \sigma_{min} + \sigma'_{min}}{(\delta + \sigma'_{min}) \cdot (\delta + \sigma_{min})}\right| \cdot C'_L |M|.$$

*Proof.* We will use the one-step Newton approximation as an intermediate step. Define $\Delta g_{Nt}(\mathcal{D})$ as

$$\Delta g_{Nt}(\mathcal{D}) \triangleq H_\delta^{-1} \cdot \nabla_g L'(\mathcal{D}, M; \hat{g}),$$

where $H_\delta = \delta \cdot I + \nabla_g^2 L^-(\mathcal{D}; \hat{g})$ is the regularized empirical Hessian at $\hat{g}$ but reweighed after removing the influence of wrong data. Then the one-step Newton approximation for $\hat{g}^*_{-p_M}(\mathcal{D})$ is defined as $g_{Nt}(\mathcal{D}) \triangleq \Delta g_{Nt}(\mathcal{D}) + \hat{g}$.

In the following, we will separate the error between $\bar{g}_{-p_M}(\mathcal{D})$ and $\hat{g}^*_{-p_M}(\mathcal{D})$ into the following two parts:

$$\hat{g}^*_{-p_M}(\mathcal{D}) - \bar{g}_{-p_M}(\mathcal{D}) = \underbrace{\hat{g}^*_{-p_M}(\mathcal{D}) - g_{Nt}(\mathcal{D})}_{\text{Err}_{\text{Nt, act}}(\mathcal{D})} + \underbrace{(g_{Nt}(\mathcal{D}) - \hat{g}) - (\bar{g}_{-p_M}(\mathcal{D}) - \hat{g})}_{\text{Err}_{\text{Nt, if}}(\mathcal{D})}$$

Firstly, in **Step** 1, we will derive the bound for Newton-actual error $\text{Err}_{\text{Nt, act}}(\mathcal{D})$. Since $L^-(g)$ is strongly convex with parameter $\sigma'_{min} + \delta$ and minimized by $\hat{g}^*_{-p_M}(\mathcal{D})$, we can bound the distance $\left\|\hat{g}^*_{-p_M}(\mathcal{D}) - g_{Nt}(\mathcal{D})\right\|_2$ in terms of the norm of the gradient at $g_{Nt}$:

$$\left\|\hat{g}^*_{-p_M}(\mathcal{D}) - g_{Nt}(\mathcal{D})\right\|_2 \leq \frac{2}{\sigma'_{min} + \delta} \left\|\nabla_g L^- (g_{Nt}(\mathcal{D}))\right\|_2 \tag{43}$$

Therefore, the problem reduces to bounding $\left\|\nabla_g L^- \left(g_{Nt}(\mathcal{D})\right)\right\|_2$. Noting that $\nabla_g L'(\hat{g}) = -\nabla_g L^-$. This is because $\hat{g}$ minimizes $L^- + L'$, that is,

$$\nabla_g L^- (\hat{g}) + \nabla_g L'(\hat{g}) = 0.$$

Recall that $\Delta g_{Nt} = H_\delta^{-1} \cdot \nabla_g L'(\mathcal{D}, S_e; \hat{g}) = -H_\delta^{-1} \cdot \nabla_g L^-(\mathcal{D}; \hat{g})$. Given the above conditions, we can have this bound for $\mathrm{Err}_{\mathrm{Nt, act}}(-\mathcal{D})$.

$$
\begin{aligned}
&\left\|\nabla_g L^- \left(g_{Nt}(\mathcal{D})\right)\right\|_2 \\
&= \left\|\nabla_g L^- (\hat{g} + \Delta g_{Nt}(\mathcal{D}))\right\|_2 \\
&= \left\|\nabla_g L^- (\hat{g} + \Delta g_{N_t}(\mathcal{D})) - \nabla_g L^- (\hat{g}) - \nabla_g^2 L^- (\hat{g}) \cdot \Delta g_{N_t}(\mathcal{D})\right\|_2 \\
&= \left\|\int_0^1 \left(\nabla_g^2 L^- (\hat{g} + t \cdot \Delta g_{Nt}(\mathcal{D})) - \nabla_g^2 L^- (\hat{g})\right) \Delta g_{Nt}(\mathcal{D}) \, dt\right\|_2 \\
&\leq \frac{C_H^-}{2} \left\|\Delta g_{Nt}(\mathcal{D}^*)\right\|_2^2 = \frac{C_H^-}{2} \left\|\left[\nabla_g^2 L^-(\hat{g})\right]^{-1} \nabla_g L^-(\hat{g})\right\|_2^2 \\
&\leq \frac{C_H^-}{2(\sigma'_{\min} + \delta)^2} \left\|\nabla_g L^-(\hat{g})\right\|_2^2 = \frac{C_H^-}{2(\sigma'_{\min} + \delta)^2} \left\|\nabla_g L'(\hat{g})\right\|_2^2 \\
&\leq \frac{C_H^- C_L' |M|^2}{2(\sigma'_{\min} + \delta)^2}.
\end{aligned}
\tag{44}
$$

Now we come to **Step** 2 to bound $\mathrm{Err}_{\mathrm{Nt, if}}(-\mathcal{D})$, and we will bound the difference in parameter change between Newton and our ECBM method.

$$
\begin{aligned}
&\left\|(g_{Nt}(\mathcal{D}) - \hat{g}) - (\bar{g}_{-p_M}(\mathcal{D}) - \hat{g})\right\| \\
&= \left\|\left[\left(\delta \cdot I + \nabla_g^2 L^- (\hat{g})\right)^{-1} + \left(\delta \cdot I + \nabla_g^2 L_{\mathrm{Total}}(\hat{g})\right)^{-1}\right] \cdot \nabla_g L'(\mathcal{D}, S_e; \hat{g})\right\|
\end{aligned}
$$

For simplification, we use matrix $A$, $B$ for the following substitutions:

$$A = \delta \cdot I + \nabla_g^2 L^- (\hat{g})$$
$$B = \delta \cdot I + \nabla_g^2 L_{\mathrm{Total}}(\hat{g})$$

And $A$ and $B$ are positive definite matrices with the following properties

$$\delta + \sigma'_{\min} \prec A \prec \delta + \sigma'_{\max}$$
$$\delta + \sigma_{\min} \prec B \prec \delta + \sigma_{\max}$$

Therefore, we have

$$
\begin{aligned}
&\left\|(g_{Nt}(\mathcal{D}) - \hat{g}) - (\bar{g}_{-p_M}(\mathcal{D}) - \hat{g})\right\| \\
&= \left\|\left(A^{-1} + B^{-1}\right) \cdot \nabla_g L^-(\mathcal{D}; \hat{g})\right\| \\
&\leq \left\|A^{-1} + B^{-1}\right\| \cdot \left\|\nabla_g L^-(\mathcal{D}; \hat{g})\right\| \\
&\leq \left|\frac{2\delta + \sigma_{\min} + \sigma'_{\min}}{(\delta + \sigma'_{\min}) \cdot (\delta + \sigma_{\min})}\right| \cdot \left\|\nabla_g L^-(\mathcal{D}; \hat{g})\right\| \\
&\leq \left|\frac{2\delta + \sigma_{\min} + \sigma'_{\min}}{(\delta + \sigma'_{\min}) \cdot (\delta + \sigma_{\min})}\right| \cdot C_L' |M|
\end{aligned}
\tag{45}
$$

By combining the conclusions from Step I and Step II in Equations 43, 44 and 45, we obtain the error between the actual influence and our predicted influence as follows:

$$
\begin{aligned}
&\left\|\hat{g}^*_{-p_M}(\mathcal{D}) - \bar{g}_{-p_M}(\mathcal{D})\right\| \\
&\leq \frac{C_H^- C_L' |M|^2}{2(\sigma'_{\min} + \delta)^3} + \left|\frac{2\delta + \sigma_{\min} + \sigma'_{\min}}{(\delta + \sigma'_{\min}) \cdot (\delta + \sigma_{\min})}\right| \cdot C_L' |M|.
\end{aligned}
$$

$\square$

*Remark* E.8. Theorem E.7 reveals one key finding about influence function estimation: The estimation error scales inversely with the regularization parameter $\delta$ ($\mathcal{O}(1/\delta)$), indicating that increased regularization improves approximation accuracy. Besides, the error bound is linearly increasing with the number of removed concepts $|M|$. This implies that the estimation error increases with the number of erroneous concepts removed.

# F. Data-level Influence

## F.1. Proof of Data-level Influence Function

We address situations that for dataset $\mathcal{D} = \{(x_i, y_i, c_i)\}_{i=1}^n$, given a set of data $z_r = (x_r, y_r, c_r)$, $r \in G$ to be removed. Our goal is to estimate $\hat{g}_{-z_G}$, $\hat{f}_{-z_G}$, which is the concept and label predictor trained on the $z_r$ for $r \in G$ removed dataset.

**Proof Sketch.** (i) First, we estimate the retrained concept predictor $\hat{g}_{-z_G}$. (ii) Then, we define a new label predictor $\tilde{f}_{-z_G}$ and estimate $\tilde{f}_{-z_G} - \hat{f}$. (iii) Next, in order to reduce computational complexity, use the lemma method to obtain the approximation of the Hessian matrix of $\tilde{f}_{-z_G}$. (iv) Next, we compute the difference $\hat{f}_{-z_G} - \tilde{f}_{-z_G}$ as

$$-H_{\tilde{f}_{-z_G}}^{-1} \cdot \left( \nabla_{\hat{f}} L_Y \left( \tilde{f}_{-z_G}(\hat{g}_{-z_G}(x_{i_r})), y_{i_r} \right) - \nabla_{\hat{f}} L_Y \left( \tilde{f}_{-z_G}(\hat{g}(x_{i_r})), y_{i_r} \right) \right).$$

(v) Finally, we divide $\hat{f}_{-z_G} - \hat{f}$, which we actually concerned with, into $\left( \hat{f}_{-z_G} - \tilde{f}_{-z_G} \right) + \left( \tilde{f}_{-z_G} - \hat{f} \right)$.

**Theorem F.1.** *For dataset $\mathcal{D} = \{(x_i, y_i, c_i)\}_{i=1}^n$, given a set of data $z_r = (x_r, y_r, c_r)$, $r \in G$ to be removed. Suppose the updated concept predictor $\hat{g}_{-z_G}$ is defined by*

$$\hat{g}_{-z_G} = \arg\min_g \sum_{j \in [k]} \sum_{i \in [n]-G} L_{C_j}(\hat{g}(x_i), c_i)$$

*where $L_C(\hat{g}(x_i), c_i) \triangleq \sum_{j=1}^k L_C^j(\hat{g}(x_i), c_i)$. Then we have the following approximation for $\hat{g}_{-z_G}$*

$$\hat{g}_{-z_G} \approx \bar{g}_{-z_G} \triangleq \hat{g} + H_{\hat{g}}^{-1} \cdot \sum_{r \in G} \sum_{j=1}^M G_C^j(x_r, c_r; \hat{g}), \tag{46}$$

*where $H_{\hat{g}} = \nabla_g \sum_{i,j} G_C^j(x_i, c_i; \hat{g})$ is the Hessian matrix of the loss function with respect to $\hat{g}$.*

*Proof.* Firstly, we rewrite $\hat{g}_{-z_G}$ as

$$\hat{g}_{-z_G} = \arg\min_g \left[ \sum_{i=1}^n L_C(\hat{g}(x_i), c_i) - \sum_{r \in G} L_C(g(x_r), c_r) \right],$$

Then we up-weighted the $r$-th data by some $\epsilon$ and have a new predictor $\hat{g}_{-z_G, \epsilon}$, which is abbreviated as $\hat{g}_\epsilon$:

$$\hat{g}_\epsilon \triangleq \arg\min_g \left[ \sum_{i=1}^n L_C(g(x_i), c_i) - \epsilon \cdot \sum_{r \in G} L_C(g(x_r), c_r) \right]. \tag{47}$$

Because $\hat{g}_\epsilon$ minimizes the right side of equation 47, we have

$$\nabla_{\hat{g}_\epsilon} \sum_{i=1}^n L_Y(\hat{g}_\epsilon(x_i), c_i) - \epsilon \cdot \nabla_{\hat{g}_\epsilon} \sum_{r \in G} L_Y(\hat{g}_\epsilon(x_r), c_r) = 0.$$

When $\epsilon \to 0$, $\hat{g}_\epsilon \to \hat{g}$. So we can perform a first-order Taylor expansion with respect to $\hat{g}$, and we have

$$\nabla_g \sum_{i=1}^n L_C(\hat{g}(x_i), c_i) - \epsilon \cdot \nabla_g \sum_{r \in G} L_C(\hat{g}(x_r), c_r) + \nabla_g^2 \sum_{i=1}^n L_C(\hat{g}(x_i), c_i) \cdot (\hat{g}_\epsilon - \hat{g}) \approx 0. \tag{48}$$

Recap the definition of $\hat{g}$:

$$\hat{g} = \arg\min_g \sum_{i=1}^{n} L_Y(g(x_i), c_i),$$

Then, the first term of equation 48 equals 0. Let $\epsilon \to 0$, then we have

$$\left. \frac{\mathrm{d}\hat{g}_\epsilon}{\mathrm{d}\epsilon} \right|_{\epsilon=0} = H_{\hat{g}}^{-1} \cdot \sum_{r \in G} \nabla_g L_C(\hat{g}(x_r), c_r),$$

where $H_{\hat{g}} = \nabla_g^2 \sum_{i=1}^{n} \ell(\hat{g}(x_i), c_i)$.

Remember when $\epsilon \to 0$, $\hat{g}_\epsilon \to \hat{g}_{-z_G}$. Perform a Newton step at $\hat{g}$, then we obtain the method to edit the original concept predictor under concept level:

$$\hat{g}_{-z_G} \approx \bar{g}_{-z_G} \triangleq \hat{g} + H_{\hat{g}}^{-1} \cdot \sum_{r \in G} \nabla_g L_C(\hat{g}(x_r), c_r).$$

Recall the definition of $G_C^j(x_i, c_i; \hat{g})$, then we can edit the initial $\hat{g}$ to $\hat{g}_{-p_G}^*$, defined as:

$$\hat{g}_{-z_G} \approx \bar{g}_{-z_G} \triangleq \hat{g} + H_{\hat{g}}^{-1} \cdot \sum_{r \in G} \sum_{j=1}^{M} G_C^j(x_r, c_r; \hat{g}), \tag{49}$$

where $H_{\hat{g}} = \nabla_g \sum_{i,j} G_C^j(x_i, c_i; \hat{g})$ is the Hessian matrix of the loss function with respect to $\hat{g}$. □

**Theorem F.2.** *For dataset* $\mathcal{D} = \{(x_i, y_i, c_i)\}_{i=1}^{n}$, *given a set of data* $z_r = (x_r, y_r, c_r)$, $r \in G$ *to be removed. The label predictor* $\hat{f}_{-z_G}$ *trained on the revised dataset becomes*

$$\hat{f}_{-z_G} = \arg\min_f \sum_{i \in [n]-G} L_{Y_i}(f, \hat{g}_{-z_G}). \tag{50}$$

*The intermediate label predictor* $\tilde{f}_{-z_G}$ *is defined by*

$$\tilde{f}_{-z_G} = \arg\min_f \sum_{i \in [n]-G} L_{Y_i}(f, \hat{g}),$$

*Then* $\tilde{f}_{-z_G} - \hat{f}$ *can be approximated by*

$$\tilde{f}_{-z_G} = \arg\min_f \sum_{i \in [n]-G} L_Y(f(\hat{g}(x_i)), y_i). \tag{51}$$

*We denote the edited version of* $\tilde{f}_{-z_G}$ *as* $\bar{f}_{-z_G}^* \triangleq \hat{f} + A_G$. *Define* $B_G$ *as*

$$-H_{\bar{f}_{-z_G}^*}^{-1} \sum_{i \in [n]-G} G_Y(x_i; \bar{g}_{-z_G}, \bar{f}_{-z_G}^*) - G_Y(x_i; \hat{g}, \bar{f}_{-z_G}^*),$$

*where* $H_{\bar{f}_{-z_G}^*} = \nabla_{\bar{f}} \sum_{i \in [n]-G} G_Y(x_i; \hat{g}, \bar{f}_{-z_G}^*)$ *is the Hessian matrix concerning* $\bar{f}_{-z_G}^*$. *Then* $\hat{f}_{-z_G}$ *can be estimated by* $\tilde{f}_{-z_G} + B_G$. *Combining the above two-stage approximation, then, the final edited label predictor* $\bar{f}_{-z_G}$ *can be obtained by*

$$\bar{f}_{-z_G} = \bar{f}_{-z_G}^* + B_G = \hat{f} + A_G + B_G. \tag{52}$$

*Proof.* We can see that there is a huge gap between $\hat{f}_{-z_G}$ and $\hat{f}$. Thus, firstly, we define $\tilde{f}_{-z_G}$ as

$$\tilde{f}_{-z_G} = \arg\min_f \sum_{i=1}^{n} L_Y\left(f(\hat{g}(x_i)), y_i\right) - \sum_{r \in G} L_Y\left(f(\hat{g}(x_r)), y_r\right).$$

Then, we define $\tilde{f}_{\epsilon,-z_G}$ as follows to estimate $\tilde{f}_{-z_G}$.

$$\tilde{f}_{\epsilon,-z_G} = \arg\min_f \sum_{i=1}^{n} L_Y\left(f(\hat{g}(x_i)), y_i\right) - \epsilon \cdot \sum_{r \in G} L_Y\left(f(\hat{g}(x_r)), y_r\right).$$

From the minimization condition, we have

$$\nabla_{\tilde{f}} \sum_{i=1}^{n} L_Y\left(\tilde{f}_{\epsilon,-z_G}(\hat{g}(x_i)), y_i\right) - \epsilon \cdot \sum_{r \in G} \nabla_{\tilde{f}} L_Y\left(\tilde{f}_{\epsilon,-z_G}(\hat{g}(x_r)), y_r\right) = 0.$$

Perform a first-order Taylor expansion at $\hat{f}$,

$$\nabla_{\hat{f}} \sum_{i=1}^{n} L_Y\left(\hat{f}(\hat{g}(x_i)), y_i\right) - \epsilon \cdot \nabla_{\hat{f}} \sum_{r \in G} L_Y\left(\hat{f}(\hat{g}(x_r)), y_r\right)$$
$$+ \nabla_{\hat{f}}^2 \sum_{i=1}^{n} L_Y\left(\hat{f}(\hat{g}(x_i)), y_i\right) \cdot \left(\tilde{f}_{\epsilon,-z_G} - \hat{f}\right) = 0.$$

Then $\tilde{f}_{-z_G}$ can be approximated by

$$\tilde{f}_{-z_G} \approx \hat{f} + H_{\hat{f}}^{-1} \cdot \sum_{r \in G} \nabla_{\hat{f}} L_Y\left(\hat{f}(\hat{g}(x_r)), y_r\right) \triangleq A_G. \tag{53}$$

Then the edit version of $\tilde{f}_{-z_G}$ is defined as

$$\bar{f}_{-z_G}^* = \hat{f} + A_G \tag{54}$$

Then we estimate the difference between $\hat{f}_{-z_G}$ and $\tilde{f}_{-z_G}$. Rewrite $\tilde{f}_{-z_G}$ as

$$\tilde{f}_{-z_G} = \arg\min_f \sum_{i \in S}^{n} L_Y\left(f(\hat{g}(x_i)), y_i\right), \tag{55}$$

where $S \triangleq [n] - G$.

Compare equation 50 with 55, we still need to define an intermediary predictor $\hat{f}_{-z_G,ir}$ as

$$\hat{f}_{-z_G,ir} = \arg\min_f \left[\sum_{\substack{i \in S \\ i \neq i_r}} L_{Y_i}\left(f, \hat{g}(x_i)\right) + L_{Y_{ir}}\left(f, \hat{g}_{-z_G}\right)\right]$$
$$= \arg\min_f \left[\sum_{i \in S} L_{Y_i}\left(f, \hat{g}\right) + L_{Y_{ir}}\left(f, \hat{g}_{-z_G}\right) - L_{Y_{ir}}\left(f, \hat{g}\right)\right].$$

Up-weight the $i_r$ data by some $\epsilon$, we define $\hat{f}_{\epsilon,-z_G,ir}$ as

$$\hat{f}_{\epsilon,-z_G,ir} = \arg\min_f \left[\sum_{i \in S} L_{Y_i}\left(f, \hat{g}\right) + \epsilon \cdot L_{Y_{ir}}\left(f, \hat{g}_{-z_G}\right) - \epsilon \cdot L_{Y_{ir}}\left(f, \hat{g}\right)\right].$$

We denote $\hat{f}_{\epsilon,-z_G,ir}$ as $\hat{f}_\epsilon^*$ in the following proof. Then, from the minimization condition, we have

$$\nabla_{\hat{f}} \sum_{i \in S} L_{Y_i}\left(\hat{f}_\epsilon^*, \hat{g}\right) + \epsilon \cdot \nabla_{\hat{f}} L_{Y_{ir}}\left(\hat{f}_\epsilon^*, \hat{g}_{-z_G}\right) - \epsilon \cdot \nabla_{\hat{f}} L_{Y_{ir}}\left(\hat{f}_\epsilon^*, \hat{g}(x_{i_r})\right). \tag{56}$$

When $\epsilon \to 0$, $\hat{f}_\epsilon^* \to \tilde{f}_{-z_G}$. Then we perform a Taylor expansion at $\tilde{f}_{-z_G}$ of equation 56 and have

$$\nabla_{\hat{f}} \sum_{i \in S} L_{Y_i}\left(\tilde{f}_{-z_G}, \hat{g}\right) + \epsilon \cdot \nabla_{\hat{f}} L_{Y_{ir}}\left(\tilde{f}_{-z_G}, \hat{g}_{-z_G}\right)$$
$$- \epsilon \cdot \nabla_{\hat{f}} L_{Y_{ir}}\left(\tilde{f}_{-z_G}, \hat{g}\right) + \nabla_{\hat{f}}^2 \sum_{i \in S} L_{Y_i}\left(\tilde{f}_{-z_G}, \hat{g}\right) \cdot (\hat{f}_\epsilon^* - \tilde{f}_{-z_G}) \approx 0.$$

Organizing the above equation gives

$$\hat{f}_\epsilon^* - \tilde{f}_{-z_G} \approx -\epsilon \cdot H_{\tilde{f}_{-z_G}}^{-1} \cdot \left(\nabla_{\hat{f}} L_{Y_{ir}}\left(\tilde{f}_{-z_G}, \hat{g}_{-z_G}\right) - \nabla_{\hat{f}} L_{Y_{ir}}\left(\tilde{f}_{-z_G}, \hat{g}\right)\right),$$

where $H_{\tilde{f}_{-z_G}} = \nabla_{\hat{f}}^2 \sum_{i \in S} L_{Y_i}\left(\tilde{f}_{-z_G}, \hat{g}\right)$.

When $\epsilon = 1$, $\hat{f}_\epsilon^* = \hat{f}_{-z_G, ir}$. Then we perform a Newton iteration with step size 1 at $\tilde{f}_{-z_G}$,

$$\hat{f}_{-z_G, ir} - \tilde{f}_{-z_G} \approx -H_{\tilde{f}_{-z_G}}^{-1} \cdot \left(\nabla_{\hat{f}} L_{Y_{ir}}\left(\tilde{f}_{-z_G}, \hat{g}_{-z_G}\right) - \nabla_{\hat{f}} L_{Y_{ir}}\left(\tilde{f}_{-z_G}, \hat{g}\right)\right)$$

Iterate $i_r$ through set $S$, and we have

$$\hat{f}_{-z_G} - \tilde{f}_{-z_G} \approx -H_{\tilde{f}_{-z_G}}^{-1} \cdot \left(\nabla_{\hat{f}} \sum_{i \in S} L_{Y_i}\left(\tilde{f}_{-z_G}, \hat{g}_{-z_G}\right) - \nabla_{\hat{f}} \sum_{i \in S} L_{Y_i}\left(\tilde{f}_{-z_G}, \hat{g}\right)\right) \tag{57}$$

The edited version of $\hat{g}_{-z_G}$ has been deduced as $\bar{g}_{-z_G}$ in theorem F.1, substituting this approximation into (57), then we have

$$\hat{f}_{-z_G} - \tilde{f}_{-z_G} \approx -H_{\tilde{f}_{-z_G}}^{-1} \cdot \left(\nabla_{\hat{f}} \sum_{i \in S} L_{Y_i}\left(\tilde{f}_{-z_G}, \bar{g}_{-z_G}\right) - \nabla_{\hat{f}} \sum_{i \in S} L_{Y_i}\left(\tilde{f}_{-z_G}, \hat{g}\right)\right). \tag{58}$$

Noting that we cannot obtain $\hat{f}_{-z_G}$ and $H_{\tilde{f}_{-z_G}}$ directly because we do not retrain the label predictor but edit it to $\bar{f}_{-z_G}^*$ as a substitute. Therefore, we approximate $\hat{f}_{-z_G}$ with $\bar{f}_{-z_G}^*$ and $H_{\tilde{f}_{-z_G}}$ with $H_{\bar{f}_{-z_G}^*}$ which is defined by:

$$H_{\bar{f}_{-z_G}^*} = \nabla_{\hat{f}}^2 \sum_{i \in S} L_{Y_i}\left(\bar{f}_{-z_G}^*, \hat{g}\right)$$

Then we define $B_G$ as

$$B_G \triangleq -H_{\bar{f}_{-z_G}^*}^{-1} \cdot \left(\nabla_{\hat{f}} \sum_{i \in S} L_{Y_i}\left(\bar{f}_{-z_G}^*, \bar{g}_{-z_G}\right) - \nabla_{\hat{f}} \sum_{i \in S} L_{Y_i}\left(\bar{f}_{-z_G}^*, \hat{g}\right)\right) \tag{59}$$

Combining (54) and (59), then we deduce the final closed-form edited label predictor as

$$\bar{f}_{-z_G} = \bar{f}_{-z_G}^* + B_G = \hat{f} + A_G + B_G.$$

Replace the definition of gradient of $L_C$ and $L_C$, then we obtain the final version of approximation. $\qquad\square$

## F.2. Theoretical Bound for the Influence Function

Consider the dataset $\mathcal{D} = \{(x_i, c_i, y_i\}_{i=1}^n$, the loss function of the concept predictor $g$ is defined as:

$$L_{\text{Total}}(\mathcal{D}; g) = \sum_{i=1}^n L_C(g(x_i), c_i) + \frac{\delta}{2} \cdot \|g\|^2 = \sum_{i=1}^n \sum_{j=1}^k L_C^j(g(x_i), c_i) + \frac{\delta}{2} \cdot \|g\|^2 = \sum_{i=1}^n \sum_{j=1}^k g^j(x_i)^\top \log(c_i{}^j) + \frac{\delta}{2} \cdot \|g\|^2.$$

Mathematically, we have a set of erroneous data $z_r = (x_r, y_r, c_r)$, $r \in G$ need to be removed. Then the retrained concept predictor becomes

$$\hat{g}_{-z_G} = \arg\min_g \sum_{j=1}^{k} \sum_{i \in [n]-G} L_C^j(g(x_i), c_i) + \frac{\delta}{2} \cdot \|g\|^2.$$

Define the new dataset as $\mathcal{D}^* = \{(x_i, c_i, y_i)\}_{i \in [n]-G}$, then the loss function with the influence of erroneous data removed becomes

$$L^-(\mathcal{D}^*; g) = \sum_{j=1}^{k} \sum_{i \in [n]-G} L_C^j(g(x_i), c_i) + \frac{\delta}{2} \cdot \|g\|^2 = L_{\text{Total}}(\mathcal{D}; g) - \sum_{j=1}^{k} \sum_{i \in G} L_C^j(g(x_i), c_i). \tag{60}$$

Assume $\hat{g} = \arg\min L_{\text{Total}}(\mathcal{D}; g)$ is the original model parameter. $\hat{g}_{-z_G}$ is the minimizer of $L^-(\mathcal{D}^*; g)$. Denote $\bar{g}_{-z_G}$ as the updated model with the influence of erroneous data removed and is obtained by the influence function method in theorem F.1, which is an estimation for $\hat{g}_{-z_G}$.

$$\hat{g}_{-z_G} \approx \bar{g}_{-z_G} \triangleq \hat{g} + H_{\hat{g}}^{-1} \cdot \sum_{r \in G} \sum_{j=1}^{M} G_C^j(x_r, c_r; \hat{g}), \tag{61}$$

In this part, we will study the error between the estimated influence given by the theorem F.1 method and $\hat{g}_{-z_G}$. We use the parameter changes as the evaluation metric:

$$|(\bar{g}_{-z_G} - \hat{g}) - (\hat{g}_{-z_G} - \hat{g})| = |\bar{g}_{-z_G} - \hat{g}_{-z_G}| \tag{62}$$

**Assumption F.3.** The loss $L_C(x, c; g; j)$

$$L_C(x, c; g) = \sum_{j=1}^{k} L_C^j(g(x), c).$$

is convex and twice-differentiable in $g$, with positive regularization $\delta > 0$. There exists $C_H \in \mathbb{R}$ such that

$$\|\nabla_g^2 L_C(x, c; g_1) - \nabla_g^2 L_C(x, c; g_2)\|_2 \leq C_H \|g_1 - g_2\|_2$$

for all $(x, c) \in \mathcal{D}$ and $g_1, g_2 \in \Gamma$.

**Definition F.4.**

$$C_L' = \|\nabla_g L_C(\mathcal{D}; \hat{g})\|_2,$$

$$\sigma_{\min}' = \text{smallest singular value of } \nabla_g^2 L^-(\mathcal{D}; \hat{g}),$$

$$\sigma_{\min} = \text{smallest singular value of } \nabla_g^2 L_{\text{Total}}(\mathcal{D}; \hat{g}),$$

$$L'(\mathcal{D}, G; g) = \sum_{i \in G} L_C(x_i, c_i; g) \tag{63}$$

**Corollary F.5.**

$$L^-(\mathcal{D}; g) = L_{Total}(\mathcal{D}; g) - L'(\mathcal{D}, G; g) \tag{64}$$

$$\|\nabla_g^2 L^-(\mathcal{D}; g_1) - \nabla_g^2 L^-(\mathcal{D}; g_2)\|_2 \leq ((n + |G|) \cdot C_H) \|g_1 - g_2\|$$

*Define $C_H^- \triangleq (n + |G|) \cdot C_H$*

Based on above corollaries and assumptions, we derive the following theorem.

**Theorem F.6.** *We obtain the error between the actual influence and our predicted influence as follows:*

$$\|\hat{g}_{-z_G}(\mathcal{D}) - \bar{g}_{-z_G}(\mathcal{D})\|$$

$$\leq \frac{C_H^- C_L' |G|^2}{2(\sigma_{min}' + \delta)^3} + \left| \frac{2\delta + \sigma_{min} + \sigma_{min}'}{(\delta + \sigma_{min}') \cdot (\delta + \sigma_{min})} \right| \cdot C_L' |G|.$$

*Proof.* We will use the one-step Newton approximation as an intermediate step. Define $\Delta g_{Nt}(\mathcal{D})$ as

$$\Delta g_{Nt}(\mathcal{D}) \triangleq H_\delta^{-1} \cdot \nabla_g L'(\mathcal{D}, G; \hat{g}),$$

where $H_\delta = \delta \cdot I + \nabla_g^2 L^-(\mathcal{D}; \hat{g})$ is the regularized empirical Hessian at $\hat{g}$ but reweighed after removing the influence of wrong data. Then the one-step Newton approximation for $\hat{g}_{-z_G}(\mathcal{D})$ is defined as $g_{Nt}(\mathcal{D}) \triangleq \Delta g_{Nt}(\mathcal{D}) + \hat{g}$.

In the following, we will separate the error between $\bar{g}_{-z_G}(\mathcal{D})$ and $\hat{g}_{-z_G}(\mathcal{D})$ into the following two parts:

$$\hat{g}_{-z_G}(\mathcal{D}) - \bar{g}_{-z_G}(\mathcal{D}) = \underbrace{\hat{g}_{-z_G}(\mathcal{D}) - g_{Nt}(\mathcal{D})}_{\mathrm{Err}_{\mathrm{Nt,\,act}}(\mathcal{D})} + \underbrace{(g_{Nt}(\mathcal{D}) - \hat{g}) - (\bar{g}_{-z_G}(\mathcal{D}) - \hat{g})}_{\mathrm{Err}_{\mathrm{Nt,\,if}}(\mathcal{D})}$$

Firstly, in **Step** 1, we will derive the bound for Newton-actual error $\mathrm{Err}_{\mathrm{Nt,\,act}}(\mathcal{D})$. Since $L^-(g)$ is strongly convex with parameter $\sigma'_{\min} + \delta$ and minimized by $\hat{g}_{-z_G}(\mathcal{D})$, we can bound the distance $\|\hat{g}_{-z_G}(\mathcal{D}) - g_{Nt}(\mathcal{D})\|_2$ in terms of the norm of the gradient at $g_{Nt}$:

$$\|\hat{g}_{-z_G}(\mathcal{D}) - g_{Nt}(\mathcal{D})\|_2 \leq \frac{2}{\sigma'_{\min} + \delta} \left\| \nabla_g L^- \left( g_{Nt}(\mathcal{D}) \right) \right\|_2 \tag{65}$$

Therefore, the problem reduces to bounding $\| \nabla_g L^- (g_{Nt}(\mathcal{D})) \|_2$. Noting that $\nabla_g L'(\mathcal{D}, G; \hat{g}) = -\nabla_g L^-$. This is because $\hat{g}$ minimizes $L^- + L'$, that is,

$$\nabla_g L^-(\hat{g}) + \nabla_g L'(\mathcal{D}, G; \hat{g}) = 0.$$

Recall that $\Delta g_{Nt} = H_\delta^{-1} \cdot \nabla_g L'(\mathcal{D}, G; \hat{g}) = -H_\delta^{-1} \cdot \nabla_g L^-(\mathcal{D}; \hat{g})$. Given the above conditions, we can have this bound for $\mathrm{Err}_{\mathrm{Nt,\,act}}(-\mathcal{D})$.

$$
\begin{aligned}
& \left\| \nabla_g L^- \left( g_{Nt}(\mathcal{D}) \right) \right\|_2 \\
&= \left\| \nabla_g L^- \left( \hat{g} + \Delta g_{Nt}(\mathcal{D}) \right) \right\|_2 \\
&= \left\| \nabla_g L^- \left( \hat{g} + \Delta g_{N_t}(\mathcal{D}) \right) - \nabla_g L^-(\hat{g}) - \nabla_g^2 L^-(\hat{g}) \cdot \Delta g_{N_t}(\mathcal{D}) \right\|_2 \\
&= \left\| \int_0^1 \left( \nabla_g^2 L^- (\hat{g} + t \cdot \Delta g_{Nt}(\mathcal{D})) - \nabla_g^2 L^-(\hat{g}) \right) \Delta g_{Nt}(\mathcal{D}) \, dt \right\|_2 \\
&\leq \frac{C_H^-}{2} \|\Delta g_{Nt}(\mathcal{D}^*)\|_2^2 = \frac{C_H^-}{2} \left\| \left[ \nabla_g^2 L^-(\hat{g}) \right]^{-1} \nabla_g L^-(\hat{g}) \right\|_2^2 \\
&\leq \frac{C_H^-}{2(\sigma'_{\min} + \delta)^2} \left\| \nabla_g L^-(\hat{g}) \right\|_2^2 = \frac{C_H^-}{2(\sigma'_{\min} + \delta)^2} \|\nabla_g L'(\mathcal{D}, G; \hat{g})\|_2^2 \\
&\leq \frac{C_H^- C_L' |G|^2}{2(\sigma'_{\min} + \delta)^2}.
\end{aligned}
\tag{66}
$$

Now we come to **Step** 2 to bound $\mathrm{Err}_{\mathrm{Nt,\,if}}(-\mathcal{D})$, and we will bound the difference in parameter change between Newton and our ECBM method.

$$
\begin{aligned}
& \left\| (g_{Nt}(\mathcal{D}) - \hat{g}) - (\bar{g}_{-z_G}(\mathcal{D}) - \hat{g}) \right\| \\
&= \left\| \left[ \left( \delta \cdot I + \nabla_g^2 L^-(\hat{g}) \right)^{-1} + \left( \delta \cdot I + \nabla_g^2 L_{\mathrm{Total}}(\hat{g}) \right)^{-1} \right] \cdot \nabla_g L'(\mathcal{D}, G; \hat{g}) \right\|
\end{aligned}
$$

For simplification, we use matrix $A$, $B$ for the following substitutions:

$$
\begin{aligned}
A &= \delta \cdot I + \nabla_g^2 L^-(\hat{g}) \\
B &= \delta \cdot I + \nabla_g^2 L_{\mathrm{Total}}(\hat{g})
\end{aligned}
$$

And $A$ and $B$ are positive definite matrices with the following properties

$$
\begin{aligned}
\delta + \sigma'_{\min} &\prec A \prec \delta + \sigma'_{\max} \\
\delta + \sigma_{\min} &\prec B \prec \delta + \sigma_{\max}
\end{aligned}
$$

Therefore, we have

$$
\begin{aligned}
&\|(g_{Nt}(\mathcal{D}) - \hat{g}) - (\bar{g}_{-z_G}(\mathcal{D}) - \hat{g})\| \\
&= \left\|\left(A^{-1} + B^{-1}\right) \cdot \nabla_g L^-(\mathcal{D}; \hat{g})\right\| \\
&\leq \left\|A^{-1} + B^{-1}\right\| \cdot \left\|\nabla_g L^-(\mathcal{D}; \hat{g})\right\| \\
&\leq \left|\frac{2\delta + \sigma_{\min} + \sigma'_{\min}}{(\delta + \sigma'_{\min}) \cdot (\delta + \sigma_{\min})}\right| \cdot \left\|\nabla_g L^-(\mathcal{D}; \hat{g})\right\| \\
&\leq \left|\frac{2\delta + \sigma_{\min} + \sigma'_{\min}}{(\delta + \sigma'_{\min}) \cdot (\delta + \sigma_{\min})}\right| \cdot C'_L |G|
\end{aligned}
\tag{67}
$$

By combining the conclusions from Step I and Step II in Equations 65, 66 and 67, we obtain the error between the actual influence and our predicted influence as follows:

$$
\begin{aligned}
&\|\hat{g}_{-z_G}(\mathcal{D}) - \bar{g}_{-z_G}(\mathcal{D})\| \\
&\leq \frac{C_H^- C'_L |G|^2}{2(\sigma'_{\min} + \delta)^3} + \left|\frac{2\delta + \sigma_{\min} + \sigma'_{\min}}{(\delta + \sigma'_{\min}) \cdot (\delta + \sigma_{\min})}\right| \cdot C'_L |G|.
\end{aligned}
$$

$\square$

*Remark* F.7. The error bound is linearly increasing with the number of removed data $|G|$. This implies that the estimation error increases with the number of erroneous data removed.

# G. Algorithm

---

**Algorithm 1** Concept-label-level ECBM

---

1: **Input:** Dataset $\mathcal{D} = \{(x_i, y_i, c_i)\}_{i=1}^n$, original concept predictor $\hat{f}$, and label predictor $\hat{g}$, a set of erroneous data $D_e$ and its associated index set $S_e$.

2: For the index $(w, r)$ in $S_e$, correct $c_w^r$ to the right label $c_w^r{}'$ for the $w$-th data $(x_w, y_w, c_w)$.

3: Compute the Hessian matrix of the loss function respect to $\hat{g}$:

$$
H_{\hat{g}} = \nabla_{\hat{g}}^2 \sum_{i,j} L_{C_j}(\hat{g}^j(x_i), c_i^j).
$$

4: Update concept predictor $\tilde{g}$:

$$
\tilde{g} = \hat{g} - H_{\hat{g}}^{-1} \cdot \sum_{(w,r) \in S_e} \left(\nabla_{\hat{g}} L_{C_r}\left(\hat{g}^r(x_w), c_w^r{}'\right) - \nabla_{\hat{g}} L_{C_r}\left(\hat{g}^r(x_w), c_w^r\right)\right).
$$

5: Compute the Hessian matrix of the loss function respect to $\hat{f}$:

$$
H_{\hat{f}} = \nabla_{\hat{f}}^2 \sum_{i=1}^n L_{Y_i}(\hat{f}, \hat{g}).
$$

6: Update label predictor $\tilde{f}$:

$$
\tilde{f} = \hat{f} + H_{\hat{f}}^{-1} \cdot \nabla_f \sum_{i=1}^n L_Y\left(\hat{f}\left(\hat{g}(x_i)\right), y_i\right) - H_{\hat{f}}^{-1} \cdot \nabla_f \sum_{l=1}^n \left(L_Y\left(\hat{f}\left(\tilde{g}(x_l)\right), y_l\right)\right).
$$

7: **Return:** $\tilde{f}, \tilde{g}$.

---

**Algorithm 2** Concept-level ECBM

---

1: **Input:** Dataset $\mathcal{D} = \{(x_i, y_i, c_i)\}_{i=1}^n$, original concept predictor $\hat{f}$, label predictor $\hat{g}$ and the to be removed concept index set $M$.

2: For $r \in M$, set $p_r = 0$ for all the data $z \in \mathcal{D}$.

3: Compute the Hessian matrix of the loss function respect to $\hat{g}$:

$$H_{\hat{g}} = \nabla_{\hat{g}}^2 \sum_{j \notin M} \sum_{i=1}^{n} L_{C_j}(\hat{g}^j(x_i), c_i^j).$$

4: Update concept predictor $\tilde{g}^*$:

$$\tilde{g}^* = \hat{g} - H_{\hat{g}}^{-1} \cdot \nabla_{\hat{g}} \sum_{j \notin M} \sum_{i=1}^{n} L_{C_j}(\hat{g}^j(x_i), c_i^j).$$

5: Compute the Hessian matrix of the loss function respect to $\hat{f}$:

$$H_{\hat{f}} = \nabla_{\hat{f}}^2 \sum_{i=1}^{n} L_Y(\hat{f}(\hat{g}(x_i)), y_i).$$

6: Update label predictor $\tilde{f}$:

$$\tilde{f} = \hat{f} - H_{\hat{f}}^{-1} \cdot \nabla_{\hat{f}} \sum_{l=1}^{n} L_Y\left(\hat{f}(\tilde{g}^*(x_l)), y_l\right).$$

7: Map $\tilde{g}^*$ to $\tilde{g}$ by removing the $r$-th row of the matrix in the final layer of $\tilde{g}^*$ for $r \in M$.

8: **Return:** $\tilde{f}, \tilde{g}$.

---

**Algorithm 3** Data-level ECBM

1: **Input:** Dataset $\mathcal{D} = \{(x_i, y_i, c_i)\}_{i=1}^{N}$, original concept predictor $\hat{f}$, label predictor $\hat{g}$, and the to be removed data index set $G$.

2: For $r \in G$, remove the $r$-th data $(x_r, y_r, c_r)$ from $\mathcal{D}$ and define the new dataset as $\mathcal{S}$.

3: Compute the Hessian matrix of the loss function with respect to $\hat{g}$:

$$H_{\hat{g}} = \nabla_{\hat{g}}^2 \sum_{i,j} L_{C_j}(\hat{g}^j(x_i), c_i^j).$$

4: Update concept predictor $\tilde{g}$:

$$\tilde{g} = \hat{g} + H_{\hat{g}}^{-1} \cdot \sum_{r \in G} \nabla_g L_C(\hat{g}(x_r), c_r)$$

5: Update label predictor $\tilde{f}$. Compute the Hessian matrix of the loss function with respect to $\hat{f}$:

$$H_{\hat{f}} = \nabla_{\hat{f}}^2 \sum_{i=1}^{n} L_Y(\hat{f}(\hat{g}(x_i)), y_i).$$

6: Compute $A$ as:

$$A = H_{\hat{f}}^{-1} \cdot \sum_{i \in [n] - G} \nabla_{\hat{f}} L_Y\left(\hat{f}(\hat{g}(x_i)), y_i\right)$$

7: Obtain $\bar{f}$ as

$$\bar{f} = \hat{f} + A$$

8: Compute the Hessian matrix of the loss function concerning $\bar{f}$:

$$H_{\bar{f}} = \nabla_{\bar{f}}^2 \sum_{i \in [n] - G} L_Y(\bar{f}(\hat{g}(x_i)), y_i).$$

9: Compute $B$ as

$$B = -H_{\bar{f}}^{-1} \cdot \sum_{i \in [n] - G} \nabla_{\hat{f}} \left(L_Y(\bar{f}(\tilde{g}(x_i)), y_i) - L_Y(\bar{f}(\hat{g}(x_i)), y_i)\right)$$

10: Update the label predictor $\tilde{f}$ as: $\tilde{f} = \hat{f} + A + B$.
11: **Return:** $\tilde{f}, \tilde{g}$.

---

**Algorithm 4** EK-FAC for Concept Predictor $g$

---

1: **Input:** Dataset $\mathcal{D} = \{(x_i, y_i, c_i)\}_{i=1}^N$, original concept predictor $\hat{g}$.
2: **for** the $l$-th convolution layer of $\hat{g}$: **do**
3:     Define the input activations $\{a_{j,t}\}$, weights $W = (w_{i,j,\delta})$, and biases $b = (b_i)$ of this layer;
4:     Obtain the expanded activations $[\![A_{l-1}]\!]$ as:

$$[\![A_{l-1}]\!]_{t,j|\Delta|+\delta} = [A_{l-1}]_{(t+\delta),j} = a_{j,t+\delta},$$

5:     Compute the pre-activations:

$$[S_l]_{i,t} = s_{i,t} = \sum_{\delta \in \Delta} w_{i,j,\delta} a_{j,t+\delta} + b_i.$$

6:     During the backpropagation process, obtain the $\mathcal{D}s_{i,t}$ as:

$$\mathcal{D}s_{i,t} = \frac{\partial \sum_{j=1}^k \sum_{i=1}^n L_{C_j}}{\partial s_{i,t}}$$

7:     Compute $\hat{\Omega}_{l-1}$ and $\hat{\Gamma}_l$:

$$\hat{\Omega}_{l-1} = \frac{1}{n} \sum_{i=1}^n \left([\![A_{l-1}^i]\!]_{\mathrm{H}}^\top [\![A_{l-1}^i]\!]_{\mathrm{H}}\right)$$

$$\hat{\Gamma}_l = \frac{1}{n} \sum_{i=1}^n \left(\frac{1}{|\mathcal{T}|} \mathcal{D}S_l^{i\top} \mathcal{D}S_l^i\right)$$

8:     Perform eigenvalue decomposition of $\hat{\Omega}_{l-1}$ and $\hat{\Gamma}_l$, obtain $Q_\Omega, \Lambda_\Omega, Q_\Gamma, \Lambda_\Gamma$, which satisfies

$$\hat{\Omega}_{l-1} = Q_\Omega \Lambda_\Omega Q_\Omega^\top$$

$$\hat{\Gamma}_l = Q_\Gamma \Lambda_\Gamma Q_\Gamma^\top$$

9:     Define a diagonal matrix $\Lambda$ and compute the diagonal element as

$$\Lambda_{ii}^* = n^{-1} \sum_{j=1}^n \left(\left(Q_{\Omega_{l-1}} \otimes Q_{\Gamma_l}\right) \nabla_{\theta_l} L_{C_j}\right)_i^2.$$

10:     Compute $\hat{H}_l^{-1}$ as

$$\hat{H}_l^{-1} = \left(Q_{\Omega_{l-1}} \otimes Q_{\Gamma_l}\right) \left(\Lambda + \lambda_l I_{d_l}\right)^{-1} \left(Q_{\Omega_{l-1}} \otimes Q_{\Gamma_l}\right)^{\mathrm{T}}$$

11: **end for**
12: Splice $H_l$ sequentially into large diagonal matrices

$$\hat{H}_{\hat{g}}^{-1} = \begin{pmatrix} \hat{H}_1^{-1} & & \mathbf{0} \\ & \ddots & \\ \mathbf{0} & & \hat{H}_d^{-1} \end{pmatrix}$$

    where $d$ is the number of the convolution layer of the concept predictor.
13: **Return: the inverse Hessian matrix** $\hat{H}_{\hat{g}}^{-1}$.

---

**Algorithm 5** EK-FAC for Label Predictor $f$

---

1: **Input:** Dataset $\mathcal{D} = \{(x_i, y_i, c_i)\}_{i=1}^N$, original label predictor $\hat{f}$.

2: Denote the pre-activated output of $\hat{f}$ as $f'$, Compute $A$ as

$$A = \frac{1}{n} \cdot \sum_{i=1}^{n} \hat{g}(x_i) \cdot \hat{g}(x_i)^{\mathrm{T}}$$

3: Comput $B$ as:

$$B = \frac{1}{n} \cdot \sum_{i=1}^{n} \nabla_{f'} L_Y (\hat{f}(\hat{g}(x_i)), y_i) \cdot \nabla_{f'} L_Y (\hat{f}(\hat{g}(x_i)), y_i)^{\mathrm{T}}$$

4: Perform eigenvalue decomposition of AA and BB, obtain $Q_A, \Lambda_A, Q_B, \Lambda_B$, which satisfies

$$A = Q_A \Lambda_A Q_A^{\top}$$
$$B = Q_B \Lambda_B Q_B^{\top}$$

5: Define a diagonal matrix $\Lambda$ and compute the diagonal element as

$$\Lambda_{ii}^* = n^{-1} \sum_{j=1}^{n} \left( (Q_A \otimes Q_B) \nabla_{\hat{f}} L_{Y_j} \right)_i^2.$$

6: Compute $\hat{H}_{\hat{f}}^{-1}$ as

$$\hat{H}_{\hat{f}}^{-1} = (Q_A \otimes Q_B) (\Lambda + \lambda I_d)^{-1} (Q_A \otimes Q_B)^{\mathrm{T}}$$

7: **Return: the inverse Hessian matrix $\hat{H}_{\hat{f}}^{-1}$.**

---

**Algorithm 6** EK-FAC Concept-label-level ECBM

1: **Input:** Dataset $\mathcal{D} = \{(x_i, y_i, c_i)\}_{i=1}^{N}$, original concept predictor $\hat{f}$, label predictor $\hat{g}$, and the to be removed data index set $G$, and damping parameter $\lambda$.
2: For $r \in G$, remove the $r$-th data $(x_r, y_r, c_r)$ from $\mathcal{D}$ and define the new dataset as $\mathcal{S}$.
3: **Use EK-FAC method in algorithm 4 to accelerate iHVP problem for $\hat{g}$ and obtain the inverse Hessian matrix $\hat{H}_{\hat{g}}^{-1}$**
4: Update concept predictor $\tilde{g}$:

$$\tilde{g} = \hat{g} - H_{\hat{g}}^{-1} \cdot \sum_{(w,r) \in S_e} \left( \nabla_{\hat{g}} L_{C_r} (\hat{g}^r(x_w), c_w^r{}') - \nabla_{\hat{g}} L_{C_r} (\hat{g}^r(x_w), c_w^r) \right).$$

5: **Use EK-FAC method in algorithm 5 to accelerate iHVP problem for $\hat{f}$ and obtain $\hat{H}_{\hat{f}}^{-1}$**
6: Update label predictor $\tilde{f}$:

$$\tilde{f} = \hat{f} + H_{\hat{f}}^{-1} \cdot \nabla_f \sum_{i=1}^{n} L_Y \left( \hat{f}(\hat{g}(x_i)), y_i \right) - H_{\hat{f}}^{-1} \cdot \nabla_f \sum_{l=1}^{n} \left( L_Y \left( \hat{f}(\tilde{g}(x_l)), y_l \right) \right).$$

7: **Return:** $\tilde{f}, \tilde{g}$.

---

**Algorithm 7** EK-FAC Concept-level ECBM

1: **Input:** Dataset $\mathcal{D} = \{(x_i, y_i, c_i)\}_{i=1}^{n}$, original concept predictor $\hat{f}$, label predictor $\hat{g}$ and the to be removed concept index set $M$, and damping parameter $\lambda$.
2: For $r \in M$, set $p_r = 0$ for all the data $z \in \mathcal{D}$.
3: **Use EK-FAC method in algorithm 4 to accelerate iHVP problem for $\hat{g}$ and obtain the inverse Hessian matrix $\hat{H}_{\hat{g}}^{-1}$**
4: Update concept predictor $\tilde{g}$:

$$\tilde{g}^* = \hat{g} - H_{\hat{g}}^{-1} \cdot \nabla_{\hat{g}} \sum_{j \notin M} \sum_{i=1}^{n} L_{C_j} (\hat{g}^j(x_i), c_i^j).$$

5: **Use EK-FAC method in algorithm 5 to accelerate iHVP problem for $\hat{f}$ and obtain $\hat{H}_{\hat{f}}^{-1}$**

6: Update label predictor $\tilde{f}$:

$$\tilde{f} = \hat{f} - H_{\hat{f}}^{-1} \cdot \nabla_{\hat{f}} \sum_{l=1}^{n} L_Y \left( \hat{f} \left( \tilde{g}^*(x_l) \right), y_l \right).$$

7: Map $\tilde{g}^*$ to $\tilde{g}$ by removing the $r$-th row of the matrix in the final layer of $\tilde{g}^*$ for $r \in M$.
8: **Return:** $\tilde{f}, \tilde{g}$.

---

**Algorithm 8** EK-FAC Data-level ECBM

---

1: **Input:** Dataset $\mathcal{D} = \{(x_i, y_i, c_i)\}_{i=1}^{n}$, original concept predictor $\hat{f}$, and label predictor $\hat{g}$, a set of erroneous data $D_e$ and its associated index set $S_e$, and damping parameter $\lambda$.
2: For the index $(w, r)$ in $S_e$, correct $c_w^r$ to the right label $c_w^r{}'$ for the $w$-th data $(x_w, y_w, c_w)$.
3: **Use EK-FAC method in algorithm 4 to accelerate iHVP problem for $\hat{g}$ and obtain the inverse Hessian matrix $\hat{H}_{\hat{g}}^{-1}$**
4: Update concept predictor $\tilde{g}$:

$$\tilde{g} = \hat{g} - H_{\hat{g}}^{-1} \cdot \sum_{(w,r) \in S_e} \left( \nabla_{\hat{g}} L_{C_r} \left( \hat{g}^r(x_w), c_w^r{}' \right) - \nabla_{\hat{g}} L_{C_r} \left( \hat{g}^r(x_w), c_w^r \right) \right).$$

5: **Use EK-FAC method in algorithm 5 to accelerate iHVP problem for $\hat{f}$ and obtain $H_{\hat{f}}^{-1}$ Compute $A$ as:**

$$A = H_{\hat{f}}^{-1} \cdot \sum_{i \in [n]-G} \nabla_{\hat{f}} L_Y \left( \hat{f}(\hat{g}(x_i)), y_i \right)$$

Obtain $\bar{f}$ as

$$\bar{f} = \hat{f} + A$$

6: **Use EK-FAC method in algorithm 5 to accelerate iHVP problem for $\bar{f}$ and obtain $H_{\bar{f}}^{-1}$ Compute $B'$ as**

$$B' = -H_{\bar{f}}^{-1} \cdot \sum_{i \in [n]-G} \nabla_{\hat{f}} \left( L_Y(\bar{f}(\tilde{g}(x_i)), y_i) - L_Y(\bar{f}(\hat{g}(x_i)), y_i) \right)$$

Update the label predictor $\tilde{f}$ as: $\tilde{f} = \hat{f} + A + B'$.
7: **Return:** $\tilde{f}, \tilde{g}$.

---

# H. Additional Experiments

## H.1. Experimental Setting

**Methodology for Processing CUB Dataset** For CUB dataset, we follow the setting in (Koh et al., 2020). We aggregate instance-level concept annotations into class-level concepts via majority voting: e.g., if more than 50% of crows have black wings in the data, then we set all crows to have black wings.

**RMIA score.** The RMIA score is computed as:

$$LR_\theta(x, z) \approx \frac{\Pr(f_\theta(x)|\mathcal{N}(\mu_{x,\bar{z}}(x), \sigma_{x,\bar{z}}^2(x)))}{\Pr(f_\theta(x)|\mathcal{N}(\mu_{\bar{x},z}(x), \sigma_{\bar{x},z}^2(x)))} \times \frac{\Pr(f_\theta(z)|\mathcal{N}(\mu_{x,\bar{z}}(z), \sigma_{x,\bar{z}}^2(z)))}{\Pr(f_\theta(z)|\mathcal{N}(\mu_{\bar{x},z}(z), \sigma_{\bar{x},z}^2(z)))}$$

where $f_\theta(x)$ represents the model's output (logits) for the data point $x$, $\mathcal{N}(\mu, \sigma^2)$ denotes a Gaussian distribution with mean $\mu$ and variance $\sigma^2$, $\mu_{x,\bar{z}}(x)$ is the mean of the model's outputs for $x$ under the assumption that $x$ belongs to the training set, and $\sigma_{x,\bar{z}}^2(x)$ is the variance of the model's outputs for $x$. The likelihoods $\Pr(f_\theta(x)|\mathcal{N})$ represent the probability that the model's output $f_\theta(x)$ follows the Gaussian distribution parameterized by $\mu$ and $\sigma^2$, under the two different hypotheses: $x$ being a member of the training set versus not being a member.

## H.2. Improvement via Harmful Data Removal

We conducted additional experiments on CUB datasets with synthetically introduced noisy concepts or labels. Firstly, we introduce noises under three levels. At the concept level, we choose 10% of the concepts and flip these concept labels for a portion of the data. At the data level, we choose 10% of the data and flip their labels. At the concept-label level, we choose 10% of the total concepts and flip them. Then, we conduct the following experiments.

We introduce noises into the three levels and train the model. After that, we remove the noise and obtain the retrained model, which is the ground truth(gt) of this harmful data removal task. In contrast, we use ECBM to remove the harmful data.

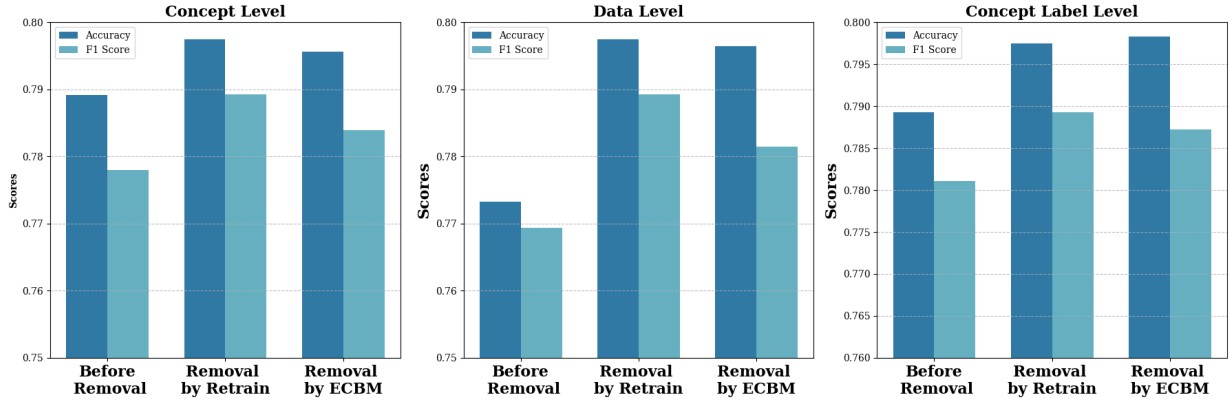

Figure 5: Model performance after the removal of harmful data.

From Figure 5, it can be observed that the model performance improves across all three settings after noise removal and subsequent retraining or ECBM editing. This confirms that the performance of ECBM is nearly equivalent to retraining in various experimental scenarios, further providing evidence of the robustness of our method.

## H.3. Periodic Editing Performance

ECBM can perform periodic editing. To evaluate the multiple editing performance of ECBM, we conduct the following experiments. Firstly, we introduce noises under three levels. At the concept level, we choose 10% of the concepts and flip these concept labels for a portion of the data. At the data level, we choose 10% of the data and flip their labels. At the concept-label level, we choose 10% of the total concepts and flip them. Then, we conduct the following experiments.

At the concept level, we first remove 1% of the concepts, then retrain or use ECBM to edit and repeat. In the data level, we first remove 1% of the data, then retrain or use ECBM to edit. At the concept label level, we first remove one concept label from 1% of the data, then retrain or use ECBM to edit. Note that when removing the next 1% of the concepts, ECBM edits the model based on the last editing result. The results at each level are shown in Figure 6, 7 and 8.

From the above three levels, we can find that with the mislabeled information removed, the retrained model achieves better performance in both accuracy and F1 score than the initial model. Furthermore, the performance of the ECBM-edited model is similar to that of the retrained model, even after 10 rounds of editing, which demonstrates the ability of our ECBM method to handle multiple edits.

## H.4. More Visualization Results and Explanation

**Visualization.** Since CBM is an explainable model, we aim to evaluate the interpretability of our ECBM (compared to the retraining). We will present some visualization results for the concept-level edit. Figure 9 presents the top 10 most influential concepts and their corresponding predicted concept labels obtained by our ECBM and the retrain method after randomly deleting concepts for the CUB dataset. (Detailed explanation can be found in Appendix H.4.1.) Our ECBM can provide explanations for which concepts are crucial and how they assist the prediction. Specifically, among the top 10 most important concepts in the ground truth (retraining), ECBM can accurately recognize 9 within them. For instance, we correctly identify "has_upperparts_color::orange", "has_upper_tail_color::red", and "has_breast_color::black" as some of the most important concepts when predicting categories. Additional visualization results under data level and concept-label

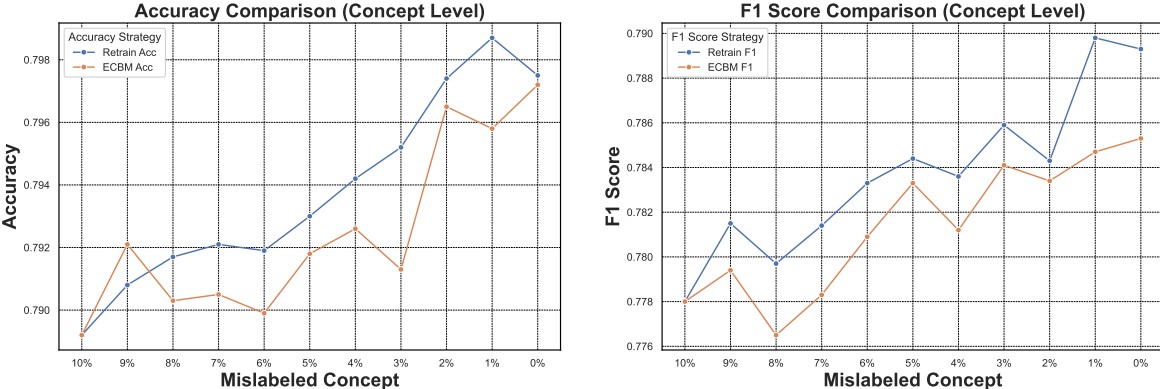

(a) The accuracy of the edited model compared with retrained. (b) The F1 score of the edited model compared with retrained.

Figure 6: Accuracy and F1 score difference of the edited model compared with retrained at concept level.

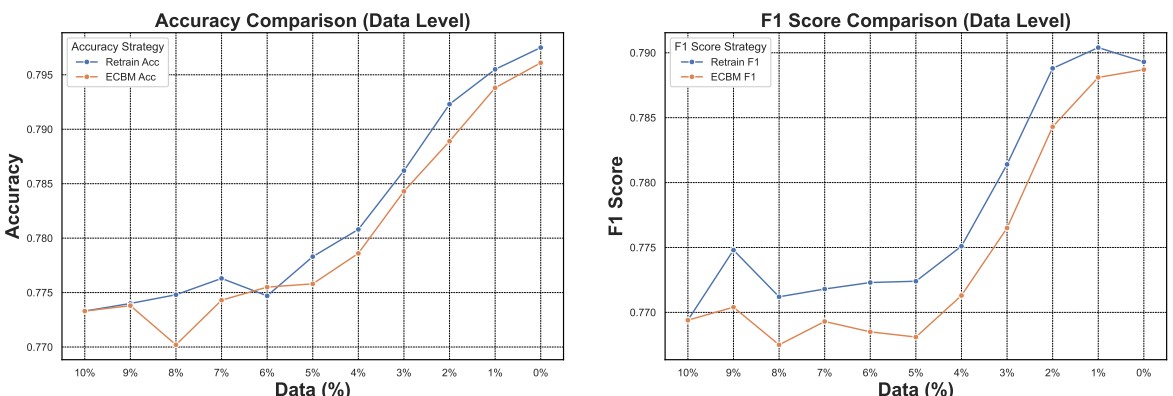

(a) The accuracy of the edited model compared with retrained. (b) he F1 score of the edited model compared with retrained.

Figure 7: Accuracy and F1 score difference of the edited model compared with retrained at data level.

level on OAI and CUB datasets are included in Appendix H.4.2.

### H.4.1. EXPLANATION FOR VISUALIZATION RESULTS

At the concept level, we remove each concept one at a time, retrain the CBM, and subsequently evaluate the model performance. We rank the concepts in descending order based on the model performance loss. Concepts that, when removed, cause significant changes in model performance are considered influential concepts. The top 10 concepts are shown in the retrain column as illustrated in Figure 9. In contrast, we use our ECBM method instead of the retrain method, as outlined in Algorithm 7, and the top 10 concepts are shown in the ECBM column of Figure 9.

To help readers connect the top 10 influential concepts with the input image, we provide visualizations of the data and list the concept labels corresponding to the top 10 influential concepts, which are shown in Figure 9,10, 11.

For the other two levels and for additional datasets, we also conduct a similar procedure, and the corresponding visualization results are presented in Figure 12, 13, 14, 15, and 16.

### H.4.2. VISUALIZATION RESULTS

We provide our additional visualization results in Figure 10, 11, 12, 13, 14, 15, and 16.

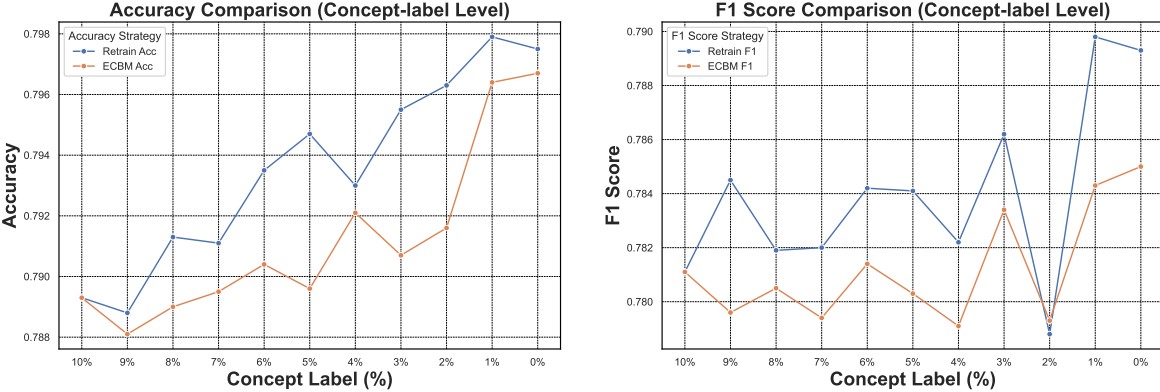

(a) The accuracy of the edited model compared with retrained. (b) The F1 score of the edited model compared with retrained.

Figure 8: Accuracy and F1 score difference of the edited model compared with retrained at concept-label level.

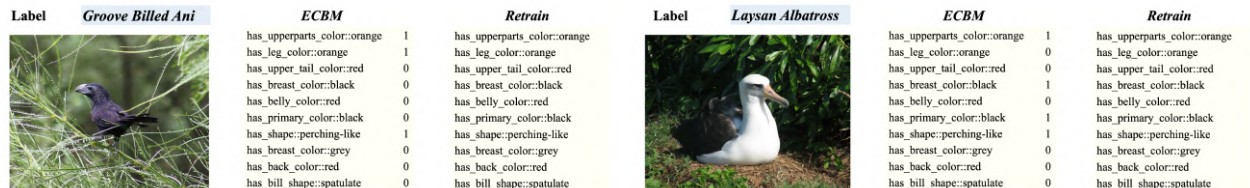

Figure 9: Visualization of the Top 10 Most Influential Concepts for CBM(Identified by ECBM or Retrain) Highlighted on an Extracted Image.

## I. More Related Work

**Influence Function.** The influence function, initially a staple in robust statistics (Cook, 2000; Cook & Weisberg, 1980), has seen extensive adoption within machine learning since (Koh & Liang, 2017) introduced it to the field. Its versatility spans various applications, including detecting mislabeled data, interpreting models, addressing model bias, and facilitating machine unlearning tasks. Notable works in machine unlearning encompass unlearning features and labels (Warnecke et al., 2023), minimax unlearning (Liu et al., 2024), forgetting a subset of image data for training deep neural networks (Golatkar et al., 2020a; 2021), graph unlearning involving nodes, edges, and features. Recent advancements, such as the LiSSA method (Agarwal et al., 2017; Kwon et al., 2023) and kNN-based techniques (Guo et al., 2021), have been proposed to enhance computational efficiency. Besides, various studies have applied influence functions to interpret models across different domains, including natural language processing (Han et al., 2020) and image classification (Basu et al., 2021), while also addressing biases in classification models (Wang et al., 2019), word embeddings (Brunet et al., 2019), and finetuned models (Chen et al., 2020). Despite numerous studies on influence functions, we are the first to utilize them to construct the editable CBM. Moreover, compared to traditional neural networks, CBMs are more complicated in their influence function. Because we only need to change the predicted output in the traditional influence function. While in CBMs, we should first remove the true concept, then we need to approximate the predicted concept in order to approximate the output. Bridging the gap between the true and predicted concepts poses a significant theoretical challenge in our proof.

**Model Unlearning.** Model unlearning has gained significant attention in recent years, with various methods (Bourtoule et al., 2021; Brophy & Lowd, 2021; Cao & Yang, 2015; Chen et al., 2022a;b) proposed to efficiently remove the influence of certain data from trained machine learning models. Existing approaches can be broadly categorized into exact and approximate unlearning methods. Exact unlearning methods aim to replicate the results of retraining by selectively updating only a portion of the dataset, thereby avoiding the computational expense of retraining on the entire dataset (Sekhari et al., 2021; Chowdhury et al., 2024). Approximate unlearning methods, on the other hand, seek to adjust model parameters to approximately satisfy the optimality condition of the objective function on the remaining data (Golatkar et al., 2020a; Guo et al., 2019; Izzo et al., 2021). These methods are further divided into three subcategories: (1) Newton step-based updates that leverage Hessian-related terms [22, 26, 31, 34, 40, 43, 49], often incorporating Gaussian noise to mitigate

residual data influence. To reduce computational costs, some works approximate the Hessian using the Fisher information matrix (Golatkar et al., 2020a) or small Hessian blocks (Mehta et al., 2022). (2) Neural tangent kernel (NTK)-based unlearning approximates training as a linear process, either by treating it as a single linear change (Golatkar et al., 2020b). (3) SGD path tracking methods, such as DeltaGrad (Wu et al., 2020) and unrollSGD (Thudi et al., 2022), reverse the optimization trajectory of stochastic gradient descent during training. Despite their advancements, these methods fail to handle the special architecture of CBMs. Moreover, given the high cost of obtaining data, we sometimes prefer to correct the data rather than remove it, which model unlearning is unable to achieve.

## J. Limitations and Broader Impacts

It is important to acknowledge that the ECBM approach is essentially an approximation of the model that would be obtained by retraining with the edited data. However, results indicate that this approximation is effective in real-world applications.

Concept Bottleneck Models (CBMs) have garnered much attention for their ability to elucidate the prediction process through a human-understandable concept layer. However, most previous studies focused on cases where the data, including concepts, are clean. In many scenarios, we always need to remove/insert some training data or new concepts from trained CBMs due to different reasons, such as data mislabeling, spurious concepts, and concept annotation errors. Thus, the challenge of deriving efficient editable CBMs without retraining from scratch persists, particularly in large-scale applications. To address these challenges, we propose Editable Concept Bottleneck Models (ECBMs). Specifically, ECBMs support three different levels of data removal: concept-label-level, concept-level, and data-level. ECBMs enjoy mathematically rigorous closed-form approximations derived from influence functions that obviate the need for re-training. Experimental results demonstrate the efficiency and effectiveness of our ECBMs, affirming their adaptability within the realm of CBMs. Our ECBM can be an interactive model with doctors in the real world, which is an editable explanation tool.

| Label | Black Footed Albatross | ECBM | | Retrain |
|---|---|---|---|---|
| | | has_upperparts_color::orange | 0 | has_upperparts_color::orange |
| | | has_leg_color::orange | 0 | has_leg_color::orange |
| | | has_upper_tail_color::red | 0 | has_upper_tail_color::red |
| | | has_breast_color::black | 0 | has_breast_color::black |
| | | has_belly_color::red | 0 | has_belly_color::red |
| | | has_primary_color::black | 0 | has_primary_color::black |
| | | has_shape::perching-like | 1 | has_shape::perching-like |
| | | has_breast_color::grey | 0 | has_breast_color::grey |
| | | has_back_color::red | 0 | has_back_color::red |
| | | has_bill_shape::spatulate | 0 | has_bill_shape::spatulate |

| Label | Crested Auklet | ECBM | | Retrain |
|---|---|---|---|---|
| | | has_upperparts_color::orange | 1 | has_upperparts_color::orange |
| | | has_leg_color::orange | 0 | has_leg_color::orange |
| | | has_upper_tail_color::red | 0 | has_upper_tail_color::red |
| | | has_breast_color::black | 0 | has_breast_color::black |
| | | has_belly_color::red | 0 | has_belly_color::red |
| | | has_primary_color::black | 0 | has_primary_color::black |
| | | has_shape::perching-like | 1 | has_shape::perching-like |
| | | has_breast_color::grey | 0 | has_breast_color::grey |
| | | has_back_color::red | 0 | has_back_color::red |
| | | has_bill_shape::spatulate | 0 | has_bill_shape::spatulate |

| Label | Least Auklet | ECBM | | Retrain |
|---|---|---|---|---|
| | | has_upperparts_color::orange | 1 | has_upperparts_color::orange |
| | | has_leg_color::orange | 0 | has_leg_color::orange |
| | | has_upper_tail_color::red | 0 | has_upper_tail_color::red |
| | | has_breast_color::black | 1 | has_breast_color::black |
| | | has_belly_color::red | 0 | has_belly_color::red |
| | | has_primary_color::black | 0 | has_primary_color::black |
| | | has_shape::perching-like | 0 | has_shape::perching-like |
| | | has_breast_color::grey | 0 | has_breast_color::grey |
| | | has_back_color::red | 0 | has_back_color::red |
| | | has_bill_shape::spatulate | 0 | has_bill_shape::spatulate |

| Label | Rhinoceros Auklet | ECBM | | Retrain |
|---|---|---|---|---|
| | | has_upperparts_color::orange | 1 | has_upperparts_color::orange |
| | | has_leg_color::orange | 0 | has_leg_color::orange |
| | | has_upper_tail_color::red | 0 | has_upper_tail_color::red |
| | | has_breast_color::black | 0 | has_breast_color::black |
| | | has_belly_color::red | 0 | has_belly_color::red |
| | | has_primary_color::black | 0 | has_primary_color::black |
| | | has_shape::perching-like | 1 | has_shape::perching-like |
| | | has_breast_color::grey | 0 | has_breast_color::grey |
| | | has_back_color::red | 0 | has_back_color::red |
| | | has_bill_shape::spatulate | 0 | has_bill_shape::spatulate |

Figure 10: Visualization of the top-10 most influential concepts for different classes in CUB.

| Label | Brewer Blackbird | | ECBM | | Retrain |
|---|---|---|---|---|---|
| | | has_upperparts_color::orange | 1 | has_upperparts_color::orange | |
| | | has_leg_color::orange | 1 | has_leg_color::orange | |
| | | has_upper_tail_color::red | 0 | has_upper_tail_color::red | |
| | | has_breast_color::black | 0 | has_breast_color::black | |
| | | has_belly_color::red | 0 | has_belly_color::red | |
| | | has_primary_color::black | 0 | has_primary_color::black | |
| | | has_shape::perching-like | 1 | has_shape::perching-like | |
| | | has_breast_color::grey | 0 | has_breast_color::grey | |
| | | has_back_color::red | 0 | has_back_color::red | |
| | | has_bill_shape::spatulate | 1 | has_bill_shape::spatulate | |

| Label | Red Winged Blackbird | | ECBM | | Retrain |
|---|---|---|---|---|---|
| | | has_upperparts_color::orange | 1 | has_upperparts_color::orange | |
| | | has_leg_color::orange | 1 | has_leg_color::orange | |
| | | has_upper_tail_color::red | 0 | has_upper_tail_color::red | |
| | | has_breast_color::black | 0 | has_breast_color::black | |
| | | has_belly_color::red | 0 | has_belly_color::red | |
| | | has_primary_color::black | 0 | has_primary_color::black | |
| | | has_shape::perching-like | 1 | has_shape::perching-like | |
| | | has_breast_color::grey | 0 | has_breast_color::grey | |
| | | has_back_color::red | 0 | has_back_color::red | |
| | | has_bill_shape::spatulate | 0 | has_bill_shape::spatulate | |

| Label | Rusty Blackbird | | ECBM | | Retrain |
|---|---|---|---|---|---|
| | | has_upperparts_color::orange | 1 | has_upperparts_color::orange | |
| | | has_leg_color::orange | 1 | has_leg_color::orange | |
| | | has_upper_tail_color::red | 0 | has_upper_tail_color::red | |
| | | has_breast_color::black | 0 | has_breast_color::black | |
| | | has_belly_color::red | 0 | has_belly_color::red | |
| | | has_primary_color::black | 0 | has_primary_color::black | |
| | | has_shape::perching-like | 0 | has_shape::perching-like | |
| | | has_breast_color::grey | 0 | has_breast_color::grey | |
| | | has_back_color::red | 0 | has_back_color::red | |
| | | has_bill_shape::spatulate | 1 | has_bill_shape::spatulate | |

| Label | Yellow Headed Blackbird | | ECBM | | Retrain |
|---|---|---|---|---|---|
| | | has_upperparts_color::orange | 1 | has_upperparts_color::orange | |
| | | has_leg_color::orange | 1 | has_leg_color::orange | |
| | | has_upper_tail_color::red | 0 | has_upper_tail_color::red | |
| | | has_breast_color::black | 0 | has_breast_color::black | |
| | | has_belly_color::red | 0 | has_belly_color::red | |
| | | has_primary_color::black | 0 | has_primary_color::black | |
| | | has_shape::perching-like | 1 | has_shape::perching-like | |
| | | has_breast_color::grey | 1 | has_breast_color::grey | |
| | | has_back_color::red | 0 | has_back_color::red | |
| | | has_bill_shape::spatulate | 1 | has_bill_shape::spatulate | |

Figure 11: Visualization of the top-10 most influential concepts for different classes in CUB.

**Label**   *Pine_Warbler*

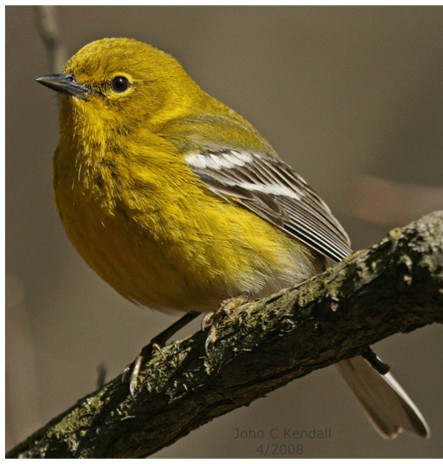

| Concept Name | Influence Score |
| --- | --- |
| has_belly_color::grey | 0.037985 |
| has_underparts_color::grey | 0.037982 |
| has_breast_color::grey | 0.03798 |
| has_bill_length::longer_than_head | 0.037946 |
| has_throat_color::grey | 0.037901 |
| has_back_color::grey | 0.037894 |
| has_crown_color::grey | 0.037868 |
| has_primary_color::grey | 0.037866 |
| has_shape::swallow-like | 0.037811 |
| has_nape_color::grey | 0.037763 |

**Label**   *Bewick_Wren*

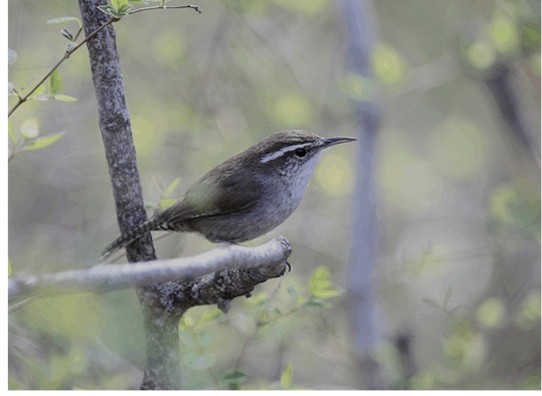

| Concept Name | Influence Score |
| --- | --- |
| has_wing_color::blue | 0.04231 |
| has_crown_color::blue | 0.042196 |
| has_forehead_color::blue | 0.042055 |
| has_bill_shape::spatulate | 0.041994 |
| has_under_tail_color::blue | 0.041622 |
| has_head_pattern::unique_pattern | 0.041412 |
| has_upper_tail_color::blue | 0.041179 |
| has_nape_color::blue | 0.040844 |
| has_shape::swallow-like | 0.040686 |
| has_tail_pattern::spotted | 0.040507 |

**Label**   *Song_Sparrow*

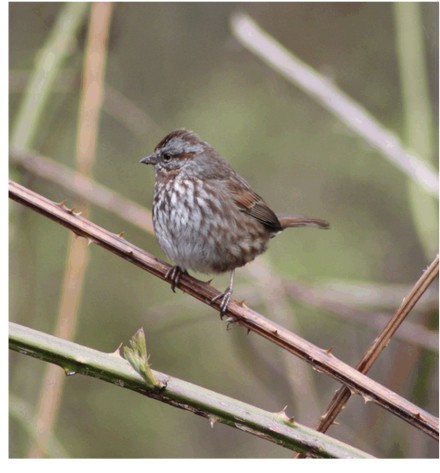

| Concept Name | Influence Score |
| --- | --- |
| has_upperparts_color::blue | 0.036309 |
| has_wing_color::blue | 0.036304 |
| has_primary_color::blue | 0.036271 |
| has_back_color::blue | 0.036261 |
| has_crown_color::blue | 0.036219 |
| has_breast_color::blue | 0.036178 |
| has_underparts_color::blue | 0.03616 |
| has_nape_color::blue | 0.036104 |
| has_upper_tail_color::blue | 0.036083 |
| has_forehead_color::blue | 0.035959 |

Figure 12: Visualization of the most influential concept label related to different data in CUB.

**Label**      *Brewer_Blackbird*

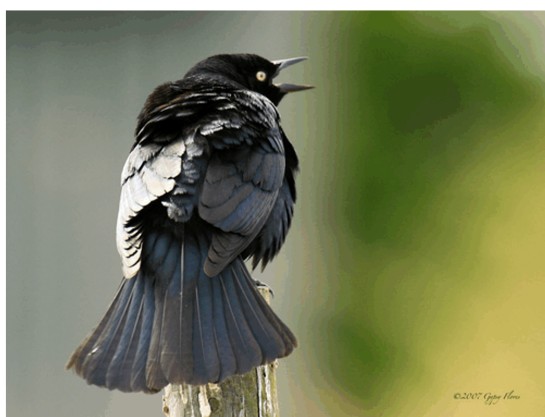

| Concept Name | Influence Score |
| --- | --- |
| has_forehead_color::orange | 0.042692 |
| has_breast_color::orange | 0.042647 |
| has_crown_color::orange | 0.042646 |
| has_throat_color::orange | 0.042588 |
| has_upper_tail_color::orange | 0.042574 |
| has_upperparts_color::orange | 0.042569 |
| has_primary_color::orange | 0.042546 |
| has_back_color::orange | 0.042543 |
| has_nape_color::orange | 0.042484 |
| has_belly_color::orange | 0.042463 |

**Label**      *Frigatebird*

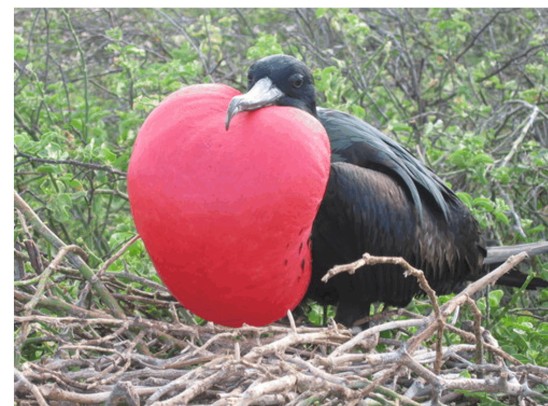

| Concept Name | Influence Score |
| --- | --- |
| has_tail_pattern::multi-colored | 0.053243 |
| has_bill_shape::needle | 0.053006 |
| has_back_pattern::multi-colored | 0.052768 |
| has_primary_color::orange | 0.052117 |
| has_underparts_color::orange | 0.051954 |
| has_under_tail_color::orange | 0.051617 |
| has_crown_color::buff | 0.050712 |
| has_head_pattern::eyering | 0.049705 |
| has_shape::perching-like | 0.049511 |
| has_forehead_color::orange | 0.049194 |

**Label**      *Philadelphia_Vireo*

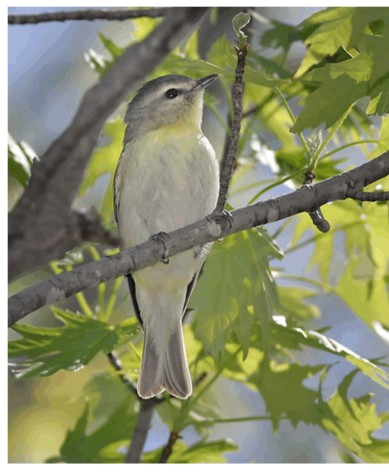

| Concept Name | Influence Score |
| --- | --- |
| has_eye_color::orange | 0.04754 |
| has_shape::swallow-like | 0.047265 |
| has_bill_shape::spatulate | 0.047145 |
| has_crown_color::rufous | 0.047015 |
| has_tail_pattern::multi-colored | 0.046809 |
| has_forehead_color::rufous | 0.046604 |
| has_back_color::rufous | 0.046068 |
| has_size::large_(16_-_32_in) | 0.045287 |
| has_nape_color::rufous | 0.044857 |
| has_upperparts_color::rufous | 0.043474 |

Figure 13: Visualization of the most influential concept label related to different data in CUB.

**Label**   *Seaside_Sparrow*

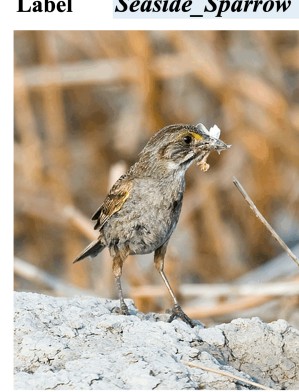

| Concept Name | Influence Score |
|---|---|
| has_bill_color::rufous | 0.083579 |
| has_shape::swallow-like | 0.08343 |
| has_nape_color::rufous | 0.08309 |
| has_tail_pattern::multi-colored | 0.081434 |
| has_bill_length::longer_than_head | 0.079826 |
| has_size::large_(16_-_32_in) | 0.078111 |
| has_belly_color::buff | 0.069643 |
| has_back_color::blue | 0.067222 |
| has_eye_color::orange | 0.063228 |
| has_upperparts_color::blue | 0.057842 |

**Label**   *Vesper_Sparrow*

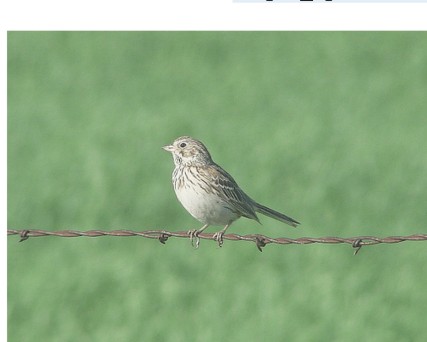

| Concept Name | Influence Score |
|---|---|
| has_wing_color::red | 0.035092 |
| has_back_pattern::spotted | 0.035083 |
| has_bill_color::red | 0.035072 |
| has_breast_pattern::spotted | 0.035042 |
| has_bill_shape::all-purpose | 0.03491 |
| has_upper_tail_color::blue | 0.034787 |
| has_wing_pattern::spotted | 0.034754 |
| has_back_color::red | 0.034625 |
| has_nape_color::red | 0.034548 |
| has_throat_color::black | 0.034427 |

Figure 14: Visualization of the most influential concept label related to different data in CUB.

**Label** *Vermilion_Flycatcher*

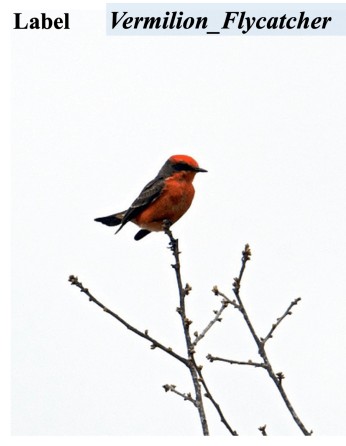

| Concept Name | Influence Score |
|---|---|
| has_bill_length::longer_than_head | 0.082524 |
| has_size::large_(16_-_32_in) | 0.082308 |
| has_tail_pattern::multi-colored | 0.079543 |
| has_leg_color::orange | 0.079385 |
| has_shape::swallow-like | 0.078894 |
| has_back_pattern::multi-colored | 0.074584 |
| has_underparts_color::buff | 0.073978 |
| has_bill_shape::all-purpose | 0.063468 |
| has_tail_shape::rounded_tail | 0.059044 |
| has_shape::perching-like | 0.053268 |

**Label** *Fox_Sparrow*

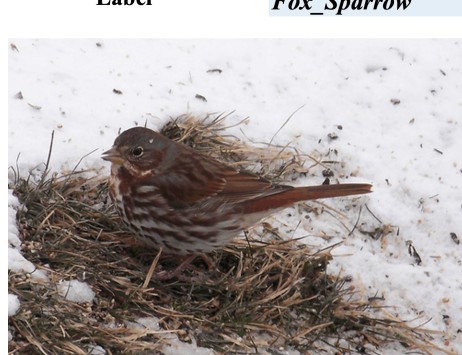

| Concept Name | Influence Score |
|---|---|
| has_breast_color::blue | 0.041734 |
| has_underparts_color::blue | 0.04173 |
| has_belly_color::blue | 0.041652 |
| has_upper_tail_color::blue | 0.041646 |
| has_breast_pattern::spotted | 0.041567 |
| has_crown_color::blue | 0.041521 |
| has_nape_color::blue | 0.041439 |
| has_back_color::blue | 0.041307 |
| has_forehead_color::blue | 0.041287 |
| has_under_tail_color::blue | 0.041208 |

Figure 15: Visualization of the most influential concept label related to different data in CUB.

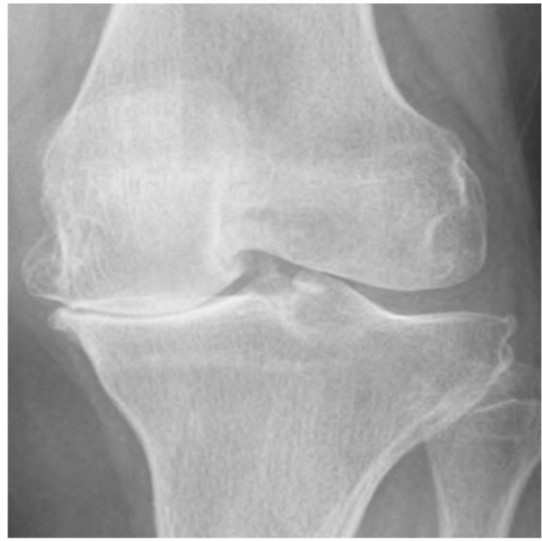

| Concept Name | Influence Score |
|---|---|
| Joint space narrowing | 0.3358 |
| Joint space narrowing lateral | 0.1622 |
| Sclerosis femur medial | 0.1161 |
| Sclerosis femur lateral | 0.0993 |
| Sclerosis tibia lateral | 0.0878 |
| Osteophytes tibia medial | 0.0724 |
| Osteophytes femur lateral | 0.047 |
| Osteophytes tibia lateral | 0.031 |
| Osteophytes femur medial | 0.0271 |
| Sclerosis tibia medial | 0.0213 |

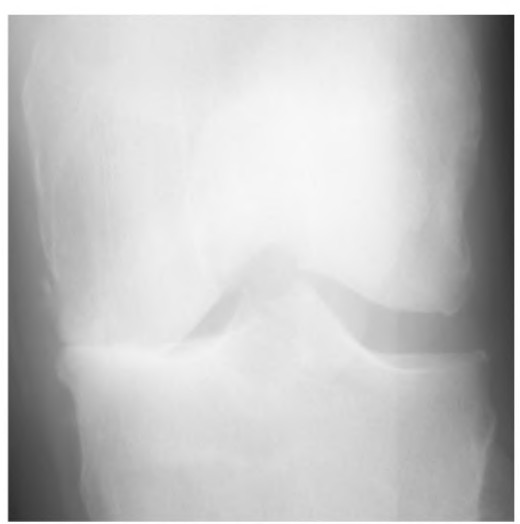

| Concept Name | Influence Score |
|---|---|
| Joint space narrowing | 0.3506 |
| Osteophytes femur medial | 0.1698 |
| Osteophytes tibia medial | 0.0991 |
| Osteophytes tibia lateral | 0.0824 |
| Joint space narrowing lateral | 0.0728 |
| Sclerosis tibia lateral | 0.0674 |
| Osteophytes femur lateral | 0.0595 |
| Sclerosis femur lateral | 0.0467 |
| Sclerosis femur medial | 0.0272 |
| Sclerosis tibia medial | 0.0245 |

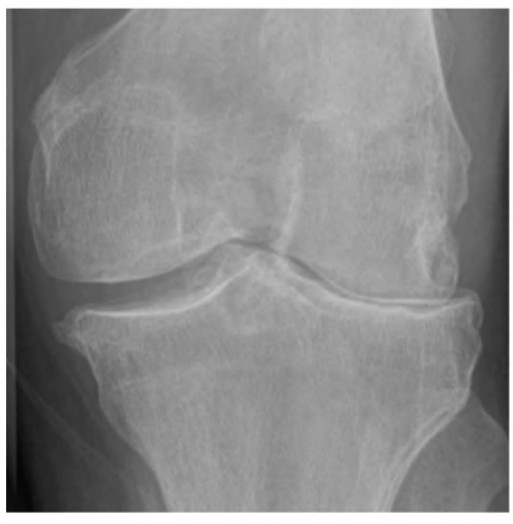

| Concept Name | Influence Score |
|---|---|
| Joint space narrowing | 0.2978 |
| Joint space narrowing lateral | 0.2018 |
| Osteophytes femur lateral | 0.1247 |
| Sclerosis tibia lateral | 0.0949 |
| Sclerosis tibia medial | 0.0892 |
| Osteophytes femur medial | 0.055 |
| Sclerosis femur medial | 0.0463 |
| Osteophytes tibia medial | 0.0387 |
| Sclerosis femur lateral | 0.0321 |
| Osteophytes tibia lateral | 0.0195 |

Figure 16: Visualization of the most influential concept label related to different data in OAI.

