# OpenReview forum: "Editable Concept Bottleneck Models"
_ICML.cc/2025/Conference — ICML 2025 poster_

### Official Review · Reviewer_iqcY · 2025-03-09

**Overall Recommendation:** 3

**Summary:**

Editable CBMs provide the ability to edit a trained CBM to account for issues in annotation errors, concept set changes, and problems with specific data points. That is done with the help of influence functions that approximate the model.

**Claims And Evidence:**

The claims made are largely clear and supported by evidence. There is one exception, which is that in the motivation, the concept-level editing is motivated by the fact that oftentimes one wants to *add* concepts to the concept set post-hoc. As far as I understand this work, only the removal of a concept is possible, but not the addition thereof. Thus, I suggest changing this framing.

**Essential References Not Discussed:**

I recommend moving the related work section on Machine Unlearning into the main text. It would help framing the paper correctly and understanding the contribution of this work.

**Experimental Designs Or Analyses:**

The empirical evaluation is sound and thorough, i.e. the authors measure the traits that ECBMs are supposed to fulfill. It is impressive that ECBMs' performance is close to "Retrain", which functions as an Oracle.

**Methods And Evaluation Criteria:**

The proposed method is well-motivated and sound. The datasets used, while simplistic, are established benchmarks in the field.

**Other Comments Or Suggestions:**

The abbreviation ECBM is already used by the published energy-based concept bottleneck models

**Other Strengths And Weaknesses:**

I am not well-read in the domain of machine unlearning, but I am very sure that they are highly relevant to this work. That is, my intuition tells me that this work is essentially CBM + Machine Unlearning, and I am unsure how much novelty there is with respect to the existing methods in the field of Machine Unlearning.

**Questions For Authors:**

My main reasons for not giving a higher score are
1. that model editing in the context of CBMs is not a big problem in my opinion. I think existing methods such as [1] could easily be adapted to quickly retrain with the adapted dataset. As such, the significance is limited in my opinion.

2. I am unsure of the novelty with respect to existing methods in Machine Unlearning, as the usage of Influence functions for this purpose appears quite standard, and I would imagine that the editing of encoder and predictor of the CBM can be "mapped" to some existing tasks in that field.

While these are not questions, I invite the authors to comment on these opinions.

3. Can the authors provide code for reproducibility?


[1] Laguna, Sonia, et al. "Beyond Concept Bottleneck Models: How to Make Black Boxes Intervenable?." The Thirty-eighth Annual Conference on Neural Information Processing Systems.

**Relation To Broader Scientific Literature:**

This work contributes to the literature in CBMs. To the best of my knowledge, in the context of CBMs, model editing of this sort has not been explored. Personally, I am not convinced that model editing is such an important task in CBMs, as their concept bottleneck prevents them anyways from being too large to be retrained.

**Theoretical Claims:**

I did not check the proofs in the Appendix, however, the equations in the main text intuitively do not contradict my intuition of how they should be.

---

> ### Author Rebuttal · Authors · 2025-04-01
>
> -*Response to Claims And Evidence*
>
> We respectfully disagree with your opinion. The primary goal of CBM is to explicitly decompose the model's intermediate representation into a set of interpretable concepts, typically predefined by domain experts or specific task requirements before training. However, when task requirements change, new concepts may need to be added to the existing concept set, making such scenarios possible in practice. Most CBM methods, however, fix the concept set after training, with the model's structure and parameters tightly bound to these concepts. As a result, directly adding new concepts post-training is challenging and often requires retraining the model.
>
> -*Response to Relation To Broader Scientific Literature*
>
> Thank you for sharing your perspective. I understand your concerns and would like to highlight why CBM editing is important:
>
> 1. Correcting Labeling Errors: In fields like healthcare, training data is valuable, and discarding mislabeled data isn’t ideal. CBM editing allows targeted corrections without costly retraining.
>
> 2. Updating Concepts: During deployment, missing or irrelevant concepts may arise. CBM editing enables efficient updates.
>
> 3. Privacy Constraints: CBM editing allows data removal requests to be handled accurately without full model retraining.
>
> As you mentioned, retraining CBMs can be computationally expensive due to their bottleneck structure. This makes CBM editing a practical alternative.
>
> In summary, model editing effectively addresses practical challenges such as correcting data errors, ensuring privacy, and adapting to new requirements, all without the high cost of retraining. These factors highlight the importance of CBM editing in real-world applications, as acknowledged by the other reviewers.
>
>
> -*Response to Essential References Not Discussed, Other Strengths And Weaknesses and Questions For Authors 2*
>
> Thank you for your thoughtful perspective. While we acknowledge that editing CBM within the context of privacy constraints does share some similarities with machine unlearning, this work is not solely focused on CBM and machine unlearning. Our primary objective is to enable the flexible editing of CBM across three levels: data, concept, and concept-label. This process goes beyond unlearning to encompass modification and optimization as well. Ultimately, the core goal of this work is to enhance the applicability and adaptability of CBM, rather than to design a new machine unlearning algorithm specifically for CBM.
>
> -*Response to Questions For Authors 1*
>
> Thanks for your information. [1] proposes a method for intervening on the intermediate representations of neural networks, but its network architecture is not based on CBMs, thereby diverging from the goal of editing the original CBM framework.
>
> In contrast, our work focuses on the specific application scenarios of CBMs. By addressing model editing at the levels of data, concepts, and concept labels, we systematically formulate this problem mathematically for the first time and develop editing algorithms with theoretical guarantees. Our research fills a critical gap in this area.
>
>
> -*Response to Questions For Authors 3*
>
> Yes. Our code can be found here.
> https://anonymous.4open.science/r/ECBM-4B14
>
>
> Thank you very much for your valuable feedback and recognition of our article. If we have addressed all of your concerns, we kindly ask you to consider giving a higher rating.

---

> > ### Comment · Reviewer_iqcY · 2025-04-03
> >
> > I thank the authors for their response.
> >
> > Claims And Evidence: I think I was misunderstood. I completely agree that adding a concept post-hoc can be a desirable task. What I meant was the following: As far as I understand the proposed method, the method can only remove concepts, not add them. That is, it can not cover this important task.
> > Please let me know if this is not the case.
> >
> > Relation To Broader Scientific Literature: I disagree that CBMs are computationally expensive to retrain.
> >
> > I thank the authors for their response and keep my score as my opinion on the raised points has not been changed.

---

> > > ### Author Response · Authors · 2025-04-04
> > >
> > > Thanks for your feedback.
> > >
> > > -*Reponse to Claims And Evidence*
> > >
> > > We appreciate your clarification. You are absolutely correct; the ECBM method cannot accommodate requests for adding new concepts to the CBM and can only facilitate concept removal. We will ensure that this statement is revised in the camera-ready version. We sincerely appreciate your insights and for bringing this issue to our attention.
> > >
> > >
> > > -*Response to Relation To Broader Scientific Literature*
> > >
> > >
> > > From the results in Table 1, it can be observed that retraining a CBM based on ResNet-18 on the OAI dataset (which consists of approximately 30,000 entries) requires at least 250 minutes. Consequently, the time cost of retraining a CBM is considerable.
> > >
> > > While this time cost may initially seem acceptable, it is important to note that CBMs are typically utilized in dynamic environments, such as those involving frequent data deletion requests for privacy reasons, as well as label or concept corrections. In this context, the demand for retraining CBMs is both present and frequent. This situation significantly limits the effectiveness of CBMs in practical applications.

---

### Official Review · Reviewer_mn5G · 2025-03-11

**Overall Recommendation:** 5

**Summary:**

The paper introduces Editable Concept Bottleneck Models (ECBMs), an extension of Concept Bottleneck Models (CBMs) that allows efficient data and concept removal without full retraining. Using influence functions and Hessian-based approximations, ECBMs support three levels of editability: concept-label, concept, and data-level. Experiments on multiple datasets show that ECBMs achieve similar performance to retraining while being 20-30x faster, making them highly efficient for real-world applications. The work enhances CBMs' adaptability and interpretability but could further explore concept addition and real-world deployment.

**Claims And Evidence:**

Yes, the experiments and analyses support the claims.

**Essential References Not Discussed:**

No.

**Experimental Designs Or Analyses:**

Yes. They design experiments to evaluate the utility and efficiency of ECBMs, comparing them with retraining and CBM-IF. They analyze the impact of different edit settings, concept importance, and data removal. They further validate ECBMs using membership inference attacks. These designs and analyses are valid.

**Methods And Evaluation Criteria:**

Yes, the proposed methods and evaluation criteria are appropriate.

**Other Comments Or Suggestions:**

Please refer to the weaknesses section above.

**Other Strengths And Weaknesses:**

Strengths
- The paper extends CBMs by incorporating editable capabilities, addressing practical issues such as privacy concerns, annotation errors, and dataset corrections. This is particularly useful for dynamic datasets that require frequent updates.

- The paper evaluates ECBMs on multiple datasets (OAI, CUB, and CelebA), demonstrating that ECBMs achieve near-identical performance to retraining while reducing computation time by up to 30x.

- The paper presents closed-form solutions using influence functions, avoiding costly retraining while maintaining accuracy.

- ECBMs provide an efficient way to remove concept biases and erase data influences, addressing model privacy and fairness concerns.

- The incorporation of EK-FAC further accelerates computation, making ECBMs scalable.



Weaknesses
- While the paper discusses the computational advantages of ECBMs, a more detailed analysis of time and space complexity would strengthen the scalability argument.

- The font size in Figure 1 is too small, making it difficult to read. Given its importance, it would be beneficial to adjust the layout (e.g., expanding it to a two-column format).

- Some mathematical notation could be better explained, particularly for readers unfamiliar with influence functions. A more intuitive explanation or additional background material would improve clarity.

**Questions For Authors:**

- Could the authors provide a more detailed analysis of the time and space complexity of ECBMs? While the empirical results demonstrate efficiency, a formal complexity discussion would further support the claims regarding scalability.
- Some derivations rely on influence functions, which may be unfamiliar to many readers. Could the authors provide a more accessible explanation or an appendix section with a high-level intuition behind these functions?


Overall, I did not find any significant weaknesses in the paper. I may change my score based on the authors’ responses regarding weaknesses and above questions.

**Relation To Broader Scientific Literature:**

The paper extends CBM by introducing efficient editability using influence functions, building on prior CBM research and influence function applications.

**Theoretical Claims:**

The paper provides a closed-form solution for the model approximation of CBMs, which includes theoretical claims. I have checked the correctness of the derivation and did not identify any issues.

---

> ### Author Rebuttal · Authors · 2025-04-01
>
> -*Response to Weaknesses 1*
>
> Thanks for your invaluable advice. We will add this part in the revision. Here, we provide the analysis for algorithm 1.
>
> The time complexity of the algorithm is \( O(n \cdot (m^2 + d^2) + s_e \cdot m^2 + d^3) \), where \( n \) is the number of data points, \( m \) is the dimensionality of \( \hat{g} \), \( d \) is the dimensionality of \( \hat{f} \), and \( s_e \) is the size of the erroneous data set \( S_e \). Computing the Hessian matrices for \( \hat{g} \) and \( \hat{f} \) takes \( O(n \cdot m^2) \) and \( O(n \cdot d^2) \), respectively, while the updates for \( \hat{g} \) and \( \hat{f} \) contribute \( O(s_e \cdot m^2) \) and \( O(n \cdot d^2 + d^3) \). The space complexity is \( O(m^2 + d^2 + n \cdot (m + d)) \), dominated by storing the Hessian matrices and the required gradients across all data points.
>
> -*Response to Weaknesses 2 and 3*
>
> We will modify Figure 1 and improve our notation in the revision.
>
> -*Response to Question 1 and 2*
> Thank you for the valuable suggestion. We will include the time and complexity part in the revision.
>
> We agree that influence function may not be familiar to all readers, and we appreciate the opportunity to make our work more accessible. In response, we propose to include an additional appendix section that provides a high-level, intuitive explanation of influence functions.
>
> Thank you very much for your valuable feedback and recognition of our article. If we have addressed all of your concerns, we kindly ask you to consider giving a higher rating.

---

> > ### Comment · Reviewer_mn5G · 2025-04-06
> >
> > Thanks for your responses. I'm satisfied with the new analysis, which makes it more solid.

---

### Official Review · Reviewer_tX1Q · 2025-03-13

**Overall Recommendation:** 4

**Summary:**

The authors present Editable CBMs, where they consider _editability_ from the lens of retraining CBMs at three different levels: 1) Concept Label-level, i.e. when there's label noise in the concept space, 2) Concept level, i.e. removing spurious concepts from the bottleneck predictions and 3) Data-level, i.e. final label noise. For 1), the authors use influence function to estimate the retrain approximation for concept prediction and label predictor sequentially; 2) A similar strategy is applied except for adding a zero-row for the empty concept in the new model; 3) Additional steps to remove the influence of examples with label-noise. The authors use EK-FAC for the second-order approximations and iHVP based algorithms to speed up the compute.

Results are presented on three datasets and additional (extensive) proofs and details are provided in the Appendix.

**Claims And Evidence:**

1. The claims of editability on the three levels are sufficiently shown in the results and appendix.
2. My problem is with the general presentation of results: while the algorithm and results certainly show that this algorithm provides pretty close results to retraining, the results and the language do not contain the original essence of CBMs which is to provide humans the ability to _intervene_ on the intermediate concepts to _better_ the final prediction. In all of the theory to prove that the modifications made by authors to fit the CBM framework, this seems to have been lost. The authors do not compare how the concept intervention behavior changes / remains same for different levels of intervention - based on the language, it seems like final accuracy is the only focus of the paper.
3. Furthermore, a lot of comparisons with modern CBM architectures are missing - for example, the simple choice of soft concepts vs hard concept choices are not acknowledged.(Mahinpei et al) or a note to the ubiquity / lack thereof of the proposed algorithm.

**Essential References Not Discussed:**

CBMs:
[1] Mahinpei, Anita, et al. "Promises and pitfalls of black-box concept learning models." arXiv preprint arXiv:2106.13314 (2021).
Label noise
[2] Thulasidasan, Sunil, et al. "Combating label noise in deep learning using abstention." arXiv preprint arXiv:1905.10964 (2019).
[3] Rolnick, David, et al. "Deep learning is robust to massive label noise." arXiv preprint arXiv:1705.10694 (2017)
[4] Balloli, Vaibhav, Sara Beery, and Elizabeth Bondi-Kelly. "Are they the same picture? adapting concept bottleneck models for human-AI collaboration in image retrieval." arXiv preprint arXiv:2407.08908 (2024).

**Experimental Designs Or Analyses:**

Refer to point 2 and 3 in claims and evidence

**Methods And Evaluation Criteria:**

The proposed methods and evaluation criteria barring the critique above makes sense.

**Other Comments Or Suggestions:**

1. The authors are encouraged to proof-read grammar in a lot of places (missing oxford commas, unnecessarily lengthy sentences, etc.)
2. Define editability early on in the introduction to better prime the readers on what to expect (see also adaptivity -> adaptability in Section 1)

**Other Strengths And Weaknesses:**

The works is original, has signficant impact and novel.

A key weakness is the focus of results which only focuses on final accuracy while other things that could be shown is test-time intervention performance of retrain vs ECBM, robustness to intervention noise, etc.
Another weakness is the limitation and broader impact section which is poorly written - no real limitation except stating it is an approximation and broader impact vaguely mentions doctors and that's it. It is okay to simply state cost saving instead of vaguely writing "ECBM can be an interactive model with doctors in the real world, which is an editable explanation tool." Regarding limitation, the authors can choose to address how many more modifications are required to other architectures like CEM (Zarlenga et al), etc.

**Questions For Authors:**

The authors are requested to address/clarify all the weaknesses and concerns stated above - with sufficient clarification, I'm willing to raise the score to WA/A.

**Relation To Broader Scientific Literature:**

CBMs are gaining popularity, and fast re-training is important; this work certainly furthers improving adoption.

**Theoretical Claims:**

Not fully (only the main parts)

---

> ### Author Rebuttal · Authors · 2025-04-01
>
> ## Weakness:
>
> -*Response to W2: the authors have only considered sequential setting (probably the joint setting as it gives the best performance)*
>
> We sincerely thank the reviewer for highlighting the importance of the jointly training mode in CBM. We agree that joint training sometimes leads to higher accuracy in both label and concept predictions.
>
> However, model performance is not the sole priority:
> 1. Compared to joint training, sequential training is more robust under limited data conditions.
> 2. Joint training requires balancing concept loss and task loss, which may result in suboptimal performance for both. Sequential training avoids this trade-off.
> 3. The modular architecture of sequential training allows for easy post-hoc interventions.
>
> Given these advantages, we focus on the editable CBM with sequential training in this work. Our goal is to explore model editing, which represents the unique perspective and theoretical contribution of our study. We believe this approach complements, rather than replaces, CBM performance optimization research.
>
> Finally, due to the complexity and workload of designing algorithms for the three editing levels, it is not feasible to analyze both sequential and jointly training methods within a single paper. Therefore, in this work, we focus on developing editing algorithms for sequentially trained CBMs across three levels and provide theoretical guarantees. In fact, editing jointly trained CBMs using influence functions is also achievable and will be considered in our future work.
>
>
>
> -*Response to W3: Theorem 4.4, the authors insert 0 valued rows*
>
> Thank you for your suggestion.
>
> When a concept is removed, the output dimension of the concept predictor $g$ decreases accordingly. To facilitate estimation, we modify $g$ into $g'$ by inserting a zero-parameter row into its final layer. These parameters remain fixed during training and thus stay zero, ensuring that the model's effective parameter space is strictly a subset of the original space.
>
> In Theorem 4.4, we approximate $g'$ using influence functions, assuming $g'$ continues training within the original parameter space. Consequently, the algorithm's implementation, including parameter updates, remains unaffected regardless of whether the inserted rows are set to 0, 1, or any constant.
>
> -*Response to W4: Error bars*
>
> We apologize for the omission and we have addressed it in the camera-ready version.
>
> -*Response to W5: Concept-level metrics*
>
> Thanks for your suggestion. We will add experiments about concept-level metrics. We perform experiments on CUB and test the concept accuracy of ECBM and retrain. The results are listed below.
>
>
>
> |         | ECBM(%) | Retrain(%) |
> | ------- | ---- | ---------- |
> | Concept   |  93.7112    |     95.1705      |
> | Data        |  94.5184    |    95.2801|
> | Concept-label  | 95.0219 | 95.1407       |
>
> *Table A: Concept Accuracy of ECBM and Retrain under Three Levels
>
> The results show that the accuracy of Retrain and ECBM is very close, with differences generally within a small margin. For example, in the Data and Concept-label levels, the accuracy gap is less than 0.5%. The results show that ECBM not only approximates the accuracy of the retrain method's labels on the test set, but also has a similar performance in terms of concept accuracy.
>
>
> -*Response to W6: The related work section*
>
> We will include more related works in the camera-ready version.
>
>
> ## Questions:
>
> -*Response to Q1: What is $R^{d_i}$?*
>
> Here $R^{d_i}$ represents the space of all
> $d_i$-dimensional real vectors.
>
> -*Response to Q2: Why are the concepts in the log?*
>
> This is the definition of cross-entropy, identical to that in the original CBM, as described in the second paragraph of page 15. The activation function used is the sigmoid function. Note that this is distinct from the definition of the loss function.
>
> -*Response to Q3: ff problem*
>
> Thanks for your correction. It should be f.
>
> -*Response to Q4: In lines 167-169 ... Why?*
>
> This is because if we correct the concept and then retrain the CBM, the concept predictor $\hat{g}$ will be updated to $\hat{g}_e$, which differs from the scenario at test time intervention where the concept predictor remains unchanged. This distinction serves as the key motivation for editing CBMs.
>
> -*Response to Q5: Why are no other concept-level metrics utilized?*
>
> See Response to W5 for reference.
>
>
> # Reviewer tX1Q
>
> -*Response to Other Comments Or Suggestions*
>
> Thanks for your comments. We will check the paper, fix all the errors and define editable CBM explicitly in the camera-ready version.

---

> > ### Comment · Reviewer_tX1Q · 2025-04-02
> >
> > I don't see any response to the weaknesses in the rebuttal comment (regarding presenting performance under partial intervention, how this technique changes with choice of architectures, etc.).
> >
> > Edit in response to the new rebuttal:
> >
> > I appreciate the additional details and experiments. While I agree (and mention in my comment earlier) that your technique gives results that are pretty close to retraining (the original intention of the work), I brought up test time interventions because that is not necessarily supported theoretically (that it doesn't effect partial/full interventions), so showing this empirically would make a strong case to the readers and practitioners looking to use this. Similarly, showing that your method is ubiquitous to different CBM-like architectures like CEM, etc., and maybe even newer use cases of CBMs like retrieval [4] would make the contributions here much more appealing and up-to-date (writing something along the lines of what is mentioned in your rebuttal would really improve the presentation of your contributions in my opinion that mention the flexibility of your contributions and where the readers should go to adapt to these architectures / use cases)
> >
> > I really hope the new numbers, experiments (both partial and full interventions) and suggestions are taken into account for the camera-ready and therefore, I'm increasing my score accordingly.
> >
> > [4] Balloli, Vaibhav, Sara Beery, and Elizabeth Bondi-Kelly. "Are they the same picture? adapting concept bottleneck models for human-AI collaboration in image retrieval." arXiv preprint arXiv:2407.08908 (2024).

---

> > > ### Author Response · Authors · 2025-04-02
> > >
> > > Dear Reviewer,
> > >
> > > I hope this message finds you well. I sincerely apologize for any inconvenience caused by the oversight in our previous rebuttal. During the copy and paste process, we inadvertently omitted the majority of the text we intended to provide to you for the rebuttal phase.
> > >
> > > To rectify this, I am including the correct rebuttal content below. Thank you very much for your understanding and patience.
> > >
> > > -*Response to Claims And Evidence 2: test-time intervene*
> > >
> > > Thank you for your insightful feedback on this paper. We fully agree that the core value of CBMs lies in enabling human intervention on intermediate concepts to improve final predictions.
> > >
> > > It is important to highlight that the key contribution of this paper is showing that ECBM can efficiently estimate CBMs without retraining, when the training data changes at data, concept level, or concept-label level. As such, ECBM shares the same architecture as CBM and retains CBM’s core essence: enabling concept intervention during test time. These three level changes are independent of the test-time intervene. For ECBM, the concept intervention behavior is identical to that of the original CBM.
> > >
> > > To demonstrate the closeness in test-time intervention capabilities between the model estimated by the ECBM method and the retrain approach, we changed the number of test time intervention concepts and conducted a series of experiments.
> > >
> > > |Concept Number |          |          |
> > > |--------|----------|----------|
> > > |     Method   | Retrain  | ECBM     |
> > > | 0      | 0.51273  | 0.52331  |
> > > | 1      | 0.51505  | 0.52107  |
> > > | 2      | 0.50214  | 0.51616  |
> > > | 3      | 0.48848  | 0.50794  |
> > > | 4      | 0.47924  | 0.50485  |
> > > | 5      | 0.47885  | 0.48878  |
> > > | 6      | 0.46197  | 0.48699  |
> > > | 7      | 0.45029  | 0.47524  |
> > > | 8      | 0.45312  | 0.46290  |
> > > | 9      | 0.44787  | 0.46113  |
> > > | 10     | 0.44823  | 0.45707  |
> > >
> > >
> > > The experimental results further demonstrate that the concept intervention effects of ECBM are sufficiently close to those achieved by retraining the model, validating that ECBM preserves the core advantages of CBMs while enabling greater efficiency in model training.
> > >
> > >
> > >
> > >
> > >
> > > -*Response to Claims And Evidence 3: comparasion with modern CBM architectures*
> > >
> > > To validate the performance of ECBM on soft concepts, we perform the following experiments and the results are shown here.
> > >
> > > The ECBM method can be easily adapted to handle scenarios where concepts take continuous values, such as in CEM, or involve soft labels. By modifying the loss function in Equation 1, the subsequent algorithm can be directly extended to these cases. Furthermore, we demonstrate the performance of ECBM under soft label scenarios in our experiments. And the experiments are still on-going.
> > >
> > >
> > >
> > > -*Response to Other Strengths And Weaknesses*
> > >
> > > Thank you for your suggestion. We will include experimental results related to test-time intervention performance for both the ECBM and retrain models, and we will revise the Limitation and Broader Impact sections accordingly.
> > >
> > >
> > > -*Response to Other Comments Or Suggestions*
> > >
> > > Thanks for your comments. We will check the paper, fix all the errors and define editable CBM explicitly in the camera-ready version.

---

### Official Review · Reviewer_8zrQ · 2025-03-13

**Overall Recommendation:** 3

**Summary:**

This paper improves Concept Bottleneck Models (CBMs) by proposing how to update or “edit” a well-trained CBM. The issues arise when the concept-label level annotations need to be updated, concepts themselves need to be removed and certain data samples used in the training of the model themselves need to be removed. Rather than retraining a CBM from scratch which is computationally expensive, Editable CBM proposes approaches inspired by influence functions to update model parameters on the aforementioned challenges. Theoretical and empirical results demonstrate the effectiveness of the method.

**Claims And Evidence:**

Yes.

**Essential References Not Discussed:**

None

**Experimental Designs Or Analyses:**

Yes. I have detailed my reservations in the sections below.

**Methods And Evaluation Criteria:**

Yes. They are absolutely correct.

**Other Comments Or Suggestions:**

Refer Weakness.

**Other Strengths And Weaknesses:**

Strengths:
1. A very important problem studying the impact of various concept editing mechanisms on trained CBMs without retraining the entire models.
2. Theoretically sound submission with appropriate proofs and justifications

Weakness:
1. The overall paper suffers from confusing notations (refer to Questions) and some unusual choice of variable names which hinder understanding. Specifically,
2. A very important property of CBMs in the original paper [1], was the joint and sequential training paradigm of these models. However, it looks like the authors have only considered one of the settings (probably the joint setting as it gives the best performance). The results should also be demonstrated on the sequential setting to make it truly generalize to all CBMs.
2.1. In addition to 2, I am still not sure about the theoretical basis of "correcting" a jointly trained function $f$ and $g$ would work. If working with a sequential setting, it is easy to see that editing $f$ and $g$ with Hessians is straight-forward, but with joint network training, many of the assumptions are invalid as information has flowed from $f$ to $g$ during training. As an example, in Theorem 4.3, how are we measuring the impact using $\hat{g}$, which should actually be $\hat{g} - H_{\hat{g}}^{-1}$ as the changed params in the concept predictor should influence label predictor as well.
3. In Theorem 4.4, the authors insert 0 valued rows to make up for the dimensional inconsistency and then remove the same rows after the edit to achieve their desired result. This process is very uncertain, with no claims as to if the 0 valued rows lose important information or not. As a suggestion, the authors can perform a small ablation on the before and after effect of packing these rows with numbers - 0, -1, +1, etc. or report Mutual Information-type metrics.
4. (Very minor) Error bars on the Time column are not present - why is that?
5. Why would no concept-level metrics be reported? I understand why F-1 score is used, but editing the model can make a difference in the concept performance as well. In addition, intervention performance as done by [1] are also not reported.
6. (minor) Lastly, the related work section can be expanded to include other approaches to improve CBMs.

[1] CBMs, Koh et al, ICML '19

**Questions For Authors:**

1. What is $R^{d_i}$ in Line 118? It is not defined.
2. In Equation-1, why are the concepts $c_i^j$ in the $log$? This is not consistent with actual CBM, where they are only utilized as a sigmoid.
3. Typo in Line 149 - ff should be $f$ or $\hat{f}$ (unclear).
4. In lines 167-169 what do the authors mean by "if we intervene with the true concepts, the concept predictor $\hat{g}$ fluctuates to $\hat{g_e}$ accordingly". Why is this the case?
5. Why are no other concept-level metrics utilized?

**Relation To Broader Scientific Literature:**

The paper improves CBMs through Influence functions, an important problem never tried before.

**Theoretical Claims:**

Yes. I have detailed my reservations in the sections below.

---

> ### Author Rebuttal · Authors · 2025-04-01
>
> -*Response to W2: the authors have only considered sequential setting (probably the joint setting as it gives the best performance)*
>
> We sincerely thank the reviewer for highlighting the importance of the jointly training mode in CBM. We agree that joint training sometimes leads to higher accuracy in both label and concept predictions.
>
> However, model performance is not the sole priority:
> 1. Compared to joint training, sequential training is more robust under limited data conditions.
> 2. Joint training requires balancing concept loss and task loss, which may result in suboptimal performance for both. Sequential training avoids this trade-off.
> 3. The modular architecture of sequential training allows for easy post-hoc interventions.
>
> Given these advantages, we focus on the editable CBM with sequential training in this work. Our goal is to explore model editing, which represents the unique perspective and theoretical contribution of our study. We believe this approach complements, rather than replaces, CBM performance optimization research.
>
> Finally, due to the complexity and workload of designing algorithms for the three editing levels, it is not feasible to analyze both sequential and jointly training methods within a single paper. Therefore, in this work, we focus on developing editing algorithms for sequentially trained CBMs across three levels and provide theoretical guarantees. In fact, editing jointly trained CBMs using influence functions is also achievable and will be considered in our future work.
>
>
>
> -*Response to W3: Theorem 4.4, the authors insert 0 valued rows*
>
> Thank you for your suggestion.
>
> When a concept is removed, the output dimension of the concept predictor $g$ decreases accordingly. To facilitate estimation, we modify $g$ into $g'$ by inserting a zero-parameter row into its final layer. These parameters remain fixed during training and thus stay zero, ensuring that the model's effective parameter space is strictly a subset of the original space.
>
> In Theorem 4.4, we approximate $g'$ using influence functions, assuming $g'$ continues training within the original parameter space. Consequently, the algorithm's implementation, including parameter updates, remains unaffected regardless of whether the inserted rows are set to 0, 1, or any constant.
>
> -*Response to W4: Error bars*
>
> We apologize for the omission and we have addressed it in the camera-ready version.
>
> -*Response to W5: Concept-level metrics*
>
> Thanks for your suggestion. We will add experiments about concept-level metrics. We perform experiments on CUB and test the concept accuracy of ECBM and retrain. The results are listed below.
>
>
>
> |         | ECBM(%) | Retrain(%) |
> | ------- | ---- | ---------- |
> | Concept   |  93.7112    |     95.1705      |
> | Data        |  94.5184    |    95.2801|
> | Concept-label  | 95.0219 | 95.1407       |
>
> *Table A: Concept Accuracy of ECBM and Retrain under Three Levels
>
> The results show that the accuracy of Retrain and ECBM is very close, with differences generally within a small margin. For example, in the Data and Concept-label levels, the accuracy gap is less than 0.5%. The results show that ECBM not only approximates the accuracy of the retrain method's labels on the test set, but also has a similar performance in terms of concept accuracy.
>
>
> -*Response to W6: The related work section*
>
> We will include more related works in the camera-ready version.
>
>
> ## Questions:
>
> -*Response to Q1: What is $R^{d_i}$?*
>
> Here $R^{d_i}$ represents the space of all
> $d_i$-dimensional real vectors.
>
> -*Response to Q2: Why are the concepts in the log?*
>
> This is the definition of cross-entropy, identical to that in the original CBM, as described in the second paragraph of page 15. The activation function used is the sigmoid function. Note that this is distinct from the definition of the loss function.
>
> -*Response to Q3: ff problem*
>
> Thanks for your correction. It should be f.
>
> -*Response to Q4: In lines 167-169 ... Why?*
>
> This is because if we correct the concept and then retrain the CBM, the concept predictor $\hat{g}$ will be updated to $\hat{g}_e$, which differs from the scenario at test time intervention where the concept predictor remains unchanged. This distinction serves as the key motivation for editing CBMs.
>
> -*Response to Q5: Why are no other concept-level metrics utilized?*
>
> See Response to W5 for reference.

---

> > ### Comment · Reviewer_8zrQ · 2025-04-07
> >
> > The rebuttal addresses most of my pressing concerns. I trust the authors will do a good job incorporating all weaknesses (especially W5) in their final camera-ready version.
> > I have updated my ratings accordingly.

---

> > > ### Author Response · Authors · 2025-04-07
> > >
> > > Dear Reviewer,
> > >
> > > Thank you very much for your thoughtful and constructive feedback, as well as for generously raising the score of our submission. We will continue to revise our paper based on your comments and will ensure that the experimental results related to W5 are included in the camera-ready version.
> > >
> > > Your insightful comments and careful evaluation have been invaluably helpful in improving the quality of our work. We are truly grateful for the time and effort you dedicated to reviewing our paper. Your support and recognition mean a great deal to us.

---

### Decision · Program_Chairs · 2025-05-01

**Decision:**

Accept (poster)

**Comment:**

This paper presents a practical and well-motivated method for editing Concept Bottleneck Models (CBMs) without retraining, using influence function-inspired techniques. The approach is theoretically grounded and empirically validated across key scenarios such as concept and data removal. The rebuttal satisfactorily addressed most concerns, particularly around scalability and effectiveness. In the final version, the authors may consider including empirical results for test-time interventions and broader applicability to newer CBM variants (e.g., CEM, retrieval-based CBMs).